# ReAcTree: Hierarchical Task Planning with Dynamic Tree Expansion using LLM Agent Nodes

## Abstract

Recent advancements in task planning using large language models (LLMs) have made remarkable progress. However, most existing methods, such as ReAct, face limitations when handling complex, long-horizon tasks due to inefficiencies in processing entire tasks through a single sequential decision-making process. To address these challenges, we propose ReAcTree, a hierarchical task planning method that automatically decomposes complex tasks into manageable subgoals within a tree structure. This tree consists of control flow nodes, which manage the execution order of agent nodes, and agent nodes that reason, act, and expand nodes into subgoals to achieve their goals. To further enhance performance, we introduce memory systems: each agent node retrieves goal-specific, agent-level experiences from episodic memory to use as in-context examples, and all agent nodes share and recall information obtained during task execution via working memory. Experiments on the WAH-NL dataset demonstrate that ReAcTree consistently outperforms ReAct across various LLMs and model sizes. For example, when using Qwen2.5 72B, ReAcTree achieves a goal success rate of 63%, significantly surpassing ReAct's 24%.

## 1 Introduction

In recent years, large language models (LLMs) have emerged as powerful tools for task planning, extending the capabilities of traditional approaches such as task and motion planning and hierarchical reinforcement learning. Early works, including Zero-Shot Planner (Huang et al., 2022a) and Say-Can (Ahn et al., 2022), demonstrated the potential of pre-trained LLMs to generate executable skill sequences from natural language instructions without the need for parameter updates. These methods leverage the general procedural knowledge encoded during pre-training, along with in-context examples, to generate contextually appropriate skill sequences. Further advancements introduced feedback mechanisms, wherein LLMs receive textual observations from the environment following skill execution, enabling more adaptive planning (Huang et al., 2022b). Building on these developments, ReAct (Yao et al., 2023) incorporated reasoning capabilities into LLM-based task planners, significantly improving their overall planning performance.

Despite recent advancements, most existing LLM-based methods, including ReAct, still struggle with complex, long-horizon tasks. One key inefficiency stems from attempting to solve entire tasks using a single sequential decision-making process. Zhou et al. (2023) has shown that LLMs perform more effectively when breaking down complex problems into a series of simpler subproblems. Several LLM-based task planning methods have adopted such decomposition strategies (Wang et al., 2023; Sun et al., 2023; Chen et al., 2024; Wang et al., 2024a; Wong et al., 2023). However, many of these approaches either limit the hierarchy to only two stages or are effective only within predefined domains. Another significant inefficiency arises from the accumulation of long text trajectories encompassing reasoning, actions, and observations. Although providing relevant in-context examples has been shown to significantly enhance performance (Wang et al., 2024b; Rubin et al., 2022), as text trajectories grow, it becomes increasingly difficult to include diverse and detailed examples.

In this paper, we propose ReAcTree, a hierarchical task planning method designed to automatically decompose complex tasks into manageable subgoals. ReAcTree constructs a tree structure with two

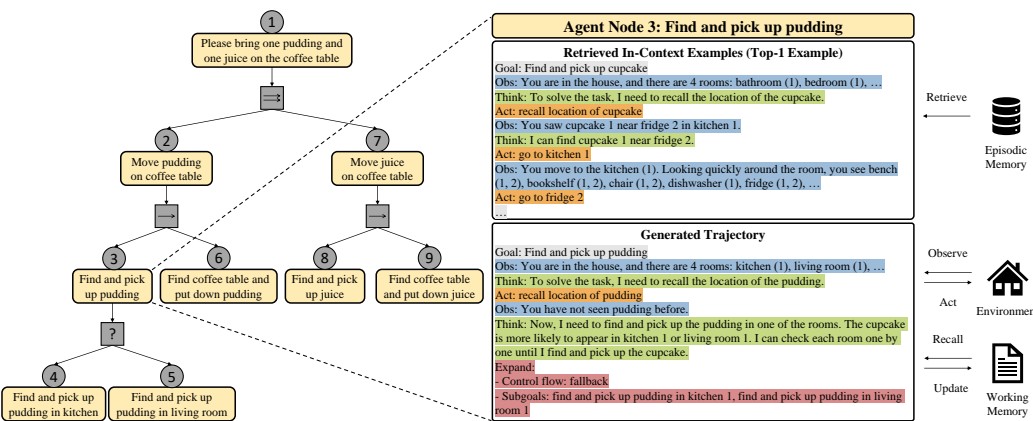

Figure 1: An illustrative example of how ReAcTree generates a tree structure for the natural language instruction: *Please bring one pudding and one juice to the coffee table.* The left side shows the tree structure with agent nodes represented as circles and control flow nodes as squares. Each agent node is annotated with its corresponding natural language goal and execution sequence. The right side presents the text trajectory generated by agent node 3, which includes reasoning, acting, expanding, and retrieval of in-context examples from episodic memory, as well as access to working memory.

primary components: control flow nodes and agent nodes. Control flow nodes, inspired by behavior trees (Colledanchise et al., 2018), are responsible for selecting which agent node to execute at each step. Agent nodes, each functioning as an LLM-based task planner, handle reasoning, acting, and expanding—where expanding involves decomposing goals into subgoals and extending the node into a subtree that includes one control flow node and new agent nodes, each assigned a specific subgoal. To further enhance planning capabilities of ReAcTree, we introduce a memory system. Each agent node retrieves relevant experiences from episodic memory, where past task planning experiences—segmented at the agent level—are stored. This enables the retrieval of experiences closely aligned with the agent node's current goal. Additionally, all agent nodes utilize working memory to update and recall observations during task execution. By sharing observations across all agent nodes, ReAcTree enables more efficient task planning. Figure 1 illustrates an example outcome produced by ReAcTree.

To evaluate the effectiveness of ReAcTree, we conducted extensive experiments using the WAH-NL dataset (Choi et al., 2024; Puig et al., 2021). The WAH-NL dataset involves solving tasks via natural language commands in household environments, each consisting of multiple rooms and long-horizon tasks composed of several subgoals. We implemented a partially observable setting within the VirtualHome simulator (Puig et al., 2018), where agents operate with limited perceptual information, simulating real-world conditions. Our experimental results demonstrate that ReAcTree consistently outperforms the ReAct baseline across various LLMs. Specifically, ReAcTree attains a 53% goal success rate (GSR) with the LLaMA-3.1 70B model, significantly surpassing ReAct's 20%. Furthermore, even with the smaller LLaMA-3.1 8B model, ReAcTree achieves a GSR of 30%, outperforming ReAct on the larger LLaMA-3.1 70B model. These results highlight ReAcTree's superior ability to handle complex tasks through its hierarchical task decomposition. Additionally, our analysis confirms the effectiveness of ReAcTree's memory systems, with both episodic memory and working memory contributing substantially to the observed performance improvements.

In summary, this paper presents the following contributions: (1) We propose ReAcTree, a novel hierarchical task planning algorithm that automatically decomposes complex tasks into manageable subgoals using a tree structure. Each subgoal is handled by an agent node, which performs LLM-based task planning through reasoning, acting, and expanding—decomposing tasks further into smaller subgoals. (2) We introduce memory systems, including episodic memory and working memory, to enhance the performance of ReAcTree by enabling the retrieval of relevant agent-level experiences and facilitating the sharing of information across agent nodes during task execution. (3) We conduct extensive experiments in a partially observable setting to demonstrate the superiority of ReAcTree over the baseline model, ReAct. Additionally, we will release our code to support future research (anonymous code for review is available at figshare.com/s/97dd86282bd050f66d11).

## 2 RELATED WORKS

**LLM-based Task Planning.** Since the inception of LLMs, researchers have continuously explored their emerging capabilities. In particular, their reasoning abilities have recently been applied to decision-making processes for embodied agents, such as robots. Huang et al. (2022a) were among the first to demonstrate that LLMs can infer task procedures for embodied agents without requiring additional training, while SayCan (Ahn et al., 2022) introduced more robust planning by integrating visual affordances from a robot's perspective. Following these foundational studies, various approaches have evolved. For example, Inner Monologue (Huang et al., 2022b) enhanced procedural inference by allowing agents to revise plans based on textual feedback, such as the success of individual steps, object detection results, and Q&A with humans. Additionally, ReAct (Yao et al., 2023) adopted Chain-of-Thought prompting (Wei et al., 2022), enabling agents to explicitly reason through tasks, leading to improved planning performance.

**Hierarchical Task Planning with LLMs.** Recently, research has increasingly focused on hierarchical task planning with LLMs to address complex, long-horizon tasks. These approaches often utilize bi-level hierarchies, where an overall plan is refined through next-step decisions, as demonstrated in DEPS (Wang et al., 2023) and AdaPlanner (Sun et al., 2023). Classical planning integrated with LLMs, such as in Ada (Chen et al., 2024), offers flexibility but remains constrained by predefined environments. While these methods enable interactive and adaptive planning, they are limited in hierarchical depth, unlike ReAcTree, which employs a deeper structure that allows for dynamic subgoal decomposition. MOSAIC (Wang et al., 2024a) proposed a task planner for robots collaborating with humans in cooking. This planner organizes reasoning into a tree structure with LLM agents, where nodes determine whether to define subtasks or seek clarification from the human. However, this study relies on a manually designed structure specific to cooking tasks and lacks generalizability to other domains. In contrast, our approach dynamically generates behavior trees for various tasks in household environments.

**Tree Search-Based Planning with LLMs.** Another line of research explores tree search-based methods. These approaches construct action/thought trees to look ahead and evaluate multiple future paths to select the best next step. Tree of Thoughts (Yao et al., 2024) and Graph of Thoughts (Besta et al., 2024) demonstrate how systematic tree search improves decision-making by exploring and evaluating reasoning paths. LLM-MCTS (Zhao et al., 2024) extends these ideas to task planning by constructing action trees and applying Monte Carlo Tree Search but relies on ground-truth transition functions, limiting its applicability to real-world scenarios. Tree-Planner (Hu et al., 2023) constructs action trees, executes actions, and observes outcomes, avoiding explicit state prediction. However, it assumes reversible actions (e.g., *pick–place*), which are not feasible for irreversible actions like slicing. While these methods focus on constructing action or thought trees to search for the best trajectory, ReAcTree takes a fundamentally different approach by building an LLM agent tree. It divides complex goals into manageable subgoals, dynamically assigns specialized agent nodes, and determines their execution order using control flow strategies such as sequence and fallback nodes.

## 3 PRELIMINARIES

**Problem Formulation.** We consider the task planning problem as a sequential decision-making problem aimed at achieving a goal $g$ expressed in natural language. At each time step $t$, the agent has access the context $c_t = (o_1, a_1, o_2, a_2, \cdots, a_{t-1}, o_t)$, where $o_i$ and $a_i$ represent the observation and action at each previous time step $i$, respectively. The objective of agent is to generate the next appropriate action $a_t$ based on the context $c_t$, with the aim of eventually achieving the goal $g$.

**ReAct (Yao et al., 2023).** ReAct is a representative method that leverages a pre-trained LLM, $p_{LLM}$, to solve the task planning problem by interleaving reasoning and action execution. The LLM defines the action policy as follows: $a_t \sim p_{LLM}(\cdot | P, g, c_t)$, where $P = (P_{sys}, P_{ic})$ is the initial prompt, composed of a system prompt, $P_{sys}$, and in-context examples, $P_{ic}$. The key idea of ReAct is to use the augmented action space, $\hat{\mathcal{A}}_t = \mathcal{A}_t \cup \mathcal{L}$, where $\mathcal{A}_t$ is the set of executable skills available at time $t$, and $\mathcal{L}$ is the language space representing reasoning steps or thoughts. If $a_t \in \mathcal{A}_t$, the agent executes the action and obtains a text observation from the environment. If $a_t \in \mathcal{L}$, it is called a thought or reasoning trace, which aids in the logical inference of the LLM. In this case, the agent does not receive a new observation from the environment, i.e., $o_{t+1} = \phi$.

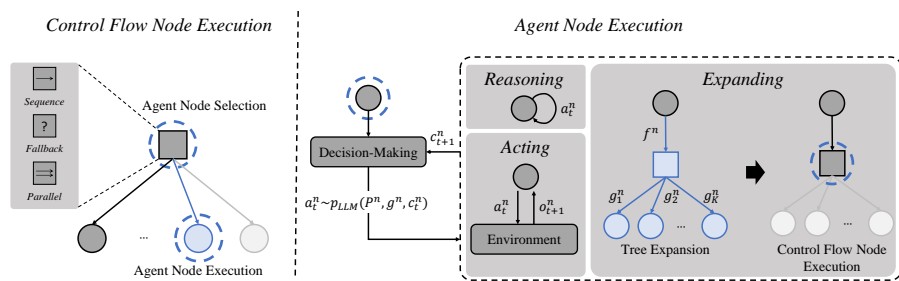

Figure 2: Illustration of control flow node execution and agent node execution in ReAcTree.

# 4 REACTREE

In this section, we introduce ReAcTree, a hierarchical task planning algorithm designed to efficiently manage complex tasks by decomposing them into subgoals using a tree structure. The tree is composed of two types of nodes: control flow nodes and agent nodes. Control flow nodes determine which agent node should be executed next. Each agent node operates as an LLM-based task planner, with its goal expressed in natural language. A key feature of ReAcTree is the augmentation of each agent node's action space to include not only reasoning and acting but also expanding, which allows the decomposition of its goal into subgoals and the creation of a subtree. This subtree consists of a child control flow node and grandchild agent nodes, each assigned a specific subgoal. Figure 2 illustrates the node execution process for both control flow nodes and agent nodes. The details are described in Section 4.1.

We introduce a memory system designed to enhance the performance of ReAcTree. Episodic memory is employed to construct in-context examples for each agent node. Each node retrieves relevant, agent-level experiences from episodic memory. Additionally, working memory stores information gathered by agent nodes through their interactions with the environment during task execution. This working memory is shared among all agent nodes, enabling them to access and utilize the stored information as needed. Further details are provided in Section 4.2.

## 4.1 REACTREE ALGORITHM

ReAcTree is an LLM-based hierarchical task planning algorithm that generates a tree structure $T$, composed of control flow nodes and agent nodes, to achieve a task goal $g$ expressed in natural language. In the following, we offer a detailed explanation of the execution process for both control flow nodes and agent nodes, as well as the overall ReAcTree algorithm.

**Control Flow Nodes.** Control flow nodes, inspired by behavior trees (Colledanchise et al., 2018), manage the selection and execution of agent nodes within the tree. Each control flow node has child nodes, which are agent nodes assigned specific natural language goals. Executing a control flow node involves executing its child nodes sequentially. Each child node reports its execution status (success or failure) to its parent, the control flow node. Based on the status of its child nodes, the control flow node decides whether to proceed with the next child node or to return its own status to its parent node. The specific behavior of the control flow node depends on its type.

ReAcTree employs three types of control flow nodes. The first is the *sequence node* ($\rightarrow$), which executes its child nodes in order. It returns success if all child nodes succeed; however, if any child node fails, the sequence node returns failure. The second type is the *fallback node* (?), which also executes its child nodes sequentially but returns success as soon as any child node succeeds. If none of the child nodes succeed, it returns failure. The third type is the *parallel node* ($\Rightarrow$), a variation of the traditional parallel node concept. While the traditional definition of a parallel node involves executing child nodes simultaneously, in ReAcTree, simultaneous execution is not possible. Instead, the parallel node executes its child nodes independently, regardless of their individual success or failure. After all nodes are executed, the outcomes are aggregated according to a predefined policy to determine the overall success or failure. This node is particularly useful in tasks such as household chores, where multiple subgoals, like picking and placing objects, need to be executed sequentially without interruption, yet their success or failure does not immediately halt the overall task.

**Agent Nodes.** Each agent node operates as an LLM-based task planner with a specific natural language goal, responsible for making sequential decisions to achieve that goal. These decisions include *acting*, *reasoning*, and *expanding*. Similar to the ReAct framework, *acting* refers to executing actions and receiving feedback in the form of textual observations, while *reasoning* enables logical inference. Additionally, *expanding* decomposes the agent node's current goal into subgoals and combines them with a specific control flow strategy to further expand the tree structure.

An agent node $n$ is first initialized with a goal $g^n$. Executing the agent node involves sequential decision-making, where the agent returns its status (whether the goal was achieved) after execution. More specifically, when executing an agent node, an initial prompt $P^n = (P_{sys}, P_{ic}^n)$ is constructed, where $P_{sys}$ is the system prompt, and $P_{ic}^n$ consists of in-context examples specific to agent node $n$. At each time step $t$, the agent node accesses its context $c_t^n = (o_1^n, a_1^n, o_2^n, a_2^n, \ldots, a_{t-1}^n, o_t^n)$, where $o_i^n$ and $a_i^n$ represent the observation and action at each previous time step $i$. The action policy is then defined using the pre-trained LLM, $P_{LLM}$, as: $a_t \sim p_{LLM}(\cdot | P^n, g^n, c_t^n)$. A key feature of ReAcTree is its extended action space, $\hat{\mathcal{A}}_t^n = \mathcal{A}_t^n \cup \mathcal{L} \cup \mathcal{E}$, where $\mathcal{A}_t^n$ represents the set of executable skills at time $t$ (e.g., *move*, *pick*, *turn on*); $\mathcal{L}$ is the language space, used for generating self-reasoning text and specifying subgoals in natural language; and $\mathcal{E} = \mathcal{F} \times \mathcal{L}$ is the expand space, where $\mathcal{F}$ represents the set of control flow types.

If the action $a_t^n \in \mathcal{A}_t^n$ or $a_t^n \in \mathcal{L}$, the agent operates as in the ReAct framework, either performing actions or engaging in reasoning. However, if $a_t^n \in \mathcal{E}$, the agent expands the tree structure by adding a control flow node and handing over execution to it. In this case, the action is represented as $a_t^n = (f^n, [g_1^n, \ldots, g_K^n])$, where $f^n$ is the control flow type and $g_i^n$ are the subgoals expressed in natural language. A control flow node $n_f$, with type $f^n$, is added as a child of node $n$, and agent nodes $n_i$ are initialized with their corresponding subgoals $g_i^n$ and added as children of $n_f$. The agent then waits for $n_f$ to complete execution. The agent node terminates when one of the following occurs: generating the action *done* (resulting in success), generating action *failure* or reaching the maximum decision count (both resulting in failure), or completing the execution of the control flow node (returning success or failure based on its result).

**Overall Algorithm.** The overall process of ReAcTree is outlined in Algorithm 1. It begins by initializing the root agent node $n$ with the goal $g$ (line 1). The agent node is then executed until meets a termination condition $TC$ (line 2). If $TC$ is *done*, it indicates that the agent has successfully achieved the goal, and the agent node returns *Success* status (lines 3-4). If $TC$ is *failure* or the maximum decision count is reached, it indicates failure, and the agent node returns *Failure* status (lines 5-6). If $TC$ is *Expand*, the agent has decided to decompose the goal into subgoals. The expansion information $(f^n, [g_1^n, \cdots, g_K^n])$ is retrieved (line 8), where $f^n$ is the control flow type and $g_i^n$ are the subgoals. A control flow node $n_f$ is initialized (line 9) and added as a child of the current agent node $n$ (line 10). For each subgoal, a new agent node $n_i$ is initialized (line 12) and added as a child of the control flow node $n_f$ (line 13). The con-

---

**Algorithm 1** ReAcTree Algorithm

**Input:** Natural language goal $g$

1: $n \leftarrow$ INITAGENTNODE($g$)
2: $TC \leftarrow$ EXECAGENTNODE($n$)
3: **if** $TC$ is *done* **then**
4:     **return** Success
5: **else if** $TC$ is *failure* or max decision **then**
6:     **return** Failure
7: **else if** $TC$ is Expand **then**
8:     $(f^n, [g_1^n, \cdots, g_K^n]) \leftarrow$ EXPANDINFO
9:     $n_f \leftarrow$ INITCTRLFLOWNODE($f^n$)
10:    ADDCHILD($n, n_f$)
11:    **for** $i = 1$ to $K$ **do**
12:       $n_i \leftarrow$ INITAGENTNODE($g_i^n$)
13:       ADDCHILD($n_f, n_i$)
14:    **end for**
15:    **return** EXECCTRLFLOWNODE($n_f$)
16: **end if**

---

trol flow node $n_f$ is then executed (line 15), where it selects one of its child agent nodes based on its control flow strategy. The selected agent node is then executed, and its status is returned to the control flow node. This process repeats until the control flow node completes execution, after which it returns its resulting status (success or failure).

## 4.2 MEMORY SYSTEMS

To enhance the performance of ReAcTree, we introduce two complementary memory systems: *episodic memory* and *working memory*. Episodic memory is used to retrieve relevant past agent-level experiences and incorporate them as in-context examples before each agent node begins its

decision-making process. On the other hand, working memory is designed to share key observations, such as the latest location of movable objects, across agent nodes during task execution. In the following, we provide a detailed explanation of how each memory is utilized and integrated with agent nodes of ReAcTree.

**Episodic Memory.** Episodic memory, $M_{ep}$, stores the agent-level experiences of all ReAcTree agent nodes involve in successfully completing tasks. The agent-level experience of an agent node $e$, with a goal sentence $g^e$ and a final time step is $T$, is defined as $(t^e, v^e, s^e)$. Here, $t^e = (g^e, o_1^e, a_1^e, \ldots, o_T^e, a_T^e)$ represents the full text trajectory, where $o_t^e$ and $a_t^e$ denote the observation and action at each time step $t$. The vector $v^e = f_{\text{sen}}(g^e)$ is the goal embedding generated using the sentence embedding model $f_{\text{sen}}$. Finally, $s^e$ indicates the termination state of the agent node, categorized as *success*, *failure*, or *expand*.

Before an agent node begins its decision-making process, it retrieves in-context examples by comparing its goal $g^n$ to the stored goals in episodic memory using cosine similarity. Specifically, the agent embeds its goal as $v^n = f_{\text{sen}}(g^n)$ and computes the similarity with stored embeddings: $sim(v^n, v^e) = v^n \cdot v^e / (|v^n||v^e|), \forall v^e \in M_{ep}$. Based on the similarity scores, the system retrieves the top $k$ examples until a predefined token limit is reached. To handle cases where multiple stored experiences yield identical similarity scores, the termination state $s$ is used to break ties. When similarity scores are tied, examples are sampled uniformly across termination states, *success*, *failure*, or *expand*. This mechanism promotes diversity in the retrieved in-context examples.

Since ReAcTree is designed to decompose complex goals into manageable subgoals, it generates concise trajectories tailored to specific tasks, unlike the monolithic trajectories used by ReAct. For instance, while ReAct stores a single trajectory for *Bring one pudding and onne juice to the coffee table,* ReAcTree breaks this down into subgoals like *find and pick up pudding in kitchen,* producing shorter and more specific examples. This enables ReAcTree to retrieve highly relevant and focused trajectories from episodic memory, enhancing its task-planning effectiveness.

**Working Memory.** Working memory serves as a shared repository for storing and recalling key observations during task execution. In this paper, working memory focuses on tracking the latest locations of movable objects to minimize redundant interactions with the environment and mitigate potential hallucinations by providing accurate, environment-specific data.

Working memory is integrated into agent nodes of ReAcTree through two key mechanisms. First, the executable skill set $\mathcal{A}_t^n$ is augmented with special actions like *recall location of <movable object>*, which are predefined for all movable objects in the environment, enabling agents to retrieve stored object locations directly from working memory instead of interacting with the environment. Second, working memory is automatically updated whenever an agent interacts with the environment and detects movable objects. For instance, if an agent opens a fridge and observes juice, working memory updates the location of juice as near fridge for future use. Such interactions can also be viewed as an extension of how tool usage is integrated into language models, as discussed in (Schick et al., 2024).

## 5 EXPERIMENTS

### 5.1 EXPERIMENTAL SETUP

**Datasets and Simulators.** We primarily evaluate LLM-based task planning using the WAH-NL dataset within the VirtualHome simulator, as introduced in LoTa-Bench (Choi et al., 2024). The WAH-NL dataset, derived from the Watch-and-Help dataset (Puig et al., 2021), was originally designed for human-agent collaboration. It has been adapted for autonomous agents, with natural language commands collected via crowdsourcing. Each data instance comprises a natural language instruction, an initial environment setup, and a goal condition. The dataset includes 250 tasks in the training set and 100 tasks in the test set, covering five task categories: *Setup a dinner table*, *Put groceries*, *Prepare a meal*, *Wash dishes*, and *Prepare snacks*. Each task involves multiple subgoals, making WAH-NL well-suited for evaluating task planning in complex, long-horizon scenarios.

All experiments are conducted in the VirtualHome simulator, which provides a simulated household environment where agents can perform various actions such as *pick up*, *open*, *close*, *turn on*, *put down*, and *go to*. Unlike the fully observable setting in LoTa-Bench, we implement a partially observable setting in VirtualHome to simulate real-world conditions. In this configuration, the agent

has access to limited observations. Specifically, objects are identified by both their class name and instance (e.g., *cup 1*, *cup 2*). After executing an action, the agent receives a text-based observation reflecting its surroundings. For instance, after performing a *go to room* action, the agent can observe all receptacles (e.g., tables, shelves) within the room. If the action is *go to object*, the agent observes nearby visible objects, provided they are not inside closed receptacles. For actions like *pick up* or *open*, the agent receives feedback indicating whether the action was successful or not. A more detailed implementation of the partially observable setting is described in Appendix A, and we will release our code and modified simulator for reproducibility (anonymous code for review is available at figshare.com/s/97dd86282bd050f66d11).

**Evaluations.** To assess the effectiveness of ReAcTree, we compare it against ReAct, with both utilizing the same underlying LLMs. We evaluate performance using the goal success rate (GSR), defined as the percentage of tasks in which the agent successfully achieves the given task goal, and the subgoal success rate (SSR), which is the ratio of successfully completed subgoals to the total number of subgoals. Both methods were evaluated under the same maximum decision count, set to a sufficiently large value of 199 for all experiments, ensuring fair comparisons. All results are averaged across the entire test set.

**Episodic Memory Construction.** To construct the episodic memory, we first collect human-annotated text trajectories for WAH-NL in both ReAct and ReAcTree formats. Specifically, one random task from each task category is selected for trajectory collection, resulting in five tasks for WAH-NL. Subsequently, we run both ReAct and ReAcTree on the training set using the LLaMA-3 70B model (Dubey et al., 2024). Only the trajectories of tasks that are successfully completed are added to the episodic memory. To encode the agent's goals, we use Sentence BERT (Reimers & Gurevych, 2019)for sentence embeddings.

## 5.2 MAIN RESULTS

Table 1 summarizes the comparison between ReAcTree and ReAct across various LLMs, including LLaMA 3, LLaMA 3.1 (Dubey et al., 2024), Qwen2, Qwen2.5 (Yang et al., 2024), Mistral (Jiang et al., 2023), Gemma (Team et al., 2024), and GPT-4o (see Appendix B for the complete list). Performance was evaluated using GSR and SSR metrics, with the primary results presented in Table 1 and additional results provided in Appendix D. Since direct access to log probabilities is not available for GPT-4o, its implementation was slightly modified. These details are in Appendix C.

Overall, ReAcTree consistently outperforms ReAct across all model types and sizes. For instance, with the Qwen2.5 72B model, ReAcTree achieves a GSR of 63.00%, compared to ReAct's GSR of 24.00%. This consistent performance improvement highlights the effectiveness of ReAcTree's task decomposition approach. The results clearly demonstrate the advantage of breaking tasks into manageable subgoals, enabling more efficient task completion, particularly in complex, long-horizon tasks. In terms of computational cost with this model, ReAcTree incurs a higher average decision steps (75.00) compared to ReAct (58.08) for tasks where both methods succeed. However, its hierarchical design resets the input prompt for each agent node, mitigating computational overhead.

Table 1: Performance comparison of ReAct and ReAcTree across different models and sizes. WM (✓) indicates the use of working memory for task planning, while (✗) denotes its absence.

| Metric | Method | WM | LLaMA 3.1 8B | LLaMA 3.1 70B | Qwen2.5 7B | Qwen2.5 72B | Mistral 7B | GPT-4o |
|---|---|---|---|---|---|---|---|---|
| GSR (%) | ReAct | ✗ | 10.00 | 23.00 | 6.00 | 20.00 | 3.00 | - |
| | | ✓ | 15.00 | 20.00 | 13.00 | 24.00 | 6.00 | 33.00 |
| | ReAcTree | ✗ | 17.00 | 30.00 | 12.00 | 51.00 | 10.00 | - |
| | | ✓ | **30.00** | **53.00** | **35.00** | **63.00** | **20.00** | **46.00** |
| SSR (%) | ReAct | ✗ | 29.45 | 49.35 | 24.55 | 45.13 | 21.72 | - |
| | | ✓ | 42.27 | 52.87 | 35.77 | 48.43 | 31.03 | 57.30 |
| | ReAcTree | ✗ | 53.28 | 62.83 | 42.43 | 70.95 | 37.43 | - |
| | | ✓ | **60.60** | **73.43** | **58.80** | **79.37** | **43.83** | **62.77** |

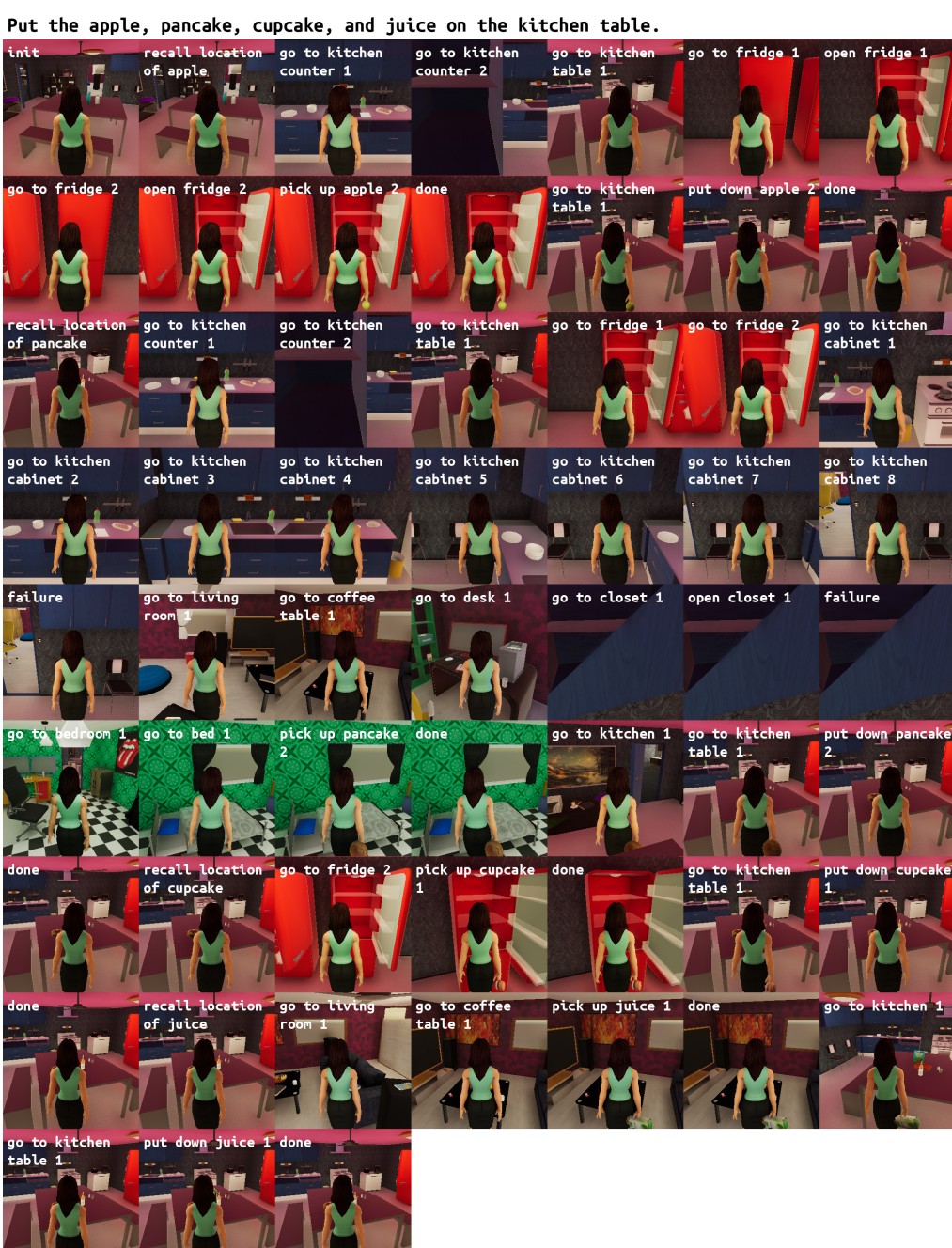

Figure 3: Success case of ReAcTree on the WAH-NL dataset using LLaMA 3 70B.

Moreover, the results suggest that ReAcTree with smaller models can perform even better than larger models running ReAct. For instance, ReAcTree using the LLaMA 3.1 8B model achieves a GSR of 30.00%, outperforming ReAct's GSR of 20.00% when using the significantly larger LLaMA 3.1 70B model. This performance improvement can be attributed to ReAcTree's ability to decompose tasks into simpler subproblems, allowing smaller models to efficiently manage less complex components while retrieving more relevant in-context examples. By focusing on these smaller, more manageable tasks, ReAcTree effectively compensates for the limitations of smaller models.

We also present qualitative results comparing failed trajectories in ReAct with successful ones in ReAcTree using the LLaMA 3 70B model. The natural language instruction for the task is, *Put the*

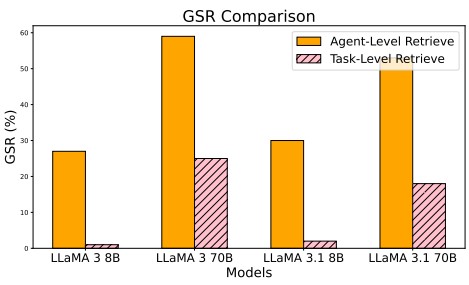 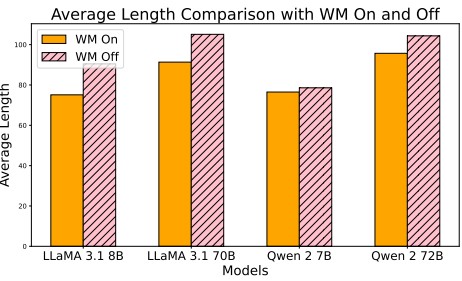

Figure 4: (a) GSR for retrieve methods and (b) Working memory average decision length.

*apple, pancake, cupcake, and juice on the kitchen table*, which requires locating and moving multiple objects. ReAcTree breaks down the instruction into specific subgoals, such as *move the apple to the kitchen table*, *move the pancake to the kitchen table*, *move the cupcake to the kitchen table*, and *move the juice to the kitchen table*, using a persistent sequence node. Notably, during the search for the pancake, ReAcTree utilizes a fallback node to explore different rooms. In contrast, ReAct struggles to locate the pancake, as it only searches the initial room, the kitchen. Furthermore, ReAct misidentifies a pudding as the pancake, and redundantly moves an already relocated apple. Figure 3 illustrates the successful trajectories from ReAcTree, while Figure 5 in Appendix G.1 shows the failed trajectories from ReAct. The full text trajectories for both ReAcTree and ReAct are provided in List 11 and List 12 in Appendix H.1, respectively. For a more detailed analysis of ReAcTree's failure cases, including examples and common error patterns, please refer to Appendix E.

## 5.3 EPISODIC MEMORY

In this section, we investigate the impact of episodic memory on the performance of ReAcTree. ReAcTree utilizes episodic memory by retrieving similar past experiences at the agent level, providing in-context examples to each agent. This allows agents to reference relevant trajectories. To evaluate the effectiveness of this approach, we compare it with task-level example retrieval.

In the agent-level retrieval method, each agent independently retrieves the most contextually similar experiences from episodic memory, irrespective of the task. This enables each agent to draw from a diverse set of agent-specific experiences. In contrast, task-level retrieval selects the most similar past task and uses the entire sequence of trajectories from all agents involved in solving that task as the in-context example. This approach gives the agent access to a complete set of trajectories used to solve a previous task that closely resembles the current one.

The performance results are shown in Figure 4a. Our findings demonstrate that agent-level retrieval yields better performance, especially as model size decreases. Smaller models benefit more from decomposing complex tasks into simpler subgoals and retrieving contextually similar examples for each subgoal, resulting in more efficient problem-solving. By contrast, task-level retrieval, which provides a broader set of trajectories, may offer a wider contextual understanding but can be less effective for fine-tuned decision-making at the subgoal level.

## 5.4 WORKING MEMORY

In this section, we analyze the impact of the working memory component on the performance of both ReAcTree and ReAct models. As shown in Table 1, incorporating working memory consistently leads to performance improvements across various LLMs and model sizes. This enhancement is observed not only in ReAcTree but also in the ReAct baseline. In most cases, the performance improvement occurs because the agent retrieves stored object locations from working memory using the *recall location of* action, rather than generating locations through text. This finding aligns with previous research on tool usage in LLMs (Schick et al., 2024), which also demonstrated similar benefits.

In ReAcTree, working memory additionally contributes to efficiency improvements through information sharing between agent nodes. When one agent node identifies the location of an object or

gathers relevant information, it updates this data in the working memory. Other agent nodes can then access this information, eliminating redundant searches and actions. This inter-node communication streamlines decision-making and accelerates task completion.

To quantify the efficiency gains provided by working memory, Figure 4b illustrates the average number of decision-making steps required to complete tasks that are commonly solved both with and without working memory. The figure shows that across all models, using working memory significantly reduces the number of steps needed to achieve task success. This reduction indicates that agents can plan more efficiently and execute tasks with fewer unnecessary actions by leveraging the shared information stored in working memory.

## 5.5 EXPERIMENTS ON ALFRED

**Datasets and Simulators.** We further extended our experiments to include the ALFRED dataset (Shridhar et al., 2020), which, similar to WAH-NL, involves task planning based on LLM in a household environment setting. This dataset provides NL instructions, an initial state of the environment, and a goal condition related to various household tasks. The task types in this experiment are similar to those used in LoTa-Bench (Choi et al., 2024). We used the AI2-THOR simulator, which corresponds to the ALFRED dataset.

**Evaluations.** The ALFRED is consists of three sets: *train, valid-seen, valid-unseen*. The planning performance of both ReAct and ReAcTree models was evaluated using the *valid-seen* set. We evaluated using only 30% of the *valid-seen* set, which consists of 208 task trajectories. We measured planning performance using the goal success rate (GSR) for ALFRED. GSR is determined by whether the goal condition of the dataset matches the final state of the simulator after completing the planning.

**Experimental Results.** Table 2 summarizes the comparison between ReAct and ReAcTree across different LLM models. The proposed ReAcTree method demonstrates a 2.88% performance improvement over ReAct in both the LLaMA 3 70B and Qwen2 72B models. For smaller models, ReAcTree also outperforms ReAct by 2.93% in the LLaMA 3 8B model. However, in the Qwen2 7B model, ReAct sur-

Table 2: Goal success rate comparison of ReAct and ReAcTree on ALFRED with working memory.

| Method | LLaMA 3 | | Qwen2 | |
|---|---|---|---|---|
| | 8B | 70B | 7B | 72B |
| ReAct | 5.77 | 16.35 | **7.73** | 14.90 |
| ReAcTree | **8.70** | **19.23** | 4.33 | **19.71** |

passes ReAcTree with a 3.4% advantage. Overall, except for the Qwen2 7B model, ReAcTree consistently improves task planning performance. Visualizations and full text trajectories for a specific ALFRED task where ReAcTree succeeds and ReAct fails are provided in Appendix G.2 and H.2, respectively. Please refer to Appendix F for details on the experiments with the ALFRED.

## 6 CONCLUSION

In this paper, we presented ReAcTree, a hierarchical task planning method that automatically decomposes complex tasks into subgoals. Our approach uses a tree structure that integrates control flow nodes for managing tasks and agent nodes to perform sequential decision-making through an extended action space, which includes not only *reasoning* and *acting*, but also *expanding*. Additionally, we introduced memory systems, with episodic memory retrieving agent-level experiences and working memory sharing observations between nodes. Our experiments on the WAH-NL dataset show that ReAcTree consistently outperforms the ReAct baseline, achieving 35% and 63% success rates with Qwen2.5 7B and 72B, respectively, compared to 24% with ReAct using Qwen2.5 72B. We also demonstrated the effectiveness of our memory systems.

One limitation of our approach is that the observations obtained after executing actions rely on ground truth from a simulator, with pre-specified formats for certain action types. These observations may not always be optimal for solving the current subgoal. Additionally, while ReAcTree shows significant improvement with smaller models, further enhancements are still needed. For future work, we will explore methods to gather more relevant observations from the environment after action execution. We will also focus on improving the performance of ReAcTree with smaller models.

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

# A    IMPLEMENTATION DETAILS FOR PARTIALLY OBSERVABLE SETTINGS

As outlined in Section 5, we adapt LoTa-bench (Choi et al., 2024) to support partial observability. Oracle-generated partial observation feedback signals are incorporated into the context information provided to the LLM. Tables 3 and 4 present detailed example trajectories for each action type under partial observation. In this setup, the agent is aware only of the receptacles in the current room and can only see the objects visible within its ego-centric view. VirtualHome is a multi-room apartment simulator, so it has a "*go to room*" action type. On the other hand, AI2-THOR is a single-room studio apartment simulator, so it does not use the "*go to room*" action type because there is no need to move between rooms. Table 3 illustrates an example of partial observation feedback from the VirtualHome simulator, while Table 4 provides a similar example from the AI2-THOR simulator.

Table 3: Action types and sample text observations in the VirtualHome simulator.

| Action Type | Action | Text Observation |
| --- | --- | --- |
| *initialization* | - | You are in the house, and there are 4 rooms: bathroom (1), bedroom (1), kitchen (1), living room (1). You are in the middle of a bathroom (1). Looking quickly around the room, you see bathroom cabinet (1), bathroom counter (1), faucet (1), sink (1), toilet (1), towel rack (1), washing machine (1). |
| *go to room* | go to kitchen 1 | You move to the kitchen (1). Looking quickly around the room, you see chair (1, 2, 3, 4), dishwasher (1), faucet (1), fridge (1), kitchen cabinet (1, 2, 3, 4, 5, 6, 7, 8), kitchen counter (1, 3), kitchen table (1), microwave oven (1), nightstand (1, 2), sink (1), sofa (1), stove (1), toaster (1). |
| *go to receptacle* | go to kitchen table 1 | You arrive at the kitchen table (1). You see apple (2), book (1), bowl (2, 3), cereal (1), chair (1, 2, 3, 4), chocolate syrup (1), condiment bottle (1, 2), condiment shaker (1, 2), cupcake (1, 2), cutlery fork (2, 3, 4), cutlery knife (3, 5, 6), juice (1), kitchen table (1), pancake (1), pear (1), plate (2, 3, 4, 5), pudding (2), water glass (1, 2, 4) |
| *pick up object* | pick up juice 1 | You pick up juice. You hold juice (1). |
| *put down object* | put down apple 1 | You put down apple on coffee table. |
| *open receptacle* | open fridge 1 | You open fridge. You see bench (1), cupcake (1), fridge (1, 2), juice (1). |
| *close receptacle* | close dishwasher 1 | You close dishwasher. |
| *turn on receptacle* | turn on dishwasher 1 | You turn on dishwasher. |

Table 4: Action types and sample text observations in the AI2-THOR simulator.

| Action Type | Action | Text Observation |
|---|---|---|
| *initialization* | - | You are in the house, and you arrive at the bathroom, Looking quickly around the room, you see TowelHolder (1), Toilet (1), ToiletPaperHanger (1), Sink (1), SinkBasin (1), Shelf (1, 2, 3), HandTowelHolder (1), GarbageCan (1), BathtubBasin (1). |
| *go to receptacle* | go to BathtubBasin (1) | You arrive at BathtubBasin (1). You see SoapBar (2), Cloth (1), Bathtub (1), BathtubBasin (1). |
| *pick up object* | pick up SoapBar (2) | You pick up SoapBar (2). You see SoapBar (2), Cloth (1), Bathtub (1), BathtubBasin (1). I am holding SoapBar (2) now. |
| *put down object* | put down SoapBar (2) | You put SoapBar (2) on Shelf (1). You see SoapBar (1, 2, 3), Shelf (1), Candle (2). |
| *open receptacle* | open Microwave (1) | You open Microwave (1). You see StoveKnob (1, 2, 3, 4), StoveBurner (3), Potato (1), Pan (1, 2), Microwave (1), Bowl (3). |
| *drop object* | put down DishSponge (1) | You put down failed, drop it. You see SoapBar (3), Sink (1), SinkBasin (1), HandTowel (1), Faucet (2), DishSponge (1). |
| *close receptacle* | close Microwave (1) | You close Microwave (1). You see StoveKnob (1, 2, 3, 4), StoveBurner (3), Pan (1, 2), Microwave (1). |
| *turn on receptacle* | turn on Microwave (1) | You turn on Microwave (1). You see Microwave (1), Drawer (5, 6), Cabinet (7, 8, 10). |
| *turn off receptacle* | turn off Microwave (1) | You turn off Microwave (1). You see Microwave (1), Drawer (5, 6), Cabinet (7, 8, 10). |

## B  LANGUAGE MODELS

Table 5 lists the exact language model names used in the experiments.

Table 5: List of language models used in the experiments. Model names are either from HuggingFace model hub or OpenAI API.

| Class | Model name | Model size |
|---|---|---|
| LLaMA 3 | meta-llama/Meta-Llama-3-8B | 8B |
| | meta-llama/Meta-Llama-3-70B | 70B |
| LLaMA 3.1 | meta-llama/Llama-3.1-8B | 8B |
| | meta-llama/Llama-3.1-70B | 70B |
| Qwen2 | Qwen/Qwen2-7B | 7B |
| | Qwen/Qwen2-72B | 72B |
| Qwen2.5 | Qwen/Qwen2.5-7B | 7B |
| | Qwen/Qwen2.5-72B | 72B |
| Mistral | mistralai/Mistral-7B-v0.3 | 7B |
| Gemma | google/gemma-7b | 7B |
| GPT-4o | gpt-4o-2024-05-13 | - |

## C  GPT-4O IMPLEMENTATION DETAILS

GPT-4o was implemented slightly differently due to the lack of access to log probabilities. At each time step $t$, the next action $a_t$ was generated using the OpenAI API. If $a_t$ was a *reasoning* action, it was handled in the same way as with other models. For *acting* actions, we first checked if $a_t$ was a valid skill in $\mathcal{A}_t$. If it was valid, it was processed as usual. If not, the following corrective observation was provided to GPT-4o:

> You should only output sentences that begin with Think, Act, or Expand. If you output Act, you should use one of actions of this list: [go to, pick up, put down, open, close, turn on, recall location of, done, failure].

Similarly, for *expanding* actions, if $a_t$ was valid, it proceeded as normal. If not, the same corrective observation was added, prompting GPT-4o to refine its decision. This method ensured GPT-4o remained consistent with the task's action space, despite the lack of log probability access.

# D  EXTENDED RESULTS

Table 6: Performance comparison of ReAct and ReAcTree on WaH dataset. GSR and SSR represent goal success rate and subgoal success rate, respectively. WM represents the use of the working memory.

| Metric | Method | WM | LLaMA 3 | | Qwen2 | | Gemma |
| | | | 8B | 70B | 7B | 72B | 7B |
| --- | --- | --- | --- | --- | --- | --- | --- |
| GSR (%) | ReAct | ✗ | 3.00 | 17.00 | 9.00 | 20.00 | 2.00 |
| | | ✓ | 8.00 | 23.00 | 13.13 | 23.00 | 8.00 |
| | ReAcTree | ✗ | 13.00 | 53.00 | 16.00 | 58.00 | 10.00 |
| | | ✓ | **27.00** | **59.00** | **28.00** | **61.00** | **36.00** |
| SSR (%) | ReAct | ✗ | 26.82 | 45.83 | 26.60 | 48.02 | 22.20 |
| | | ✓ | 34.22 | 51.25 | 32.98 | 47.58 | 28.48 |
| | ReAcTree | ✗ | 50.32 | **78.38** | 42.73 | 77.08 | 43.52 |
| | | ✓ | **61.53** | 77.08 | **54.80** | **77.28** | **59.83** |

We compare ReAct and ReAcTree on WaH using various recent LLMs. Table 6 presents results for additional LLMs not included in Table 1. Across both large-scale and lighter models, ReAcTree consistently outperforms ReAct, with or without working memory. This indicates that the ReAcTree structure consistently enhances task planning performance by adaptively breaking down complex tasks into solvable agent-level tasks with control flow, allowing the same LLM to handle tasks more effectively and easily. We used Llama 3, Qwen 2, and Gemma 2 as off-the-shelf LLMs. The strong performance of the proposed ReAcTree without working memory further emphasizes its robustness, making it an effective approach for both lightweight and large-scale language model.

## D.1  WITH WORKING MEMORY RESULTS

For the lightweight models, in the case of the LLaMA 3 8B model, ReAcTree shows significant improvements over ReAct, with a 19% increase in GSR and a 27.31% increase in SSR. Similarly, in the Qwen 2 7B model, ReAcTree achieves a 14.87% improvement in GSR and a 21.82% improvement in SSR compared to ReAct. In the Gemma 2 7B model, ReAcTree demonstrates the most impressive gains in lightweight model, with a 28% increase in GSR and a 31.35% increase in SSR.

For large-scale models such as LLaMA 3 70B and Qwen2 72B, ReAcTree also significantly outperforms ReAct. In the LLaMA 3 70B model, ReAcTree outperforms ReAct by 36% in GSR and by 25.83% in SSR. In the Qwen 2 72B model, ReAcTree shows a 38% improvement in GSR and a 29.06% improvement in SSR over ReAct.

## D.2  WITHOUT WORKING MEMORY RESULTS

Even without working memory, ReAcTree surpasses ReAct by a significant margin. In lightweight models, ReAcTree shows notable improvements over ReAct, with a 10% increase in GSR and a 23.5% increase in SSR. Similar trends are observed in the Qwen 2 7B and Gemma 2 7B models, where ReAcTree consistently outperforms ReAct in both GSR and SSR.

For large-scale models, ReAcTree continues to outperform ReAct. In the LLaMA 3 70B model, ReAcTree achieves a 36% improvement in GSR and a 32.55% increase in SSR over ReAct. Similarly, in the Qwen 2 72B model, ReAcTree surpasses ReAct by 38% in GSR and 29.06% in SSR, even without working memory.

# E    FAILURE ANALYSIS

## E.1    FAILURE TYPES

In this section, we categorize the failure cases of the proposed ReAcTree on the WAH-NL results using QWEN-2.5 72B model. The failure cases are classified into four types: *Expand-level*, *Agent-level*, *Constraints*, and *Instruction errors*. Table 7 summarizes these failure types.

First, *Expand-level* failures occur during the expand procedure of ReAcTree. These failures are categorized *Task decomposition*, *Control-flow selection*, and *Expand-level infinite loop* failures. Second, *Agent-level* failures arises during the planning process within individual agent nodes. These include *Navigation & search*, *Agent-level planning*, *Agent-level infinite loop*, and *Termination reasoning* failures. Third *Constraints* failures happen when tasks could be completed with more relaxed limits but fail due to the current thresholds. This category includes *Max step* failures, where the maximum number of decision steps is exceeded. Finally, *Instruction error* stems from issues in task instructions of WAH-NL test set. These are divided into *Incorrect instructions*, where the instruction differs from the goal, and *Ambiguous instructions*, where the instructions are unclear.

Table 7: Failure types observed in the WAH-NL results of the proposed ReAcTree using the QWEN-2.5 72B model.

| Hierarchy | Failiure Type | Details |
|---|---|---|
| **Expand-level** | Task decomposition | Failure to break down instructions into agent-level sub-goals |
| | Control-flow selection | Incorrect selection of control flow within expand-level decomposition |
| | Expand-level infinite loop | Infinite loop in expand-level decomposition |
| **Agent-level** | Navigation & search | Failure in object search & navigation under partial observability |
| | Agent-level planning | Failure in sub-goal planning |
| | Agent-level infinite loop | Infinite loop in agent-level decomposition |
| | Termination reasoning | Termination condition check failure in reasoning |
| **Constraints** | Max step | Exceeded the maximum allowable number of steps |
| **Instruction error** | Incorrect Instruction | When the goal stated in the instructions differs from the actual goal |
| | Ambiguous Instruction | When the goal state cannot be determined solely from the instructions |

## E.2    FAILURE ANALYSIS

In this section, we analyze the failure cases of ReAcTree using the QWEN-2.5 72B model, both with and without working memory. ReAcTree without working memory exhibited a total of 41 failure cases, categorized into *Expand-level* (9 cases), *Agent-level* (30 cases), *Constraints (1 case)*, and *Instruction error* failures (9 cases). In contrast, ReAcTree with WM showed fewer failures, with a total of 37 cases distributed as *Expand-level* (6 cases), *Agent-level* (22 cases), *Constraints* (0 cases), and *Instruction error* failures (9 cases). In both settings, *Agent-level* failures were the most prevalent.

Table 8: Comparison of failure cases with and without the proposed working memory. We analyze the failure cases from a test set of 100 samples on the WAH-NL benchmark. A failure is defined as any instance where at least one sub-goal remains unsatisfied. The table also report the average sub-goal success rate (SSR). In the table, "WM" denotes working memory, and "diff." represents the difference in SSR between cases with and without working memory.

| Failure Type | Number of failures | | Avg. SSR | | |
|---|---|---|---|---|---|
| | w.o. WM | w. WM | w.o. WM | w. WM | diff. |
| Task decomposition | 6 | 5 (-1) | 58.33 | 66.67 | **+8.34** |
| Control-flow selection | 2 | 1 (-1) | 16.67 | 66.67 | **+50.00** |
| Expand-level infinite loop | 1 | 0 (-1) | 0 | 100 | **+100** |
| Navigation & search | 4 | 5 (+1) | 72.08 | 70.42 | -1.66 |
| Agent-level planning | 14 | 9 (-5) | 57.70 | 73.73 | **+16.03** |
| Agent-level infinite loop | 3 | 4 (+1) | 39.72 | 41.67 | +1.95 |
| Termination reasoning | 9 | 4 (-5) | 29.24 | 57.73 | **+28.49** |
| Max step | 1 | 0 (-1) | 60 | 100 | **+40** |
| Incorrect Instruction | 4 | 4 | 37.50 | 37.50 | 0 |
| Ambiguous Instruction | 5 | 5 | 40.56 | 39.17 | -1.39 |

To further understand the impact of working memory, we measured the average subgoal success rate (SSR) for each failure type across tasks that failed in either configuration. Table 8 summarizes these results, reporting the number of failures and the corresponding average SSR for each failure type. For example, in the case of *Task decomposition* failures, the total number of unique tasks used to compute SSR was 9. These tasks included task IDs 3, 10, 12, 23, 37, 38, and 62 from the without working memory configuration, and task IDs 12, 23, 37, 38, 47, and 77 from the with working memory setting. The union of these task sets is $\{3, 10, 12, 23, 37, 38, 47, 62, 77\}$. We calculated the average SSR for both configurations using this union set of failure cases.

The following sections provide a detailed analysis of the major failure types to gain deeper insights into ReAcTree's performance.

**Task decomposition is working well.** In the expanding process of ReAcTree, the agent node selects a control flow and breaks down its goal sentence into multiple subgoals. *Task decomposition* failure occurs when this process generates incorrect subgoals. In our experiments on the WAH-NL test set, ReAcTree encountered these failures in 6 sample cases (6%) without utilizing working memory. However, with the integration of the proposed working memory, this number was reduced to 5 cases (5%), as shown in Table 8. Furthermore, the adoption of working memory significantly improved the average SSR (Avg. SSR in Table 8), with an increase of 8.34% point, from 58.33% to 66.67%. This demonstrates that the proposed working memory enhances subgoal success rates, ensuring partial success even when expand-level task decomposition is not entirely successful.

**Working memory enhance Agent-level planning performance.** The proposed working memory significantly reduce the number of *Agent-level planning* failures. As demonstrated in Table 8, incorporating the proposed working memory reduces the number of Agent-level planning failure cases from 14 to 9, while improving the average SSR by 16.03%. By recalling the locations from previous observations, the context length is shortened, and the number of nodes is reduced, enabling the LLM to generate more accurate plans. This improvement effectively minimizes Agent-level planning failures.

**Working memory improves step efficiency in long-horizon task.** The proposed working memory reduces the number of search steps required in partially observable environments. In max step failure cases, ReAcTree achieves an average SSR of 60.00% without working memory. However, with the adoption of working memory, these cases are resolved without reaching the maximum step limit.

**Limitations.** While the proposed ReAcTree and its memory mechanisms enhance embodied task planning capabilities, certain limitations persist. First, ReAcTree faces challenges in searching for target objects in partially observable environments. Unlike previous work Choi et al. (2024), which assumes a fully observable environment, ReAcTree operates in settings with partial observability, adding significant complexity. As a result, 5% of the evaluation samples fail due to navigation and search failures, with an average SSR of 70.42%.

Second, there is a risk of error propagation. While expand-level decomposition failures occur in only 5% of the evaluation samples, with an average SSR of 66.67%, such errors in expand-level planning can propagate and potentially disrupt the entire process. This limitation could be mitigated in future work by introducing an additional verification step for expand-level planning

Third, unpredictable errors, such as infinite loops in agent-level planning and termination reasoning failures, remain significant challenges. In our experiments, infinite loops in agent-level planning accounted for 4% of the evaluation set, with an average SSR of 41.67%, while termination reasoning failures also made up 4%, with an average SSR of 57.73%. Although this work focuses on designing an efficient architecture for agent nodes and implementing effective memory mechanisms, further improvements to the agent node itself are essential. To address these issues, a robust verification process is needed, where each node evaluates the current progress of the task. While this evaluation is weakly performed in the "Think stage" of ReAct, additional verification mechanisms could significantly mitigate these problems.

### E.3 DETAILED DESCRIPTIONS OF THE FAILURE TYPES

#### E.3.1 TASK DECOMPOSITION FAILURE

Listing 1: The example of the task decomposition failure.

```
### Agent Node 1
Your task is to: Home service robot, input 1 cutlery fork, 1 wine glass and 1
    plate into the dishwasher. When complete then turn the dishwasher on.

...

Think: To solve the task, I need to accomplish 4 subgoals independently. Find and
    open the dishwasher, move the cutlery fork in the dishwasher, move the wine
    glass in the dishwasher, and close and turn on the dishwasher.

Expand: {'control_flow': 'parallel', 'conditions': 'find and open the dishwasher,
    move the cutlery fork in the dishwasher, move the wine glass in the dishwasher
    , close and turn on the dishwasher'}
```

Task decomposition failure occurs when the system initially generates an incorrect sequence of sub-goal instructions to address the given instruction. For instance, in Listing 1, the agent fails to decompose one of the sub-goals during the expand-level task decomposition process.

#### E.3.2 CONTROL-FLOW SELECTION FAILURE

Listing 2: The example of the control-flow selection failure.

```
### Agent Node 1
Your task is to: Please serve a water glass on the kitchen table

...

Think: To solve the task, I need to accomplish 2 subgoals sequentially. Find and
    pick up the water glass, and find the kitchen table and put down the water
    glass.

Expand: {'control_flow': 'sequence', 'conditions': 'find and pick up the water
    glass, find the kitchen table and put down the water glass'}

### Agent Node 2
Your primary goal is to: Please serve a water glass on the kitchen table
```

```
To achieve this, you should perform your sibling tasks in sequence. At this level,
    your sibling tasks are: find and pick up the water glass, and find the
    kitchen table and put down the water glass.

Your task is to: find and pick up the water glass
You are in the house, and there are 4 rooms: bathroom (1), bedroom (1), kitchen
    (1), living room (1). You are in the middle of a bathroom (1). Looking quickly
    around the room, you see ...
Think: To solve the task, I need to recall the location of the water glass.
Act: recall location of water glass
You have not seen water glass before.
Think: Now, I need to find and pick up the water glass in one of the rooms. The
    water glass is more likely to appear in kitchen 1, living room 1, or bedroom
    1. I can check each room one by one until I find and pick up the water glass.
Expand: {'control_flow': 'fallback', 'conditions': 'find and pick up the water
    glass in kitchen 1, find and pick up the water glass in living room 1, find
    and pick up the water glass in bedroom 1'}

### Agent Node 3
Your primary goal is to: find and pick up the water glass
To achieve this, you should perform your sibling tasks using a fallback strategy.
    At this level, your sibling tasks are: find and pick up the water glass in
    kitchen 1, find and pick up the water glass in living room 1, and find and
    pick up the water glass in bedroom 1.
Your task is to: find and pick up the water glass in kitchen 1
You are in the house, and there are 4 rooms: bathroom (1), bedroom (1), kitchen
    (1), living room (1). You are in the middle of a bathroom (1). Looking quickly
    around the room, you see ...
Think: I am in bathroom 1. Now, I need to find the water glass. The water glass is
    more likely to appear on bathroom counter (1), or in bathroom cabinet (1). I
    can check each place one by one, starting from bathroom counter 1, until I
    find the water glass.
Act: go to bathroom counter 1
...
Think: I cannot find the water glass in bathroom 1.
Act: failure

### Agent Node 4
Your primary goal is to: find and pick up the water glass
To achieve this, you should perform your sibling tasks using a fallback strategy.
    At this level, your sibling tasks are: find and pick up the water glass in
    kitchen 1, find and pick up the water glass in living room 1, and find and
    pick up the water glass in bedroom 1.
Your task is to: find and pick up the water glass in living room 1
You are in the house, and there are 4 rooms: bathroom (1), bedroom (1), kitchen
    (1), living room (1). You are in the middle of a bathroom (1). Looking quickly
    around the room, you see ...
Think: I am in bathroom 1. Now, I need to go to living room 1.
Act: go to living room 1
...
Think: I cannot find the water glass in living room 1.
Act: failure

### Agent Node 5
Your primary goal is to: find and pick up the water glass
To achieve this, you should perform your sibling tasks using a fallback strategy.
    At this level, your sibling tasks are: find and pick up the water glass in
    kitchen 1, find and pick up the water glass in living room 1, and find and
    pick up the water glass in bedroom 1.
Your task is to: find and pick up the water glass in bedroom 1
You are in the house, and there are 4 rooms: bathroom (1), bedroom (1), kitchen
    (1), living room (1). You are in the middle of a living room (1). Looking
    quickly around the room, you see ...
Think: I am in living room 1. Now, I need to go to bedroom 1.
Act: go to bedroom 1
```

```
Think: I cannot find the water glass in bedroom 1.
Act: failure

[The parent sequence node terminates because the fallback node returns a failure.]
```

Control-flow selection failure arises from selecting an inappropriate control flow, often involving cases where a "fallback" node is used for object searching. If the parent node of the "fallback" node is not a "parallel" node but a "sequence" node, and all child nodes of the "fallback" fail, the "sequence" node does not fully execute and terminates prematurely. An example is summarized in Listing 2.

### E.3.3    EXPAND-LEVEL INFINITE LOOP

Listing 3: The example of the expand-level infinite loop failure

```
Your task is to: move the wine in the fridge
...
Expand: {'control_flow': 'sequence', 'conditions': 'find and pick up the wine,
    find the fridge and put down the wine'}

### find and pick up the wine

Your task is to: find and pick up the wine
...
Expand: {'control_flow': 'fallback', 'conditions': 'find and pick up the wine in
    kitchen 1, find and pick up the wine in living room 1, find and pick up the
    wine in bedroom 1'}

Your task is to: find and pick up the wine in kitchen 1
...
Act: failure

Your task is to: find and pick up the wine in living room 1
...
Act: failure

Your task is to: find and pick up the wine in bedroom 1
...
Act: done

### find the fridge and put down the wine

[The agent start the infinite loop]
Your task is to: find the fridge and put down the wine
...
Expand: {'control_flow': 'sequence', 'conditions': 'find and pick up the wine,
    find the fridge and put down the wine'}

### find and pick up the wine

Your task is to: find and pick up the wine
...
Expand: {'control_flow': 'fallback', 'conditions': 'find and pick up the wine in
    kitchen 1, find and pick up the wine in living room 1, find and pick up the
    wine in bedroom 1'}

Your task is to: find and pick up the wine in kitchen 1
...
Act: failure

Your task is to: find and pick up the wine in living room 1
```

```
...
Act: failure

Your task is to: find and pick up the wine in bedroom 1
...
Act: done

### find the fridge and put down the wine

Your task is to: find the fridge and put down the wine
Expand: {'control_flow': 'sequence', 'conditions': 'find and pick up the wine,
    find the fridge and put down the wine'}
[The agent try to find the wine again already found]

### find and pick up the wine

Your task is to: find and pick up the wine
Expand: {'control_flow': 'fallback', 'conditions': 'find and pick up the wine in
    kitchen 1, find and pick up the wine in living room 1, find and pick up the
    wine in bedroom 1'}
...
```

Lastly, an expand-level infinite loop occurs when the expand-level decomposition succeeds, but a repetitive sub-goal sequence is endlessly generated, causing the tree structure to expand infinitely. While this issue arises in only one sample with ReAcTree without working memory and is completely absent when working memory is employed, it represents a critical failure type. In such cases, the system is unable to execute actions correctly and continues to expand indefinitely, making the goal impossible to achieve.

### E.3.4 NAVIGATION & SEARCH FAILURE

Listing 4: The example of the navigation & search failure.

```
...
Your primary goal is to: move the glass of water in the dishwasher
To achieve this, you should perform your sibling tasks in sequence. At this level,
     your sibling tasks are: find and pick up the glass of water, and find the
    dishwasher and put down the glass of water.
Your task is to: find and pick up the glass of water
You are in the house, and there are 4 rooms: bathroom (1), bedroom (1), kitchen
    (1), living room (1). You are in the middle of a kitchen (1). Looking quickly
    around the room, you see chair (1, 2, 3, 4), dishwasher (1), faucet (1),
    fridge (1), kitchen cabinet (1, 2, 3, 4, 5, 6, 7, 8), kitchen counter (1, 2),
    kitchen table (1), microwave oven (1), nightstand (1, 2), sink (1), sofa (1),
    stove (1), toaster (1).
Think: To solve the task, I need to recall the location of the glass of water.
Act: recall location of glasses
You have not seen glasses before.
Think: Now, I need to find and pick up the glass of water in one of the rooms. The
     glass of water is more likely to appear in kitchen 1, living room 1, or
    bedroom 1. I can check each room one by one until I find and pick up the glass
     of water.
Expand: {'control_flow': 'fallback', 'conditions': 'find and pick up the glass of
    water in kitchen 1, find and pick up the glass of water in living room 1, find
     and pick up the glass of water in bedroom 1'}

[Starting search for the water glass.]
Your primary goal is to: find and pick up the glass of water
To achieve this, you should perform your sibling tasks using a fallback strategy.
    At this level, your sibling tasks are: find and pick up the glass of water in
    kitchen 1, find and pick up the glass of water in living room 1, and find and
    pick up the glass of water in bedroom 1.
```

```
Your task is to: find and pick up the glass of water in kitchen 1
...
Think: I cannot find the glass of water in kitchen 1.
Act: failure

Your primary goal is to: find and pick up the glass of water
To achieve this, you should perform your sibling tasks using a fallback strategy.
    At this level, your sibling tasks are: find and pick up the glass of water in
    kitchen 1, find and pick up the glass of water in living room 1, and find and
    pick up the glass of water in bedroom 1.
Your task is to: find and pick up the glass of water in living room 1
...
Think: I cannot find the glass of water in living room 1.
Act: failure

Your primary goal is to: find and pick up the glass of water
To achieve this, you should perform your sibling tasks using a fallback strategy.
    At this level, your sibling tasks are: find and pick up the glass of water in
    kitchen 1, find and pick up the glass of water in living room 1, and find and
    pick up the glass of water in bedroom 1.
Your task is to: find and pick up the glass of water in bedroom 1
...
Think: I cannot find the glass of water in bedroom 1.
Act: failure

[Unable to find the water glass, proceeding to the next parallel node]
Your primary goal is to: Please put a cutlery fork and a glass of water in the
    dishwasher, then turn the diswasher on.
To achieve this, you should perform your sibling tasks in parallel. At this level,
     your sibling tasks are: find and open the dishwasher, move the cutlery fork
    in the dishwasher, move the glass of water in the dishwasher, and close and
    turn on the dishwasher.
Your task is to: close and turn on the dishwasher
...
```

Navigation & search failure occurs when the agent is unable to find the target object despite executing the entire generated search action sequence(commonly associated with the "fallback" node in ReAcTree). Object navigation is particularly challenging due to partial observability, which can prevent the task from being completed even if the expand-level plan is successfully generated.

### E.3.5 AGENT-LEVEL PLANNING FAILURE

Listing 5: The example of the agent-level planning failure.

```
Your primary goal is to: find and pick up the cake
To achieve this, you should perform your sibling tasks using a fallback strategy.
    At this level, your sibling tasks are: find and pick up the cake in kitchen 1,
     find and pick up the cake in living room 1, and find and pick up the cake in
    bedroom 1.
Your task is to: find and pick up the cake in bedroom 1
...
Think: I am in living room 1. Now, I need to go to bedroom 1.
Act: go to bedroom 1
You move to the bedroom (1). Looking quickly around the room, you see ..
Think: Now, I need to find the cake. The cake is more likely to appear on coffee
    table (2), desk (2, 3), or in cabinet (1). I can check each place one by one,
    starting from coffee table 2, until I find the cake.
...
You open cabinet. You see apple (2), cabinet (1), computer (2), desk (2), light
    switch (4), pound cake (1), sundae (1)
Think: I find pound cake 1. Now, I need to pick it up.
Act: pick up pound cake 1
```

```
You pick up pound cake. You hold pound cake (1).
Think: Now I accomplish the goal.
Act: done
```

Agent-level planning failure occurs when the agent generates an incorrect or meaningless action sequence within action-performing nodes. This can happen when the action sequence is misaligned with the provided sub-goal instructions or when the actions executed do not align with the previous reasoning outcomes. For example, in Listing 5, the sub-goal is *"Find and pick up the cake in bedroom 1."* The agent reasons correctly, stating: *"Think: Now I need to find the cake. The cake is more likely to be on coffee table (2), desk (2, 3), or in cabinet (1). I can check each place one by one, starting from coffee table 2, until I find the cake."* However, instead of picking up the cake, the agent mistakenly picks up a pound cake.

### E.3.6 AGENT-LEVEL INFINITE LOOP

Listing 6: The example of the agent-level infinite loop.

```
Your task is to: go find a bottle of win and a pudding there on the coffee table.

[Starting to search a wine and pudding]
Think: I need to find a bottle of wine and a pudding. The wine is more likely to
    appear in kitchen 1, and the pudding is more likely to appear in living room
    1.
...
Think: I can see fridge 1. I need to open fridge 1 to find the wine.
Act: go to fridge 1
You arrive at the fridge (1). The fridge (1) is closed. You see fridge (1), mug
    (3)
Think: I need to open fridge 1.
Act: open fridge 1
You open fridge. You see apple (1), fridge (1), mug (3), sofa (1)

[Picked up an apple instead of wine]
Think: I can see apple 1. I need to pick up apple 1.
Act: pick up apple 1
You pick up apple. You hold apple (1).

[Search the pudding]
Think: Now, I need to find the pudding. The pudding is more likely to appear in
    living room 1.
...
Think: I can see coffee table 1. I need to go to coffee table 1 to find the
    pudding.
Act: go to coffee table 1
You arrive at the coffee table (1). You see apple (1), coffee table (1), sofa (2),
     tv (1) You hold apple (1).

[Put down apple]
Think: I can see apple 1. I need to put down apple 1.
Act: put down apple 1
You put down apple on coffee table
Think: Now, I need to find the pudding. The pudding is more likely to appear in
    living room 1.
Act: go to living room 1
You move to the living room (1). Looking quickly around the room, you see ...
Think: I can see coffee table 1. I need to go to coffee table 1 to find the
    pudding.
Act: go to coffee table 1
You arrive at the coffee table (1). You see apple (1), coffee table (1), sofa (2),
     tv (1)

[Pick up apple again]
```

```
Think: I can see apple 1. I need to pick up apple 1.
Act: pick up apple 1
You pick up apple. You hold apple (1).
Think: Now, I need to find the pudding. The pudding is more likely to appear in
    living room 1.
Act: go to living room 1
You move to the living room (1). Looking quickly around the room, you see
    bookshelf (2, 3), chair (6), closet (1), coffee table (1), computer (1), desk
    (2), nightstand (4), sofa (2, 3), tv (1). You hold apple (1).
Think: I can see coffee table 1. I need to go to coffee table 1 to find the
    pudding.
Act: go to coffee table 1
You arrive at the coffee table (1). You see apple (1), coffee table (1), sofa (2),
     tv (1) You hold apple (1).

[Put down apple]
Think: I can see apple 1. I need to put down apple 1.
Act: put down apple 1
You put down apple on coffee table

[Search the pudding]
Think: Now, I need to find the pudding. The pudding is more likely to appear in
    living room 1.
Act: go to living room 1
You move to the living room (1). Looking quickly around the room, you see
    bookshelf (2, 3), chair (6), closet (1), coffee table (1), computer (1), desk
    (2), nightstand (4), sofa (2, 3), tv (1).
Think: I can see coffee table 1. I need to go to coffee table 1 to find the
    pudding.
Act: go to coffee table 1
You arrive at the coffee table (1). You see apple (1), coffee table (1), sofa (2),
     tv (1)
Think: I can see apple 1. I need to pick up apple 1.
...
```

An agent-level infinite loop occurs when the agent gets trapped in a continuous cycle of actions during its execution and cannot break free. Unlike the previous planning failure, where the agent could transition to the next node, this failure is more severe as it causes the agent's performance to stall at a specific node. For example, as shown in Listing 6, the agent repeatedly picks up and puts down an apple in a loop, an action unrelated to achieving the task's sub-goal.

### E.3.7 TERMINATION REASONING FAILURE

Listing 7: The example of the termination reasoning failure.

```
Your primary goal is to: find and pick up the juice
To achieve this, you should perform your sibling tasks using a fallback strategy.
    At this level, your sibling tasks are: find and pick up the juice in kitchen
    1, find and pick up the juice in living room 1, and find and pick up the juice
     in bedroom 1.

Your task is to: find and pick up the juice in kitchen 1
You are in the house, and there are 4 rooms: bathroom (1), bedroom (1), kitchen
    (1), living room (1). You are in the middle of a kitchen (1). Looking quickly
    around the room, you see chair (1, 2, 3, 4), dishwasher (1), faucet (1),
    fridge (1), kitchen cabinet (1, 2, 3, 4, 5, 6, 7, 8), kitchen counter (1, 3),
    kitchen table (1), microwave oven (1), nightstand (1, 2), sink (1), sofa (1),
    stove (1), toaster (1).

Think: I am in kitchen 1. Now, I need to find the juice. The juice is more likely
    to appear on kitchen counter (1, 3), kitchen table (1), or in fridge (1),
```

```
     kitchen cabinet (1, 2, 3, 4, 5, 6, 7, 8). I can check each place one by one,
     starting from kitchen counter 1, until I find the juice.
Act: go to kitchen counter 1
...
Think: I cannot find the juice. Now, I need to go to living room 1.

[The agent fail to terminate the node]
Act: go to living room 1
You move to the living room (1). Looking quickly around the room, you see
     bookshelf (2, 3), chair (6), closet (1), coffee table (1), computer (1), desk
     (2), nightstand (4), sofa (2, 3), tv (1).
Think: I am in living room 1. Now, I need to find the juice. The juice is more
     likely to appear on coffee table (1), desk (2), or in nightstand (4). I can
     check each place one by one, starting from coffee table 1, until I find the
     juice.
Act: go to coffee table 1
...

[The agent fail to terminate the node]
Think: I cannot find the juice. Now, I need to go to kitchen 1.
Act: go to kitchen 1
...

[The agent fail to terminate the node]
Think: I cannot find the juice. Now, I need to go to living room 1.
Act: go to living room 1
...

[The agent fail to terminate the node]
Think: I cannot find the juice. Now, I need to go to bedroom 1.
Act: go to bedroom 1
...

[The agent fail to terminate the node]
Think: I cannot find the juice. Now, I need to go to kitchen 1.
Act: go to kitchen 1
...
Max steps
```

A termination reasoning failure occurs when the agent either fails to recognize that the necessary actions have been completed and cannot decide to terminate the node or prematurely terminates the node before fully executing the planned action sequence. Unlike the agent-level infinite loop, where the agent endlessly repeats actions that are unrelated or only partially related to task completion, a termination reasoning failure arises when the agent fails to identify task completion or failure and repeats the process, even after executing the entire action sequence derived from sub-goal instructions and reasoning processes.

### E.3.8 MAX STEP

Listing 8: The example of the Max Step failure.

```
### Generated ReAcTree
Instruction : Put a wine a juice a pound cake
a pudding and an apple on the kitchen table.

[a1]rootnode (parallel)

[a2]move the wine on the kitchen table (sequence)
  [a3]find and pick up the wine (fallback)
    [a4]find and pick up the wine in kitchen 1 [S]
    find and pick up the wine in living room 1
    find and pick up the wine in bedroom 1

  [a5]find the kitchen table and put down the wine [S]

[a6]move the juice on the kitchen table (sequence)
  [a7]find and pick up the juice (fallback)
    [a8]find and pick up the juice in kitchen 1 [F]
    [a9]find and pick up the juice in living room 1 [S]
    find and pick up the juice in bedroom 1

  [a10]find the kitchen table and put down the juice [S]

[a11]move the pound cake on the kitchen table (sequence)
  [a12]find and pick up the pound cake (sequence)
    [a13]find and pick up the pound cake in kitchen 1 [S]
    find and pick up the pound cake in living room 1
    find and pick up the pound cake in bedroom 1

  [a14]find the kitchen table and put down the pound cake [S]

[a15]move the pudding on the kitchen table (sequence)
  [a16]find and pick up the pudding (fallback)
    [a17]find and pick up the pudding in kitchen 1 [F]
    [a18]find and pick up the pudding in living room 1 [S]
    find and pick up the pudding in bedroom 1

  [a19]find the kitchen table and put down the pudding [S]

[a20]move the apple on the kitchen table (sequence)
  [a21]find and pick up the apple (fallback1)
    [a22]find and pick up the apple in kitchen 1 [MSE]
    find and pick up the apple in living room 1 [MSE]
    find and pick up the apple in bedroom 1 [MSE]

  find the kitchen table and put down the apple [MSE]
[S]  : Success,   [F] : Failure, [MSE] : Max Step Error
[An] : n-th agent

### Agent 22
Your primary goal is to: find and pick up the apple
To achieve this, you should perform your sibling tasks using a fallback strategy.
    At this level, your sibling tasks are: find and pick up the apple in kitchen
    1, find and pick up the apple in living room 1, and find and pick up the apple
     in bedroom 1.
Your task is to: find and pick up the apple in kitchen 1
You are in the house, and there are 4 rooms: bathroom (1), bedroom (1), kitchen
    (1), living room (1). You are in the middle of a kitchen (1). Looking quickly
    around the room, you see bench (1, 2), bookshelf (1, 2), chair (1, 2),
    dishwasher (1), faucet (2), fridge (1, 2), garbage can (1), kitchen cabinet
```

```
        (1, 2, 3, 4, 5, 6, 7, 8), kitchen counter (1, 2), kitchen table (1), microwave
        oven (1), sink (2), stove (1), toaster (1).
Think: I am in kitchen 1. Now, I need to find the apple. The apple is more likely
    to appear on kitchen counter (1, 2), kitchen table (1), or in fridge (1, 2),
    kitchen cabinet (1, 2, 3, 4, 5, 6, 7, 8). I can check each place one by one,
    starting from kitchen counter 1, until I find the apple.
Act: go to kitchen counter 1
You arrive at the kitchen counter (1). You see bowl (1, 2, 3, 4, 5), chair (2),
    chicken (1), cutlets (1), dishwashing liquid (1), faucet (2), kitchen cabinet
    (1, 2, 3, 4, 5, 6, 7, 8), kitchen counter (1), plate (1, 2, 3, 4), sink (2),
    washing sponge (1), wine (2)
Act: go to kitchen counter 2
You arrive at the kitchen counter (2). You see chair (1, 2), kitchen cabinet (3,
    5, 6, 7, 8), kitchen counter (1, 2), light switch (2), plate (1, 2, 3, 4),
    wine (2)
Act: go to kitchen table 1
You arrive at the kitchen table (1). You see bench (2), book (1), bowl (6), coffee
     pot (1), condiment shaker (1, 2), juice (1), kitchen table (1), pound cake
    (1), pudding (2), wine (1)
Act: go to fridge 1
You arrive at the fridge (1). The fridge (1) is open. You see bench (1), fridge
    (1)
Act: go to fridge 2
You arrive at the fridge (2). The fridge (2) is open. You see bench (1), fridge
    (1, 2), pancake (1)
Act: go to kitchen cabinet 1
You arrive at the kitchen cabinet (1). The kitchen cabinet (1) is open. You see
    chicken (1), cooking pot (1), cutlery knife (1, 2), cutlets (1), dishwashing
    liquid (1), faucet (2), frying pan (1), garbage can (1), kitchen cabinet (1,
    4), kitchen counter (1), sink (2), stove (1)
Act: go to kitchen cabinet 2
You arrive at the kitchen cabinet (2). The kitchen cabinet (2) is open. You see
    bowl (1, 2, 3, 4, 5), chair (2), chicken (1), cutlery knife (1, 2), cutlets
    (1), dishwashing liquid (1), faucet (2), kitchen cabinet (1, 2, 3, 4, 5, 6, 7)
    , kitchen counter (1), plate (1, 2, 3, 4), sink (2), washing sponge (1)
Act: go to kitchen cabinet 3
You arrive at the kitchen cabinet (3). The kitchen cabinet (3) is open. You see
    bowl (1, 2, 3, 4, 5), chair (1, 2), dishwashing liquid (1), faucet (2),
    kitchen cabinet (2, 3, 4, 5, 6, 7), kitchen counter (1, 2), plate (1, 2, 3, 4)
    , sink (2), washing sponge (1), wine (2)
Act: go to kitchen cabinet 4
You arrive at the kitchen cabinet (4). The kitchen cabinet (4) is open. You see
    bowl (1, 2, 3, 4, 5), chicken (1), cooking pot (1), cutlery knife (1, 2),
    cutlets (1), dishwashing liquid (1), faucet (2), garbage can (1), kitchen
    cabinet (1, 2, 4), kitchen counter (1), sink (2), washing sponge (1)
Act: go to kitchen cabinet 5
You arrive at the kitchen cabinet (5). The kitchen cabinet (5) is open. You see
    bowl (1, 2, 3, 4, 5), chair (1, 2), cutlets (1), dishwashing liquid (1),
    faucet (2), kitchen cabinet (2, 3, 4, 5, 6, 7, 8), kitchen counter (1, 2),
    plate (1, 2, 3, 4), sink (2), washing sponge (1), wine (2)
Max steps
```

A max step error occurs when the agent successfully generates a valid agent tree and executes the plans correctly but is forced to terminate upon reaching the maximum step limit. This type of failure is likely solvable if the agent is given a higher step limit. An example of this case is presented in Listing 8, with the generated agent tree structure. Here, five sub-goals are correctly managed within the expand-level and agent-level instruction tree and executed properly by the agent until the maximum step limit is reached. The SSR achieved in this case is 60.00% as described in Table 8. However, the proposed working memory significantly enhances step efficiency in such cases, enabling all sub-goals to be resolved within the given step limit.

### E.3.9 INCORRECT INSTRUCTION

Listing 9: The example of the incorrect instruction.

```
### Instruction
Please, put 1 wine glass, 1 water glass, and 1 plate on the
table

### Goal condition
"on_plate_kitchentable": 1,
"on_waterglass_kitchentable": 1,
"on_wineglass_kitchentable": 1,
"on_cutleryfork_kitchentable": 1
```

Incorrect instructions occur when the instructions fail to specify all the required sub-goals. In such cases, no task planning agent can complete the task due to insufficient information. For example, as shown in Listing 9, there are four sub-goals, but the instruction specifies only three, omitting the step of placing a cutlery fork on the kitchen table. Additionally, the instruction is ambiguous, as it does not clarify which type of table is the intended target receptacle. Despite this incorrectness, ReAcTree successfully completes the three specified sub-goals in Listing. 9, but achieving an SSR of 75.00%.

### E.3.10 AMBIGUOUS INSTRUCTION

Listing 10: The example of the ambiguous instruction.

```
[Case 1]
### Instruction
give me an apple, a pudding and serve a glass of wine

### Goal condition
"on_wine_coffeetable": 1,
"on_pudding_coffeetable": 1,
"on_apple_coffeetable": 1

[Case 2]
### Instruction
Always get a glass of water first before anything else,
before food on the table with a cutlery fork and knife

### Goal condition
"on_plate_kitchentable": 1,
"on_waterglass_kitchentable": 1,
"on_cutleryfork_kitchentable": 1
```

The WAH-NL dataset also contains ambiguous instructions. In such cases, the instructions fail to explicitly describe the goal condition, use unclear object names, or refer to objects with high-level conceptual nouns. This ambiguity can confuse the LLM when reasoning about the goal state, leading to the generation of incorrect plans.

## F    EXPERIMENTS DETAILS OF ALFRED DATASET

### F.1    AI2THOR SIMLUATOR AND ALFRED DATASET

AI2THOR simulator supports 9 interaction actions: "pick up," "open," "close," "turn on," "turn off," "slice," and "put down," "drop," and one navigation action "go to." ALFRED dataset consists of 7 task types: *simple pick & place*, *pick & place with movable receptacle*, *cool & place*, *heat & place*, *pick two object & place*, *clean & place*, and *look object in light*. Following previous work (Choi et al., 2024), the *pick two object and place* is excluded. In LoTa-Bench, the AI2THOR simulation used the "find" for navigation action, but since it wasn't a partially observable settings, once "find [object]" was selected as an action, the agent moved directly to the object without considering whether it was inside a receptacle or an unobserved object. However, in this experiment, we improved this limitation by applying a partially observable settings, changing the navigation action name from "find" to "go to" directing the agent to either the object or its receptacle.

### F.2    MEMORY CONSTRUCTION

To construct the episodic memory, we began by randomly selecting three task trajectories for each task type from the ALFRED training set. Then, we collected human-annotated text trajectories with working memory applied for both the ReAct and ReAcTree models. Next, while running the LLaMa 3 70B model on the training set, we embedded the human-annotated text trajectories as in-context examples. Due to the large size of the ALFRED dataset, we sampled 5% of the training set. Only the trajectories from successfully completed tasks were added to the episodic memory, with no more than five per task type. Finally, we combined these human-annotated text trajectories with the successfully completed task trajectories to form the episodic memory for both ReAct and ReAcTree.

### F.3    QUALATATIVE RESULTS

Figures 6 and 7 show the qualitative results comparing a failed trajectory in ReAct and a successful trajectory in ReAcTree for the *heat & place* task type using the LLaMA 3 70B model. The NL instruction for the task is "put the cooked tomato slice on the kitchen island", which requires slicing and moving the object. there are a challenge with the ALFRED dataset, where ambiguous NL instructions complicate task resolution. For example, in this task, the instruction ambiguously states to place the item on the "kitchen island," yet the agent placed it on the countertop since the simulator does not provide a receptacle explicitly labeled "kitchen island." Such ambiguity can hinder accurate task interpretation and execution.

ReAcTree successfully expands and executes the instruction using sequence control flow with eight subtask sqeuence, such as: "find and pick up the knife, find and slice and then pick up the tomato, find and open the microwave, find the microwave and put down the tomato, find and close and turn on the microwave, find and turn off and open microwave, find and pick up tomato in microwave, and find a countertop and put down the tomato on the countertop." In contrast, ReAct failed by initially navigating to SinkBasin (1) and picking up the tomato before the knife, causing the task to fail as the knife should have been picked up and used to slice the tomato first. Meanwhile, ReAcTree correctly searched through CounterTop (1) and CounterTop (2) to locate the knife, leading to the successful completion of the task. Please refer to Appendix H for detailed text trajectories.

## G    EXAMPLE OF PLANNING RESULTS

### G.1    WAH-NL EXAMPLE CASES

**Put the apple, pancake, cupcake, and juice on the kitchen table.**

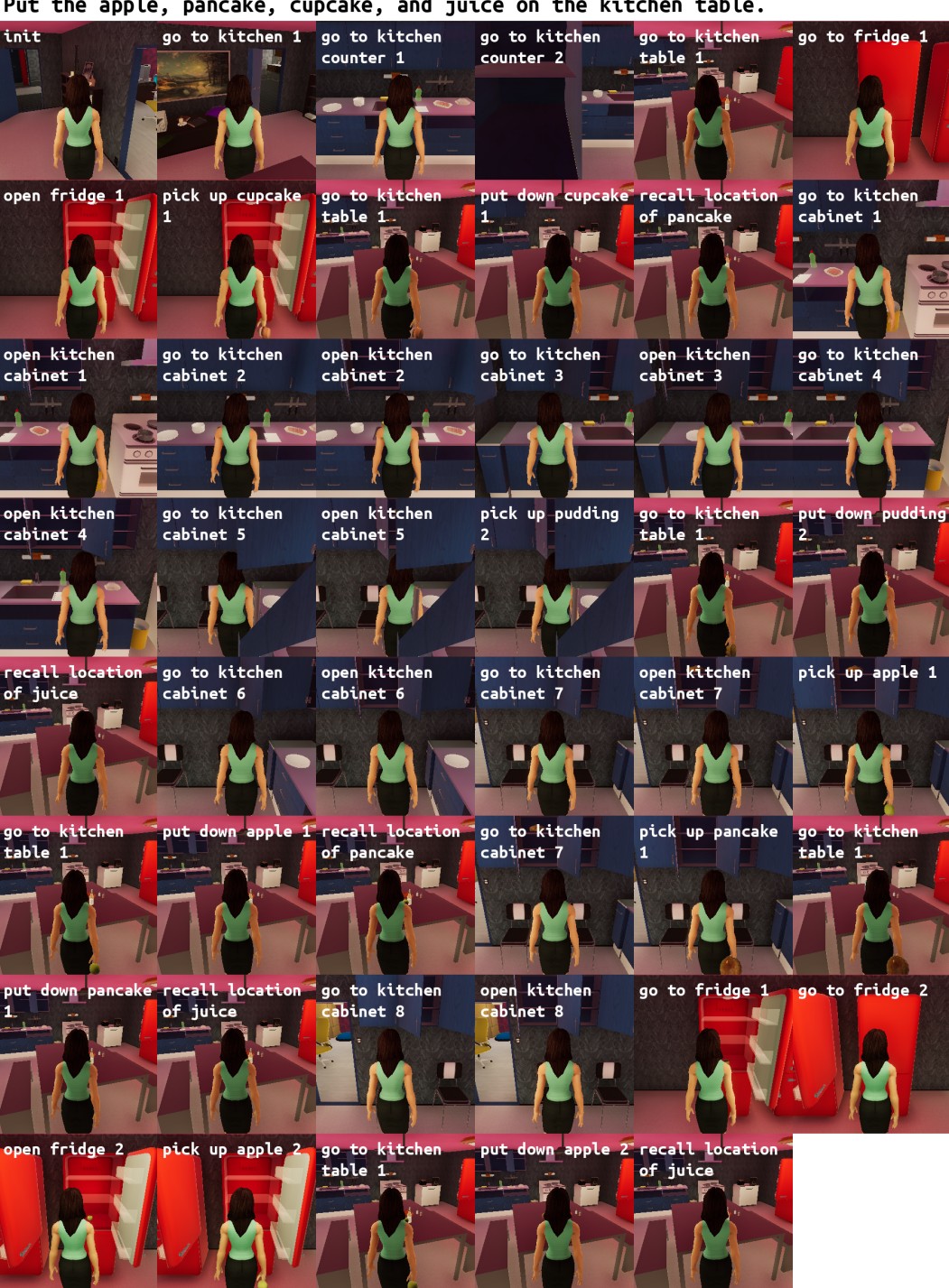

Figure 5: Failure case of ReAct on the WAH-NL dataset using LLaMA 3 70B.

## G.2 ALFRED EXAMPLE CASES

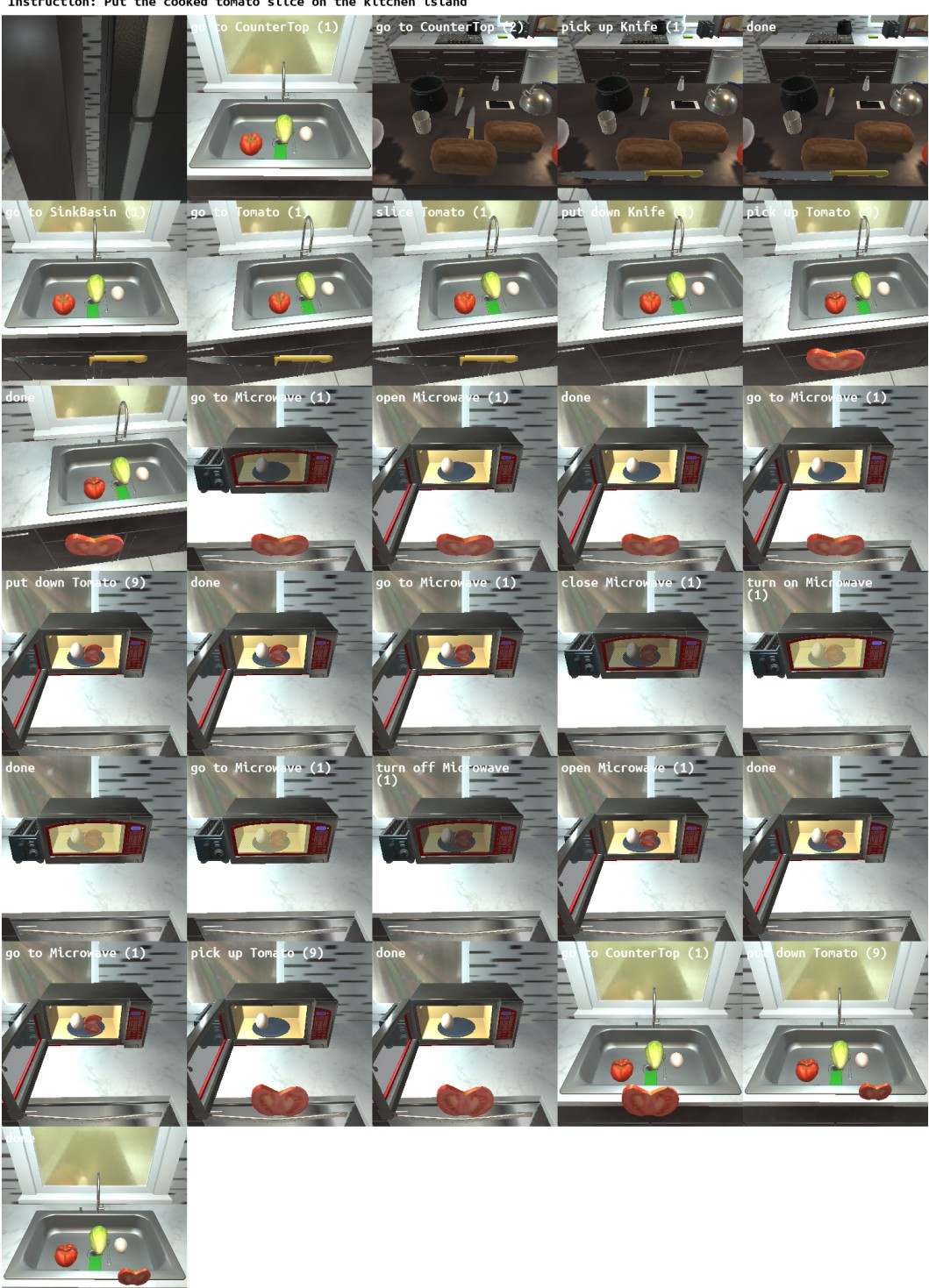

Figure 6: Success case of ReAcTree on the ALFRED dataset using LLaMA 3 70B.

Instruction: Put the cooked tomato slice on the kitchen island

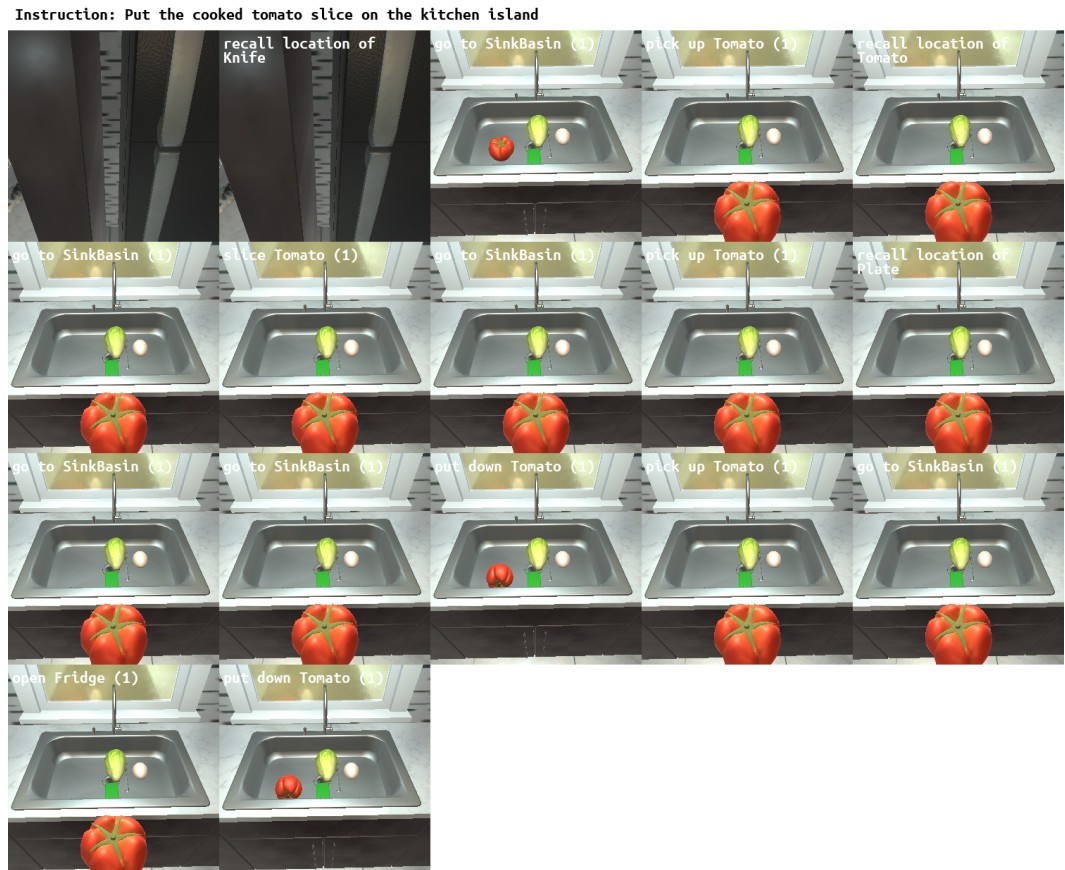

Figure 7: Failure case of ReAct on the ALFRED dataset using LLaMA 3 70B.

# H  EXAMPLE OF TEXT TRAJECTORIES

## H.1  WAH-NL EXAMPLE CASES

Listing 11: Success trajectory of ReAcTree on the WAH-NL dataset using LLaMA 3 70B.

```
### Agent Node 1
Your task is to: Put the apple, pancake, cupcake, and juice on the kitchen table.
You are in the house, and there are 4 rooms: bathroom (1), bedroom (1), kitchen
    (1), living room (1). You are in the middle of a kitchen (1). Looking quickly
    around the room, you see bench (1, 2), bookshelf (1, 2), chair (1, 2),
    dishwasher (1), faucet (2), fridge (1, 2), garbage can (1), kitchen cabinet
    (1, 2, 3, 4, 5, 6, 7, 8), kitchen counter (1, 2), kitchen table (1), microwave
     oven (1), sink (2), stove (1), toaster (1).
Think: To solve the task, I need to accomplish 4 subgoals independently. Move the
    apple on the kitchen table, move the pancake on the kitchen table, move the
    cupcake on the kitchen table, and move the juice on the kitchen table.
Expand:
 - control flow: parallel
 - subgoals: move the apple on the kitchen table, move the pancake on the kitchen
    table, move the cupcake on the kitchen table, move the juice on the kitchen
    table

### Agent Node 2
Your primary goal is to: Put the apple, pancake, cupcake, and juice on the kitchen
    table.
```

To achieve this, you should perform your sibling tasks in parallel. At this level,
    your sibling tasks are: move the apple on the kitchen table, move the pancake
    on the kitchen table, move the cupcake on the kitchen table, and move the
    juice on the kitchen table.
Your task is to: move the apple on the kitchen table
You are in the house, and there are 4 rooms: bathroom (1), bedroom (1), kitchen
    (1), living room (1). You are in the middle of a kitchen (1). Looking quickly
    around the room, you see bench (1, 2), bookshelf (1, 2), chair (1, 2),
    dishwasher (1), faucet (2), fridge (1, 2), garbage can (1), kitchen cabinet
    (1, 2, 3, 4, 5, 6, 7, 8), kitchen counter (1, 2), kitchen table (1), microwave
    oven (1), sink (2), stove (1), toaster (1).
Think: To solve the task, I need to accomplish 2 subgoals sequentially. Find and
    pick up the apple, and find the kitchen table and put down the apple.
Expand
 - control flow: sequence
 - subgoals: find and pick up the apple, find the kitchen table and put down the
    apple

### Agent Node 3
Your primary goal is to: move the apple on the kitchen table
To achieve this, you should perform your sibling tasks in sequence. At this level,
    your sibling tasks are: find and pick up the apple, and find the kitchen
    table and put down the apple.
Your task is to: find and pick up the apple
You are in the house, and there are 4 rooms: bathroom (1), bedroom (1), kitchen
    (1), living room (1). You are in the middle of a kitchen (1). Looking quickly
    around the room, you see bench (1, 2), bookshelf (1, 2), chair (1, 2),
    dishwasher (1), faucet (2), fridge (1, 2), garbage can (1), kitchen cabinet
    (1, 2, 3, 4, 5, 6, 7, 8), kitchen counter (1, 2), kitchen table (1), microwave
    oven (1), sink (2), stove (1), toaster (1).
Think: To solve the task, I need to recall the location of the apple.
Act: recall location of apple
You have not seen apple before.
Think: Now, I need to find and pick up the apple in one of the rooms. The apple is
    more likely to appear in kitchen 1, living room 1, or bedroom 1. I can check
    each room one by one until I find and pick up the apple.
Expand:
 - control flow: fallback
 - subgoals: find and pick up the apple in kitchen 1, find and pick up the apple
    in living room 1, find and pick up the apple in bedroom 1

### Agent Node 4
Your primary goal is to: find and pick up the apple
To achieve this, you should perform your sibling tasks using a fallback strategy.
    At this level, your sibling tasks are: find and pick up the apple in kitchen
    1, find and pick up the apple in living room 1, and find and pick up the apple
    in bedroom 1.
Your task is to: find and pick up the apple in kitchen 1
You are in the house, and there are 4 rooms: bathroom (1), bedroom (1), kitchen
    (1), living room (1). You are in the middle of a kitchen (1). Looking quickly
    around the room, you see bench (1, 2), bookshelf (1, 2), chair (1, 2),
    dishwasher (1), faucet (2), fridge (1, 2), garbage can (1), kitchen cabinet
    (1, 2, 3, 4, 5, 6, 7, 8), kitchen counter (1, 2), kitchen table (1), microwave
    oven (1), sink (2), stove (1), toaster (1).
Think: I am in kitchen 1. Now, I need to find the apple. The apple is more likely
    to appear on kitchen counter (1, 2), kitchen table (1), or in fridge (1, 2),
    kitchen cabinet (1, 2, 3, 4, 5, 6, 7, 8). I can check each place one by one,
    starting from kitchen counter 1, until I find the apple.
Act: go to kitchen counter 1
You arrive at the kitchen counter (1). You see bowl (1, 2, 3, 4, 5), chair (2),
    chicken (1), cutlets (1), dishwashing liquid (1), faucet (2), kitchen cabinet
    (1, 2, 3, 4, 5, 6, 7), kitchen counter (1), plate (1, 2, 3, 4), sink (2),
    washing sponge (1)

```
Act: go to kitchen counter 2
You arrive at the kitchen counter (2). You see chair (1, 2), kitchen cabinet (5,
    6, 7, 8), kitchen counter (1, 2), light switch (2), plate (1, 2, 3, 4)
Act: go to kitchen table 1
You arrive at the kitchen table (1). You see bench (2), book (1), bowl (6),
    condiment shaker (1, 2), kitchen table (1)
Act: go to fridge 1
You arrive at the fridge (1). The fridge (1) is closed. You see bench (1), fridge
    (1)
Act: open fridge 1
You open fridge. You see bench (1), cupcake (1), fridge (1), pudding (3)
Act: go to fridge 2
You arrive at the fridge (2). The fridge (2) is closed. You see bench (1), cupcake
     (1), fridge (1, 2), pudding (3)
Act: open fridge 2
You open fridge. You see apple (2), bench (1), cupcake (1), fridge (1, 2), pudding
     (3)
Think: I find apple 2. Now, I need to pick it up.
Act: pick up apple 2
You pick up apple. You hold apple (2).
Think: Now I accomplish the goal.
Act: done

### Agent Node 5
Your primary goal is to: move the apple on the kitchen table
To achieve this, you should perform your sibling tasks in sequence. At this level,
     your sibling tasks are: find and pick up the apple, and find the kitchen
    table and put down the apple.
Your task is to: find the kitchen table and put down the apple
You are in the house, and there are 4 rooms: bathroom (1), bedroom (1), kitchen
    (1), living room (1). You are in the middle of a kitchen (1). Looking quickly
    around the room, you see bench (1, 2), bookshelf (1, 2), chair (1, 2),
    dishwasher (1), faucet (2), fridge (1, 2), garbage can (1), kitchen cabinet
    (1, 2, 3, 4, 5, 6, 7, 8), kitchen counter (1, 2), kitchen table (1), microwave
    oven (1), sink (2), stove (1), toaster (1). You hold apple (2).
Think: I hold apple 2. Now, I need to find the kitchen table. I can see kitchen
    table 1.
Act: go to kitchen table 1
You arrive at the kitchen table (1). You see apple (2), bench (2), book (1), bowl
    (6), condiment shaker (1, 2), kitchen table (1) You hold apple (2).
Think: Now, I need to put down apple 2.
Act: put down apple 2
You put down apple on kitchen table
Think: Now I accomplish the goal.
Act: done

### Agent Node 6
Your primary goal is to: Put the apple, pancake, cupcake, and juice on the kitchen
     table.
To achieve this, you should perform your sibling tasks in parallel. At this level,
     your sibling tasks are: move the apple on the kitchen table, move the pancake
     on the kitchen table, move the cupcake on the kitchen table, and move the
    juice on the kitchen table.
Your task is to: move the pancake on the kitchen table
You are in the house, and there are 4 rooms: bathroom (1), bedroom (1), kitchen
    (1), living room (1). You are in the middle of a kitchen (1). Looking quickly
    around the room, you see bench (1, 2), bookshelf (1, 2), chair (1, 2),
    dishwasher (1), faucet (2), fridge (1, 2), garbage can (1), kitchen cabinet
    (1, 2, 3, 4, 5, 6, 7, 8), kitchen counter (1, 2), kitchen table (1), microwave
    oven (1), sink (2), stove (1), toaster (1).
Think: To solve the task, I need to accomplish 2 subgoals sequentially. Find and
    pick up the pancake, and find the kitchen table and put down the pancake.
Expand:
```

```
 - control flow: sequence
 - subgoals: find and pick up the pancake, find the kitchen table and put down the
     pancake

### Agent Node 7
Your primary goal is to: move the pancake on the kitchen table
To achieve this, you should perform your sibling tasks in sequence. At this level,
    your sibling tasks are: find and pick up the pancake, and find the kitchen
    table and put down the pancake.
Your task is to: find and pick up the pancake
You are in the house, and there are 4 rooms: bathroom (1), bedroom (1), kitchen
    (1), living room (1). You are in the middle of a kitchen (1). Looking quickly
    around the room, you see bench (1, 2), bookshelf (1, 2), chair (1, 2),
    dishwasher (1), faucet (2), fridge (1, 2), garbage can (1), kitchen cabinet
    (1, 2, 3, 4, 5, 6, 7, 8), kitchen counter (1, 2), kitchen table (1), microwave
    oven (1), sink (2), stove (1), toaster (1).
Think: To solve the task, I need to recall the location of the pancake
Act: recall location of pancake
You have not seen pancake before.
Think: Now, I need to find and pick up the pancake in one of the rooms. The
    pancake is more likely to appear in kitchen 1, living room 1, or bedroom 1. I
    can check each room one by one until I find and pick up the pancake.
Expand
 - control flow: fallback
 - subgoals: find and pick up the pancake in kitchen 1, find and pick up the
     pancake in living room 1, find and pick up the pancake in bedroom 1

### Agent Node 8
Your primary goal is to: find and pick up the pancake
To achieve this, you should perform your sibling tasks using a fallback strategy.
    At this level, your sibling tasks are: find and pick up the pancake in kitchen
    1, find and pick up the pancake in living room 1, and find and pick up the
    pancake in bedroom 1.
Your task is to: find and pick up the pancake in kitchen 1
You are in the house, and there are 4 rooms: bathroom (1), bedroom (1), kitchen
    (1), living room (1). You are in the middle of a kitchen (1). Looking quickly
    around the room, you see bench (1, 2), bookshelf (1, 2), chair (1, 2),
    dishwasher (1), faucet (2), fridge (1, 2), garbage can (1), kitchen cabinet
    (1, 2, 3, 4, 5, 6, 7, 8), kitchen counter (1, 2), kitchen table (1), microwave
    oven (1), sink (2), stove (1), toaster (1).
Think: I am in kitchen 1. Now, I need to find the pancake. The pancake is more
    likely to appear on kitchen counter (1, 2), kitchen table (1), or in fridge
    (1, 2), kitchen cabinet (1, 2, 3, 4, 5, 6, 7, 8). I can check each place one
    by one, starting from kitchen counter 1, until I find the pancake.
Act: go to kitchen counter 1
You arrive at the kitchen counter (1). You see bowl (1, 2, 3, 4, 5), chair (2),
    chicken (1), cutlets (1), dishwashing liquid (1), faucet (2), kitchen cabinet
    (1, 2, 3, 4, 5, 6, 7), kitchen counter (1), plate (1, 2, 3, 4), sink (2),
    washing sponge (1)
Act: go to kitchen counter 2
You arrive at the kitchen counter (2). You see chair (1, 2), kitchen cabinet (5,
    6, 7, 8), kitchen counter (1, 2), light switch (2), plate (1, 2, 3, 4)
Act: go to kitchen table 1
You arrive at the kitchen table (1). You see apple (2), bench (2), book (1), bowl
    (6), condiment shaker (1, 2), kitchen table (1)
Act: go to fridge 1
You arrive at the fridge (1). The fridge (1) is open. You see bench (1), cupcake
    (1), fridge (1), pudding (3)
Act: go to fridge 2
You arrive at the fridge (2). The fridge (2) is open. You see bench (1), cupcake
    (1), fridge (1, 2), pudding (3)
Act: go to kitchen cabinet 1
```

```
You arrive at the kitchen cabinet (1). The kitchen cabinet (1) is closed. You see
    chicken (1), cooking pot (1), cutlery knife (1, 2), cutlets (1), dishwashing
    liquid (1), faucet (2), frying pan (1), garbage can (1), kitchen cabinet (1,
    4), kitchen counter (1), sink (2), stove (1)
Act: go to kitchen cabinet 2
You arrive at the kitchen cabinet (2). The kitchen cabinet (2) is closed. You see
    bowl (1, 2, 3, 4, 5), chair (2), chicken (1), cutlery knife (1, 2), cutlets
    (1), dishwashing liquid (1), faucet (2), kitchen cabinet (1, 2, 3, 4, 5, 6),
    kitchen counter (1), plate (1, 2, 3, 4), sink (2), washing sponge (1)
Act: go to kitchen cabinet 3
You arrive at the kitchen cabinet (3). The kitchen cabinet (3) is closed. You see
    bowl (1, 2, 3, 4, 5), chair (1, 2), dishwashing liquid (1), faucet (2),
    kitchen cabinet (2, 3, 4, 5, 6, 7), kitchen counter (1, 2), plate (1, 2, 3, 4)
    , sink (2), washing sponge (1)
Act: go to kitchen cabinet 4
You arrive at the kitchen cabinet (4). The kitchen cabinet (4) is closed. You see
    bowl (1, 2, 3, 4, 5), chicken (1), cooking pot (1), cutlery knife (1, 2),
    cutlets (1), dishwashing liquid (1), faucet (2), garbage can (1), kitchen
    cabinet (1, 2, 4), kitchen counter (1), sink (2), washing sponge (1)
Act: go to kitchen cabinet 5
You arrive at the kitchen cabinet (5). The kitchen cabinet (5) is closed. You see
    bowl (1, 2, 3, 4, 5), chair (1, 2), cutlets (1), dishwashing liquid (1),
    faucet (2), kitchen cabinet (2, 3, 4, 5, 6, 7), kitchen counter (1, 2), plate
    (1, 2, 3, 4), sink (2), washing sponge (1)
Act: go to kitchen cabinet 6
You arrive at the kitchen cabinet (6). The kitchen cabinet (6) is closed. You see
    bowl (1, 2, 3, 4, 5), chair (1, 2), dishwashing liquid (1), faucet (2),
    kitchen cabinet (2, 3, 4, 5, 6, 7, 8), kitchen counter (1, 2), plate (1, 2, 3,
     4), sink (2), washing sponge (1)
Act: go to kitchen cabinet 7
You arrive at the kitchen cabinet (7). The kitchen cabinet (7) is closed. You see
    bowl (1, 2, 3, 4, 5), chair (1, 2), kitchen cabinet (2, 5, 6, 7, 8), kitchen
    counter (1), light switch (2), plate (1, 2, 3, 4), sink (2)
Act: go to kitchen cabinet 8
You arrive at the kitchen cabinet (8). The kitchen cabinet (8) is closed. You see
    bench (2), chair (1, 2), kitchen cabinet (5, 6, 7, 8), light switch (2)
Think: I cannot find the pancake in kitchen 1.
Act: failure

### Agent Node 9
Your primary goal is to: find and pick up the pancake
To achieve this, you should perform your sibling tasks using a fallback strategy.
    At this level, your sibling tasks are: find and pick up the pancake in kitchen
     1, find and pick up the pancake in living room 1, and find and pick up the
    pancake in bedroom 1.
Your task is to: find and pick up the pancake in living room 1
You are in the house, and there are 4 rooms: bathroom (1), bedroom (1), kitchen
    (1), living room (1). You are in the middle of a kitchen (1). Looking quickly
    around the room, you see bench (1, 2), bookshelf (1, 2), chair (1, 2),
    dishwasher (1), faucet (2), fridge (1, 2), garbage can (1), kitchen cabinet
    (1, 2, 3, 4, 5, 6, 7, 8), kitchen counter (1, 2), kitchen table (1), microwave
     oven (1), sink (2), stove (1), toaster (1).
Think: I am in kitchen 1. Now, I need to go to living room 1.
Act: go to living room 1
You move to the living room (1). Looking quickly around the room, you see
    bookshelf (3), chair (3), closet (1), coffee table (1), computer (1), desk (1)
    , garbage can (2), nightstand (1), sofa (1), tv (1).
Think: Now, I need to find the pancake. The pancake is more likely to appear on
    coffee table (1), desk (1), or in closet (1). I can check each place one by
    one, starting from coffee table 1, until I find the pancake.
Act: go to coffee table 1
You arrive at the coffee table (1). You see alcohol (2), bananas (1), book (3, 6),
     bowl (7), cell phone (1, 2), coffee table (1), juice (1), lime (1), mug (1),
    plum (1, 2), sofa (1)
```

```
Act: go to desk 1
You arrive at the desk (1). You see book (7), chair (3), computer (1), desk (1),
    nightstand (1), water glass (2)
Act: go to closet 1
You arrive at the closet (1). The closet (1) is closed. You see closet (1), light
    switch (3)
Act: open closet 1
You open closet. You see closet (1), light switch (3)
Think: I cannot find the pancake in living room 1.
Act: failure

### Agent Node 10
Your primary goal is to: find and pick up the pancake
To achieve this, you should perform your sibling tasks using a fallback strategy.
    At this level, your sibling tasks are: find and pick up the pancake in kitchen
     1, find and pick up the pancake in living room 1, and find and pick up the
    pancake in bedroom 1.
Your task is to: find and pick up the pancake in bedroom 1
You are in the house, and there are 4 rooms: bathroom (1), bedroom (1), kitchen
    (1), living room (1). You are in the middle of a living room (1). Looking
    quickly around the room, you see bookshelf (3), chair (3), closet (1), coffee
    table (1), computer (1), desk (1), garbage can (2), nightstand (1), sofa (1),
    tv (1).
Think: I am in living room 1. Now, I need to go to bedroom 1.
Act: go to bedroom 1
You move to the bedroom (1). Looking quickly around the room, you see bed (1),
    bookshelf (4), cabinet (1), chair (4), closet (2), coffee table (2), computer
    (2), desk (2, 3), garbage can (3).
Think: Now, I need to find the pancake. The pancake is more likely to appear on
    bed (1), coffee table (2), desk (2, 3), or in cabinet (1). I can check each
    place one by one, starting from bed 1, until I find the pancake.
Act: go to bed 1
You arrive at the bed (1). You see bed (1), candy bar (1), coffee table (2),
    pancake (2)
Think: I find pancake 2. Now, I need to pick it up.
Act: pick up pancake 2
You pick up pancake. You hold pancake (2).
Think: Now I accomplish the goal.
Act: done

### Agent Node 11
Your primary goal is to: move the pancake on the kitchen table
To achieve this, you should perform your sibling tasks in sequence. At this level,
     your sibling tasks are: find and pick up the pancake, and find the kitchen
    table and put down the pancake.
Your task is to: find the kitchen table and put down the pancake
You are in the house, and there are 4 rooms: bathroom (1), bedroom (1), kitchen
    (1), living room (1). You are in the middle of a bedroom (1). Looking quickly
    around the room, you see bed (1), bookshelf (4), cabinet (1), chair (4),
    closet (2), coffee table (2), computer (2), desk (2, 3), garbage can (3). You
    hold pancake (2).
Think: I hold pancake 2. Now, I need to find the kitchen table. The kitchen table
    is more likely to appear in kitchen 1.
Act: go to kitchen 1
You move to the kitchen (1). Looking quickly around the room, you see bench (1, 2)
    , bookshelf (1, 2), chair (1, 2), dishwasher (1), faucet (2), fridge (1, 2),
    garbage can (1), kitchen cabinet (1, 2, 3, 4, 5, 6, 7, 8), kitchen counter (1,
     2), kitchen table (1), microwave oven (1), sink (2), stove (1), toaster (1).
    You hold pancake (2).
Think: I can see kitchen table 1.
Act: go to kitchen table 1
```

```
You arrive at the kitchen table (1). You see apple (2), bench (2), book (1), bowl
    (6), condiment shaker (1, 2), kitchen table (1), pancake (2) You hold pancake
    (2).
Think: Now, I need to put down pancake 2.
Act: put down pancake 2
You put down pancake on kitchen table
Think: Now I accomplish the goal.
Act: done

### Agent Node 11
Your primary goal is to: Put the apple, pancake, cupcake, and juice on the kitchen
    table.
To achieve this, you should perform your sibling tasks in parallel. At this level,
    your sibling tasks are: move the apple on the kitchen table, move the pancake
    on the kitchen table, move the cupcake on the kitchen table, and move the
    juice on the kitchen table.
Your task is to: move the cupcake on the kitchen table
You are in the house, and there are 4 rooms: bathroom (1), bedroom (1), kitchen
    (1), living room (1). You are in the middle of a kitchen (1). Looking quickly
    around the room, you see bench (1, 2), bookshelf (1, 2), chair (1, 2),
    dishwasher (1), faucet (2), fridge (1, 2), garbage can (1), kitchen cabinet
    (1, 2, 3, 4, 5, 6, 7, 8), kitchen counter (1, 2), kitchen table (1), microwave
    oven (1), sink (2), stove (1), toaster (1).
Think: To solve the task, I need to accomplish 2 subgoals sequentially. Find and
    pick up the cupcake, and find the kitchen table and put down the cupcake.
Expand:
 - control flow: sequence
 - subgoals: find and pick up the cupcake, find the kitchen table and put down the
     cupcake

### Agent Node 12
Your primary goal is to: move the cupcake on the kitchen table
To achieve this, you should perform your sibling tasks in sequence. At this level,
    your sibling tasks are: find and pick up the cupcake, and find the kitchen
    table and put down the cupcake.
Your task is to: find and pick up the cupcake
You are in the house, and there are 4 rooms: bathroom (1), bedroom (1), kitchen
    (1), living room (1). You are in the middle of a kitchen (1). Looking quickly
    around the room, you see bench (1, 2), bookshelf (1, 2), chair (1, 2),
    dishwasher (1), faucet (2), fridge (1, 2), garbage can (1), kitchen cabinet
    (1, 2, 3, 4, 5, 6, 7, 8), kitchen counter (1, 2), kitchen table (1), microwave
    oven (1), sink (2), stove (1), toaster (1).
Think: To solve the task, I need to recall the location of the cupcake
Act: recall location of cupcake
You saw cupcake 1 near fridge 2 in kitchen 1.
Think: Now, I need to go to fridge 2.
Act: go to fridge 2
You arrive at the fridge (2). The fridge (2) is open. You see bench (1), cupcake
    (1), fridge (1, 2), pudding (3)
Think: I find cupcake 1. Now, I need to pick it up.
Act: pick up cupcake 1
You pick up cupcake. You hold cupcake (1).
Think: Now I accomplish the goal.
Act: done

### Agent Node 13
Your primary goal is to: move the cupcake on the kitchen table
To achieve this, you should perform your sibling tasks in sequence. At this level,
    your sibling tasks are: find and pick up the cupcake, and find the kitchen
    table and put down the cupcake.
Your task is to: find the kitchen table and put down the cupcake
```

```
You are in the house, and there are 4 rooms: bathroom (1), bedroom (1), kitchen
    (1), living room (1). You are in the middle of a kitchen (1). Looking quickly
    around the room, you see bench (1, 2), bookshelf (1, 2), chair (1, 2),
    dishwasher (1), faucet (2), fridge (1, 2), garbage can (1), kitchen cabinet
    (1, 2, 3, 4, 5, 6, 7, 8), kitchen counter (1, 2), kitchen table (1), microwave
    oven (1), sink (2), stove (1), toaster (1). You hold cupcake (1).
Think: I hold cupcake 1. Now, I need to find the kitchen table. I can see kitchen
    table 1.
Act: go to kitchen table 1
You arrive at the kitchen table (1). You see apple (2), bench (2), book (1), bowl
    (6), condiment shaker (1, 2), cupcake (1), kitchen table (1), pancake (2) You
    hold cupcake (1).
Think: Now, I need to put down cupcake 1.
Act: put down cupcake 1
You put down cupcake on kitchen table
Think: Now I accomplish the goal.
Act: done

### Agent Node 14
Your primary goal is to: Put the apple, pancake, cupcake, and juice on the kitchen
    table.
To achieve this, you should perform your sibling tasks in parallel. At this level,
    your sibling tasks are: move the apple on the kitchen table, move the pancake
    on the kitchen table, move the cupcake on the kitchen table, and move the
    juice on the kitchen table.
Your task is to: move the juice on the kitchen table
You are in the house, and there are 4 rooms: bathroom (1), bedroom (1), kitchen
    (1), living room (1). You are in the middle of a kitchen (1). Looking quickly
    around the room, you see bench (1, 2), bookshelf (1, 2), chair (1, 2),
    dishwasher (1), faucet (2), fridge (1, 2), garbage can (1), kitchen cabinet
    (1, 2, 3, 4, 5, 6, 7, 8), kitchen counter (1, 2), kitchen table (1), microwave
    oven (1), sink (2), stove (1), toaster (1).
Think: To solve the task, I need to accomplish 2 subgoals sequentially. Find and
    pick up the juice, and find the kitchen table and put down the juice.
Expand:
 - control flow sequence
 - subgoals: find and pick up the juice, find the kitchen table and put down the
    juice

### Agent Node 15
Your primary goal is to: move the juice on the kitchen table
To achieve this, you should perform your sibling tasks in sequence. At this level,
    your sibling tasks are: find and pick up the juice, and find the kitchen
    table and put down the juice.
Your task is to: find and pick up the juice
You are in the house, and there are 4 rooms: bathroom (1), bedroom (1), kitchen
    (1), living room (1). You are in the middle of a kitchen (1). Looking quickly
    around the room, you see bench (1, 2), bookshelf (1, 2), chair (1, 2),
    dishwasher (1), faucet (2), fridge (1, 2), garbage can (1), kitchen cabinet
    (1, 2, 3, 4, 5, 6, 7, 8), kitchen counter (1, 2), kitchen table (1), microwave
    oven (1), sink (2), stove (1), toaster (1).
Think: To solve the task, I need to recall the location of the juice.
Act: recall location of juice
You saw juice 1 near coffee table 1 in living room 1.
Think: I can find juice 1 near coffee table 1.
Act: go to living room 1
You move to the living room (1). Looking quickly around the room, you see
    bookshelf (3), chair (3), closet (1), coffee table (1), computer (1), desk (1)
    , garbage can (2), nightstand (1), sofa (1), tv (1).
Act: go to coffee table 1
You arrive at the coffee table (1). You see alcohol (2), bananas (1), book (3, 6),
    bowl (7), cell phone (1, 2), coffee table (1), juice (1), lime (1), mug (1),
    plum (1, 2), sofa (1)
```

```
Think: I find juice 1. Now I need to pick it up.
Act: pick up juice 1
You pick up juice. You hold juice (1).
Think: Now I accomplish the goal.
Act: done

### Agent 16
Your primary goal is to: move the juice on the kitchen table
To achieve this, you should perform your sibling tasks in sequence. At this level,
    your sibling tasks are: find and pick up the juice, and find the kitchen
    table and put down the juice.
Your task is to: find the kitchen table and put down the juice
You are in the house, and there are 4 rooms: bathroom (1), bedroom (1), kitchen
    (1), living room (1). You are in the middle of a living room (1). Looking
    quickly around the room, you see bookshelf (3), chair (3), closet (1), coffee
    table (1), computer (1), desk (1), garbage can (2), nightstand (1), sofa (1),
    tv (1). You hold juice (1).
Think: I hold juice 1. Now, I need to find the kitchen table. The kitchen table is
    more likely to appear in kitchen 1.
Act: go to kitchen 1
You move to the kitchen (1). Looking quickly around the room, you see bench (1, 2)
    , bookshelf (1, 2), chair (1, 2), dishwasher (1), faucet (2), fridge (1, 2),
    garbage can (1), kitchen cabinet (1, 2, 3, 4, 5, 6, 7, 8), kitchen counter (1,
     2), kitchen table (1), microwave oven (1), sink (2), stove (1), toaster (1).
    You hold juice (1).
Think: I can see kitchen table 1.
Act: go to kitchen table 1
You arrive at the kitchen table (1). You see apple (2), bench (2), book (1), bowl
    (6), condiment shaker (1, 2), cupcake (1), juice (1), kitchen table (1),
    pancake (2) You hold juice (1).
Think: Now, I need to put down juice 1.
Act: put down juice 1
You put down juice on kitchen table
Think: Now I accomplish the goal.
Act: done
```

Listing 12: Failure trajectory of ReAct on the WAH-NL dataset using LLaMA 3 70B.

```
Your task is to: Put the apple, pancake, cupcake, and juice on the kitchen table.
You are in the house, and there are 4 rooms: bathroom (1), bedroom (1), kitchen
    (1), living room (1). You are in the middle of a kitchen (1). Looking quickly
    around the room, you see bench (1, 2), bookshelf (1, 2), chair (1, 2),
    dishwasher (1), faucet (2), fridge (1, 2), garbage can (1), kitchen cabinet
    (1, 2, 3, 4, 5, 6, 7, 8), kitchen counter (1, 2), kitchen table (1), microwave
    oven (1), sink (2), stove (1), toaster (1).
Think: To solve the task, I need to find and pick up the apple, then put it on the
    kitchen table. And I need to find and pick up the pancake, then put it on the
    kitchen table. And I need to find and pick up the cupcake, then put it on the
    kitchen table. And I need to find and pick up the juice, then put it on the
    kitchen table.
Think: First, I need to find the apple. The apple is more likely to appear in
    kitchen 1.
Act: go to kitchen 1
You move to the kitchen (1). Looking quickly around the room, you see bench (1, 2)
    , bookshelf (1, 2), chair (1, 2), dishwasher (1), faucet (2), fridge (1, 2),
    garbage can (1), kitchen cabinet (1, 2, 3, 4, 5, 6, 7, 8), kitchen counter (1,
     2), kitchen table (1), microwave oven (1), sink (2), stove (1), toaster (1).
Think: Now, I need to find the apple. The apple is more likely to appear on
    kitchen conter (1, 2), kitchen table (1), or in fridge (1, 2), kitchen cabinet
     (1, 2, 3, 4, 5, 6, 7, 8). I can check each place one by one, starting from
    kitchen counter 1, until I find the apple.
Act: go to kitchen counter 1
You arrive at the kitchen counter (1). You see bowl (1, 2, 3, 4, 5), chair (2),
    chicken (1), cutlets (1), dishwashing liquid (1), faucet (2), kitchen cabinet
```

```
            (1, 2, 3, 4, 5, 6, 7), kitchen counter (1), plate (1, 2, 3, 4), sink (2),
            washing sponge (1)
Act: go to kitchen counter 2
You arrive at the kitchen counter (2). You see chair (1, 2), kitchen cabinet (5,
    6, 7, 8), kitchen counter (1, 2), light switch (2), plate (1, 2, 3, 4)
Act: go to kitchen table 1
You arrive at the kitchen table (1). You see bench (2), book (1), bowl (6),
    condiment shaker (1, 2), kitchen table (1)
Act: go to fridge 1
You arrive at the fridge (1). The fridge (1) is closed. You see bench (1), fridge
    (1)
Act: open fridge 1
You open fridge. You see bench (1), cupcake (1), fridge (1), pudding (3)
Think: I find cupcake 1. Now, I need to pick it up.
Act: pick up cupcake 1
You pick up cupcake. You hold cupcake (1).
Think: I hold cupcake 1. Now, I need to put it on the kitchen table.
Act: go to kitchen table 1
You arrive at the kitchen table (1). You see bench (2), book (1), bowl (6),
    condiment shaker (1, 2), cupcake (1), kitchen table (1) You hold cupcake (1).
Think: Now, I need to put down cupcake 1.
Act: put down cupcake 1
You put down cupcake on kitchen table
Think: I put down cupcake 1 on kitchen table 1. Now, I need to recall the location
     of the pancake.
Act: recall location of pancake
You have not seen pancake before.
Think: Now, I need to find the pancake. The pancake is more likely to appear in
    kitchen cabinet (1, 2, 3, 4, 5, 6, 7, 8). I can check each place one by one,
    starting from kitchen cabinet 1, until I find the pancake.
Act: go to kitchen cabinet 1
You arrive at the kitchen cabinet (1). The kitchen cabinet (1) is closed. You see
    chicken (1), cooking pot (1), cutlery knife (1, 2), cutlets (1), dishwashing
    liquid (1), faucet (2), frying pan (1), garbage can (1), kitchen cabinet (1,
    4), kitchen counter (1), sink (2), stove (1)
Act: open kitchen cabinet 1
You open kitchen cabinet. You see chicken (1), cooking pot (1), cutlery knife (1,
    2), cutlets (1), dishwashing liquid (1), faucet (2), frying pan (1), garbage
    can (1), kitchen cabinet (1, 4), kitchen counter (1), sink (2), stove (1)
Act: go to kitchen cabinet 2
You arrive at the kitchen cabinet (2). The kitchen cabinet (2) is closed. You see
    bowl (1, 2, 3, 4, 5), chair (2), chicken (1), cutlery knife (1, 2), cutlets
    (1), dishwashing liquid (1), faucet (2), kitchen cabinet (1, 2, 3, 4, 5, 6),
    kitchen counter (1), plate (1, 2, 3, 4), sink (2), washing sponge (1)
Act: open kitchen cabinet 2
You open kitchen cabinet. You see bowl (1, 2, 3, 4, 5), chair (2), chicken (1),
    cutlery knife (1, 2), cutlets (1), dishwashing liquid (1), faucet (2), kitchen
     cabinet (1, 2, 3, 4, 5, 6), kitchen counter (1), plate (1, 2, 3, 4), sink (2)
    , washing sponge (1)
Act: go to kitchen cabinet 3
You arrive at the kitchen cabinet (3). The kitchen cabinet (3) is closed. You see
    bowl (1, 2, 3, 4, 5), chair (1, 2), dishwashing liquid (1), faucet (2),
    kitchen cabinet (2, 3, 4, 5, 6, 7), kitchen counter (1, 2), plate (1, 2, 3, 4)
    , sink (2), washing sponge (1)
Act: open kitchen cabinet 3
You open kitchen cabinet. You see bowl (1, 2, 3, 4, 5), chair (1, 2), dishwashing
    liquid (1), faucet (2), kitchen cabinet (2, 3, 4, 5, 6, 7), kitchen counter
    (1, 2), plate (1, 2, 3, 4), sink (2), washing sponge (1)
Act: go to kitchen cabinet 4
You arrive at the kitchen cabinet (4). The kitchen cabinet (4) is closed. You see
    bowl (1, 2, 3, 4, 5), chicken (1), cooking pot (1), cutlery knife (1, 2),
    cutlets (1), dishwashing liquid (1), faucet (2), garbage can (1), kitchen
    cabinet (1, 2, 4), kitchen counter (1), sink (2), washing sponge (1)
Act: open kitchen cabinet 4
```

```
2268  You open kitchen cabinet. You see bowl (1, 2, 3, 4, 5), chicken (1), cooking pot
2269     (1), cutlery knife (1, 2), cutlets (1), dishwashing liquid (1), faucet (2),
2270     garbage can (1), kitchen cabinet (1, 2, 4), kitchen counter (1), sink (2),
2271     washing sponge (1)
2272  Act: go to kitchen cabinet 5
2273  You arrive at the kitchen cabinet (5). The kitchen cabinet (5) is closed. You see
2274     bowl (1, 2, 3, 4, 5), chair (1, 2), cutlets (1), dishwashing liquid (1),
2275     faucet (2), kitchen cabinet (2, 3, 4, 5, 6, 7), kitchen counter (1, 2), plate
       (1, 2, 3, 4), sink (2), washing sponge (1)
2276  Act: open kitchen cabinet 5
2277  You open kitchen cabinet. You see bowl (1, 2, 3, 4, 5), chair (1, 2), cutlets (1),
2278      dishwashing liquid (1), faucet (2), kitchen cabinet (2, 3, 4, 5, 6, 7),
2279     kitchen counter (1, 2), plate (1, 2, 3, 4), pudding (2), sink (2), washing
       sponge (1)
2280  Think: I find pudding 2. Now, I need to pick it up.
2281  Act: pick up pudding 2
2282  You pick up pudding. You hold pudding (2).
2283  Think: I hold pudding 2. Now, I need to put it on the kitchen table.
2284  Act: go to kitchen table 1
2285  You arrive at the kitchen table (1). You see bench (2), book (1), bowl (6),
2286     condiment shaker (1, 2), cupcake (1), kitchen table (1), pudding (2) You hold
       pudding (2).
2287  Think: Now, I need to put down pudding 2.
2288  Act: put down pudding 2
2289  You put down pudding on kitchen table
2290  Think: I put down pudding 2 on kitchen table 1. Now, I need to recall the location
        of the juice.
2291  Act: recall location of juice
2292  You have not seen juice before.
2293  Think: Now, I need to find the juice. The juice is more likely to appear in
2294     kitchen cabinet (1, 2, 3, 4, 5, 6, 7, 8). I can check each place one by one,
       starting from kitchen cabinet 1, until I find the juice.
2295  Act: go to kitchen cabinet 6
2296  You arrive at the kitchen cabinet (6). The kitchen cabinet (6) is closed. You see
2297     bowl (1, 2, 3, 4, 5), chair (1, 2), dishwashing liquid (1), faucet (2),
2298     kitchen cabinet (2, 3, 4, 5, 6, 7, 8), kitchen counter (1, 2), plate (1, 2, 3,
2299      4), sink (2), washing sponge (1)
2300  Act: open kitchen cabinet 6
      You open kitchen cabinet. You see bowl (1, 2, 3, 4, 5), chair (1, 2), dishwashing
2301     liquid (1), faucet (2), kitchen cabinet (2, 3, 4, 5, 6, 7, 8), kitchen counter
2302      (1, 2), plate (1, 2, 3, 4), sink (2), washing sponge (1)
2303  Act: go to kitchen cabinet 7
2304  You arrive at the kitchen cabinet (7). The kitchen cabinet (7) is closed. You see
2305     bowl (1, 2, 3, 4, 5), chair (1, 2), kitchen cabinet (2, 3, 4, 5, 6, 7, 8),
       kitchen counter (1), light switch (2), plate (1, 2, 3, 4), sink (2)
2306  Act: open kitchen cabinet 7
2307  You open kitchen cabinet. You see apple (1), bowl (1, 2, 3, 4, 5), chair (1, 2),
2308     kitchen cabinet (2, 3, 4, 5, 6, 7, 8), kitchen counter (1), light switch (2),
       pancake (1), plate (1, 2, 3, 4), sink (2)
2309  Think: I find apple 1. Now, I need to pick it up.
2310  Act: pick up apple 1
2311  You pick up apple. You hold apple (1).
2312  Think: I hold apple 1. Now, I need to put it on the kitchen table.
2313  Act: go to kitchen table 1
2314  You arrive at the kitchen table (1). You see apple (1), bench (2), book (1), bowl
2315     (6), condiment shaker (1, 2), cupcake (1), kitchen table (1), pudding (2) You
       hold apple (1).
2316  Think: Now, I need to put down apple 1.
2317  Act: put down apple 1
2318  You put down apple on kitchen table
2319  Think: I put down apple 1 on kitchen table 1. Now, I need to recall the location
       of the pancake.
2320  Act: recall location of pancake
2321  You saw pancake 1 near kitchen cabinet 7 in kitchen 1.
      Act: go to kitchen cabinet 7
```

```
You arrive at the kitchen cabinet (7). The kitchen cabinet (7) is open. You see
    bowl (1, 2, 3, 4, 5), chair (1, 2), kitchen cabinet (2, 3, 4, 5, 6, 7, 8),
    kitchen counter (1), light switch (2), pancake (1), plate (1, 2, 3, 4), sink
    (2)
Think: I find pancake 1. Now, I need to pick it up.
Act: pick up pancake 1
You pick up pancake. You hold pancake (1).
Think: I hold pancake 1. Now, I need to put it on the kitchen table.
Act: go to kitchen table 1
You arrive at the kitchen table (1). You see apple (1), bench (2), book (1), bowl
    (6), condiment shaker (1, 2), cupcake (1), kitchen table (1), pancake (1),
    pudding (2) You hold pancake (1).
Think: Now, I need to put down pancake 1.
Act: put down pancake 1
You put down pancake on kitchen table
Think: I put down pancake 1 on kitchen table 1. Now, I need to recall the location
     of the juice.
Act: recall location of juice
You have not seen juice before.
Think: Now, I need to find the juice. The juice is more likely to appear in
    kitchen cabinet (1, 2, 3, 4, 5, 6, 7, 8). I can check each place one by one,
    starting from kitchen cabinet 1, until I find the juice.
Act: go to kitchen cabinet 8
You arrive at the kitchen cabinet (8). The kitchen cabinet (8) is closed. You see
    bench (2), chair (1, 2), kitchen cabinet (5, 6, 7, 8), light switch (2)
Act: open kitchen cabinet 8
You open kitchen cabinet. You see bench (2), chair (1, 2), kitchen cabinet (5, 6,
    7, 8), light switch (2)
Think: I cannot find juice in kitchen 1. The juice is more likely to appear in
    fridge 1, 2.
Act: go to fridge 1
You arrive at the fridge (1). The fridge (1) is open. You see bench (1), fridge
    (1), pudding (3)
Act: go to fridge 2
You arrive at the fridge (2). The fridge (2) is closed. You see bench (1), fridge
    (1, 2), pudding (3)
Act: open fridge 2
You open fridge. You see apple (2), bench (1), fridge (1, 2), pudding (3)
Think: I find apple 2. Now, I need to pick it up.
Act: pick up apple 2
You pick up apple. You hold apple (2).
Think: I hold apple 2. Now, I need to put it on the kitchen table.
Act: go to kitchen table 1
You arrive at the kitchen table (1). You see apple (1, 2), bench (2), book (1),
    bowl (6), condiment shaker (1, 2), cupcake (1), kitchen table (1), pancake (1)
    , pudding (2) You hold apple (2).
Think: Now, I need to put down apple 2.
Act: put down apple 2
You put down apple on kitchen table
Think: I put down apple 2 on kitchen table 1. Now, I need to recall the location
    of the juice.
Act: recall location of juice
You have not seen juice before.
Failure: maximum context window size of 8192
```

## H.2 ALFRED EXAMPLE CASES

Listing 13: Success trajectory of ReAcTree on the ALFRED dataset using LLaMA 3 70B.

```
### Agent Node 1
Your task is to: Put the cooked tomato slice on the kitchen island
You are in the house, and you arrive at the kitchen, Looking quickly around the
    room, you see Toaster (1), StoveBurner (1, 2, 3, 4, 5, 6), SinkBasin (1),
    Microwave (1), GarbageCan (1), Fridge (1), Drawer (1, 2, 3, 4, 5, 6, 7, 8, 9,
```

```
10, 11, 12, 13), CounterTop (1, 2), CoffeeMachine (1), Cabinet (1, 2, 3, 4, 5,
    6, 7, 8, 9).
Think: To solve the task, I need to accomplish 8 subgoals sequentially. find and
    pick up the knife, find and slice and then pick up the tomato, find and open
    the microwave, find the microwave and put down the tomato, find and close and
    turn on the microwave, find and turn off and open microwave, find and pick up
    tomato in microwave, and find a countertop and put down the tomato on the
    countertop.
Expand:
- control flow: sequence
- subgoals: find and pick up the knife, find and slice and then pick up the tomato
    , find and open the microwave, find the microwave and put down the tomato,
    find and close and turn on the microwave, find and turn off and open microwave
    , find and pick up tomato in microwave, and find a countertop and put down the
     tomato on the countertop.

### Agent Node 2
Your primary goal is to: Put the cooked tomato slice on the kitchen island
To achieve this, you should perform your sibling tasks in sequence. At this level,
     your sibling tasks are: find and pick up the knife, find and slice and then
    pick up the tomato, find and open the microwave, find the microwave and put
    down the tomato, find and close and turn on the microwave, find and turn off
    and open microwave, find and pick up tomato in microwave, and and find a
    countertop and put down the tomato on the countertop.
Your task is to: find and pick up the knife
You are in the house, and you arrive at the kitchen, Looking quickly around the
    room, you see Toaster (1), StoveBurner (1, 2, 3, 4, 5, 6), SinkBasin (1),
    Microwave (1), GarbageCan (1), Fridge (1), Drawer (1, 2, 3, 4, 5, 6, 7, 8, 9,
    10, 11, 12, 13), CounterTop (1, 2), CoffeeMachine (1), Cabinet (1, 2, 3, 4, 5,
     6, 7, 8, 9).
Think: To solve the task, I need to recall location of knife.
Think: Now I need to find Knife. A knife is more likely to appear in CounterTop
    (1-2), SinkBasin (1), Cabinet (1-9). I can check one by one, starting with
    CounterTop (1).
Act: go to CounterTop (1)
You arrive at CounterTop (1). You see Window (2), Tomato (1), Sink (1), SinkBasin
    (1), Lettuce (2), Fork (1), Faucet (1), Egg (2), DishSponge (1), CounterTop
    (1), Cabinet (1, 8).
Act: go to CounterTop (2)
You arrive at CounterTop (2). You see Tomato (2), SoapBottle (2), Pot (1),
    PepperShaker (2), Mug (2), Knife (1, 2), Kettle (2), CounterTop (1, 2),
    CellPhone (2), Bread (1, 2).
Think: Now I find Knife (1, 2). Next, I need to pick it up.
Act: pick up Knife (1)
You pick up Knife (1). You see Tomato (2), SoapBottle (2), Pot (1), PepperShaker
    (2), Mug (2), Knife (1, 2), Kettle (2), CounterTop (1, 2), CellPhone (2),
    Bread (1, 2). I am holding Knife (1) now.
Think: Now I find and pick up Knife (1)
Act: done

### Agent Node 3
Your primary goal is to: Put the cooked tomato slice on the kitchen island
To achieve this, you should perform your sibling tasks in sequence. At this level,
     your sibling tasks are: find and pick up the knife, find and slice and then
    pick up the tomato, find and open the microwave, find the microwave and put
    down the tomato, find and close and turn on the microwave, find and turn off
    and open microwave, find and pick up tomato in microwave, and and find a
    countertop and put down the tomato on the countertop.
Your task is to: find and slice and then pick up the tomato
You are in the house, and you arrive at the kitchen, Looking quickly around the
    room, you see Toaster (1), StoveBurner (1, 2, 3, 4, 5, 6), SinkBasin (1),
    Microwave (1), GarbageCan (1), Fridge (1), Drawer (1, 2, 3, 4, 5, 6, 7, 8, 9,
```

```
      10, 11, 12, 13), CounterTop (1, 2), CoffeeMachine (1), Cabinet (1, 2, 3, 4, 5,
         6, 7, 8, 9). I am holding Knife (1) now.
Think: To solve the task, I need to recall location of tomato.
Think: I saw Tomato (1) near SinkBasin (1). I need to go to SinkBasin (1), then
      slice the Tomato (1).
Act: go to SinkBasin (1)
You arrive at SinkBasin (1). You see Tomato (1), Sink (1), SinkBasin (1), Lettuce
      (2), Knife (1), Fork (1), Faucet (1), Egg (2), Drawer (1, 6), DishSponge (1),
      CounterTop (1), Cabinet (1, 8).  I am holding Knife (1) now.
Think: Now I find a part of the Tomato (1). I need to go in front of the Tomato
      (1) to see the whole thing.
Act: go to Tomato (1)
You see Tomato (1), Sink (1), SinkBasin (1), Lettuce (2), Knife (1), Fork (1),
      Faucet (1), Egg (2), Drawer (1, 6), DishSponge (1), CounterTop (1), Cabinet
      (1, 8).  I am holding Knife (1) now.
Act: slice Tomato (1)
You slice Tomato (1). You see Tomato (9, 10), Sink (1), SinkBasin (1), Lettuce (2)
      , Knife (1), Fork (1), Faucet (1), Egg (2), Drawer (1, 6), DishSponge (1),
      CounterTop (1), Cabinet (1, 8). I am holding Knife (1) now.
Think: Now I slice the Tomato (1), and I do not need to hold Knife (1). I need to
      drop the Knife (1).
Act: put down Knife (1)
You put down failed, drop it. You see Tomato (9, 10), Sink (1), SinkBasin (1),
      Lettuce (2), Fork (1), Faucet (1), Egg (2), Drawer (1, 6), DishSponge (1),
      CounterTop (1), Cabinet (1, 8).
Think: Now I put down Knife (1). Next, I need to pick up the Tomato (9).
Act: pick up Tomato (9)
You pick up Tomato (9). You see Tomato (7, 8, 9, 10), Sink (1), SinkBasin (1),
      Lettuce (2), Fork (1), Faucet (1), Egg (2), Drawer (1, 6), DishSponge (1),
      CounterTop (1), Cabinet (1, 8). I am holding Tomato (9) now.
Think: Now I pick up Tomato (9).
Act: done

### Agent Node 4
Your primary goal is to: Put the cooked tomato slice on the kitchen island
To achieve this, you should perform your sibling tasks in sequence. At this level,
       your sibling tasks are: find and pick up the knife, find and slice and then
      pick up the tomato, find and open the microwave, find the microwave and put
      down the tomato, find and close and turn on the microwave, find and turn off
      and open microwave, find and pick up tomato in microwave, and and find a
      countertop and put down the tomato on the countertop.
Your task is to: find and open the microwave
You are in the house, and you arrive at the kitchen, Looking quickly around the
      room, you see Toaster (1), StoveBurner (1, 2, 3, 4, 5, 6), SinkBasin (1),
      Microwave (1), GarbageCan (1), Fridge (1), Drawer (1, 2, 3, 4, 5, 6, 7, 8, 9,
      10, 11, 12, 13), CounterTop (1, 2), CoffeeMachine (1), Cabinet (1, 2, 3, 4, 5,
         6, 7, 8, 9). I am holding Tomato (9) now.
Act: go to Microwave (1)
You arrive at Microwave (1). You see Window (1), Tomato (9), Toaster (1),
      Microwave (1), Drawer (4), CounterTop (1). Microwave (1) is closed.  I am
      holding Tomato (9) now.
Act: open Microwave (1)
You open Microwave (1). You see Window (1), Tomato (9), Toaster (1), Microwave (1)
      , Egg (1), Drawer (4), CounterTop (1). I am holding Tomato (9) now.
Think: Now I find and open the Microwave (1).
Act: done

### Agent Node 5
Your primary goal is to: Put the cooked tomato slice on the kitchen island
To achieve this, you should perform your sibling tasks in sequence. At this level,
       your sibling tasks are: find and pick up the knife, find and slice and then
      pick up the tomato, find and open the microwave, find the microwave and put
      down the tomato, find and close and turn on the microwave, find and turn off
```

```
        and open microwave, find and pick up tomato in microwave, and and find a
        countertop and put down the tomato on the countertop.
Your task is to: find the microwave and put down the tomato
You are in the house, and you arrive at the kitchen, Looking quickly around the
    room, you see Toaster (1), StoveBurner (1, 2, 3, 4, 5, 6), SinkBasin (1),
    Microwave (1), GarbageCan (1), Fridge (1), Drawer (1, 2, 3, 4, 5, 6, 7, 8, 9,
    10, 11, 12, 13), CounterTop (1, 2), CoffeeMachine (1), Cabinet (1, 2, 3, 4, 5,
    6, 7, 8, 9). I am holding Tomato (9) now.
Think: To solve the task, I need to go to Microwave (1) and put down the Tomato
    (9).
Act: go to Microwave (1)
You arrive at Microwave (1). You see Window (1), Tomato (9), Toaster (1),
    Microwave (1), Egg (1), Drawer (4), CounterTop (1). Microwave (1) is open.  I
    am holding Tomato (9) now.
Think: Now I find a Microwave (1). The Microwave (1) is already open, so I need to
     put the Tomato (9) down.
Act: put down Tomato (9)
You put Tomato (9) on Microwave (1). You see Window (1), Tomato (9), Toaster (1),
    Microwave (1), Egg (1), Drawer (4), CounterTop (1).
Think: Now I put down Tomato (9) in the Microwave (1).
Act: done

### Agent Node 6
Your primary goal is to: Put the cooked tomato slice on the kitchen island
To achieve this, you should perform your sibling tasks in sequence. At this level,
     your sibling tasks are: find and pick up the knife, find and slice and then
    pick up the tomato, find and open the microwave, find the microwave and put
    down the tomato, find and close and turn on the microwave, find and turn off
    and open microwave, find and pick up tomato in microwave, and and find a
    countertop and put down the tomato on the countertop.
Your task is to: find and close and turn on the microwave
You are in the house, and you arrive at the kitchen, Looking quickly around the
    room, you see Toaster (1), StoveBurner (1, 2, 3, 4, 5, 6), SinkBasin (1),
    Microwave (1), GarbageCan (1), Fridge (1), Drawer (1, 2, 3, 4, 5, 6, 7, 8, 9,
    10, 11, 12, 13), CounterTop (1, 2), CoffeeMachine (1), Cabinet (1, 2, 3, 4, 5,
    6, 7, 8, 9).
Think: To solve the task, I need to go to Microwave (1), close the Microwave (1)
    and turn on the Microwave (1).
Act: go to Microwave (1)
You arrive at Microwave (1). You see Window (1), Tomato (9), Toaster (1),
    Microwave (1), Egg (1), Drawer (4), CounterTop (1). Microwave (1) is open.
Think: The Microwave (1) is already open and I need to close the Microwave (1).
Act: close Microwave (1)
You close Microwave (1). You see Window (1), Toaster (1), Microwave (1), Drawer
    (4), CounterTop (1).
Act: turn on Microwave (1)
You turn on Microwave (1). You see Window (1), Toaster (1), Microwave (1), Drawer
    (4), CounterTop (1).
Think: Now I find and close and turn on the Microwave (1).
Act: done

### Agent Node 7
Your primary goal is to: Put the cooked tomato slice on the kitchen island
To achieve this, you should perform your sibling tasks in sequence. At this level,
     your sibling tasks are: find and pick up the knife, find and slice and then
    pick up the tomato, find and open the microwave, find the microwave and put
    down the tomato, find and close and turn on the microwave, find and turn off
    and open microwave, find and pick up tomato in microwave, and and find a
    countertop and put down the tomato on the countertop.
Your task is to: find and turn off and open microwave
You are in the house, and you arrive at the kitchen, Looking quickly around the
    room, you see Toaster (1), StoveBurner (1, 2, 3, 4, 5, 6), SinkBasin (1),
    Microwave (1), GarbageCan (1), Fridge (1), Drawer (1, 2, 3, 4, 5, 6, 7, 8, 9,
```

```
10, 11, 12, 13), CounterTop (1, 2), CoffeeMachine (1), Cabinet (1, 2, 3, 4, 5,
    6, 7, 8, 9).
Think: To solve the task, I need to go to Microwave (1), turn off and open the
    Microwave (1).
Act: go to Microwave (1)
You arrive at Microwave (1). You see Window (1), Toaster (1), Microwave (1),
    Drawer (4), CounterTop (1). Microwave (1) is closed. Microwave (1) is already
    turned on.
Think: Now I find a Microwave (1). The Microwave (1) is already turned on, so I
    need to turn off the Microwave (1).
Act: turn off Microwave (1)
You turn off Microwave (1). You see Window (1), Toaster (1), Microwave (1), Drawer
     (4), CounterTop (1).
Act: open Microwave (1)
You open Microwave (1). You see Window (1), Tomato (9), Toaster (1), Microwave (1)
    , Egg (1), Drawer (4), CounterTop (1).
Think: Now I turn off and open the Microwave (1).
Act: done

### Agent Node 8
Your primary goal is to: Put the cooked tomato slice on the kitchen island
To achieve this, you should perform your sibling tasks in sequence. At this level,
     your sibling tasks are: find and pick up the knife, find and slice and then
    pick up the tomato, find and open the microwave, find the microwave and put
    down the tomato, find and close and turn on the microwave, find and turn off
    and open microwave, find and pick up tomato in microwave, and and find a
    countertop and put down the tomato on the countertop.
Your task is to: find and pick up tomato in microwave
You are in the house, and you arrive at the kitchen, Looking quickly around the
    room, you see Toaster (1), StoveBurner (1, 2, 3, 4, 5, 6), SinkBasin (1),
    Microwave (1), GarbageCan (1), Fridge (1), Drawer (1, 2, 3, 4, 5, 6, 7, 8, 9,
    10, 11, 12, 13), CounterTop (1, 2), CoffeeMachine (1), Cabinet (1, 2, 3, 4, 5,
    6, 7, 8, 9).
Think: To solve the task, I need to recall the location of tomato.
Think: I saw Tomato (2) near Microwave (1). I need to go to Microwave (1), then
    pick up Tomato (2).
Act: go to Microwave (1)
You arrive at Microwave (1). You see Window (1), Tomato (9), Toaster (1),
    Microwave (1), Egg (1), Drawer (4), CounterTop (1). Microwave (1) is open.
Think: Now, I find a Microwave (1). The Microwave (1) is already open, so I need
    to pick up Tomato (2).
Act: pick up Tomato (9)
You pick up Tomato (9). You see Window (1), Tomato (9), Toaster (1), Microwave (1)
    , Egg (1), Drawer (4), CounterTop (1). I am holding Tomato (9) now.
Think: Now I find and pick up the Tomato (9) in Microwave (1).
Act: done

### Agent Node 9
Your primary goal is to: Put the cooked tomato slice on the kitchen island
To achieve this, you should perform your sibling tasks in sequence. At this level,
     your sibling tasks are: find and pick up the knife, find and slice and then
    pick up the tomato, find and open the microwave, find the microwave and put
    down the tomato, find and close and turn on the microwave, find and turn off
    and open microwave, find and pick up tomato in microwave, and and find a
    countertop and put down the tomato on the countertop.
Your task is to: and find a countertop and put down the tomato on the countertop.
You are in the house, and you arrive at the kitchen, Looking quickly around the
    room, you see Toaster (1), StoveBurner (1, 2, 3, 4, 5, 6), SinkBasin (1),
    Microwave (1), GarbageCan (1), Fridge (1), Drawer (1, 2, 3, 4, 5, 6, 7, 8, 9,
    10, 11, 12, 13), CounterTop (1, 2), CoffeeMachine (1), Cabinet (1, 2, 3, 4, 5,
    6, 7, 8, 9). I am holding Tomato (9) now.
Think: To solve the task, I need to go to CounterTop (1) and put down the Tomato
    (9) on the CounterTop (1).
```

```
Act: go to CounterTop (1)
You arrive at CounterTop (1). You see Window (2), Tomato (7, 8, 9, 10), Sink (1),
    SinkBasin (1), Lettuce (2), Fork (1), Faucet (1), Egg (2), DishSponge (1),
    CounterTop (1), Cabinet (1, 8).  I am holding Tomato (9) now.
Act: put down Tomato (9)
You put Tomato (9) on CounterTop (1). You see Window (2), Tomato (7, 8, 9, 10),
    Sink (1), SinkBasin (1), Lettuce (2), Fork (1), Faucet (1), Egg (2),
    DishSponge (1), CounterTop (1), Cabinet (1, 8).
Think: Now I put Tomato (9) on the CounterTop (1).
Act: done
```

Listing 14: Failure trajectory of ReAct on the ALFRED dataset using LLaMA 3 70B.

```
Your task is to : Put the cooked tomato slice on the kitchen island
You are in the house, and you arrive at the kitchen, Looking quickly around the
    room, you see Toaster (1), StoveBurner (1, 2, 3, 4, 5, 6), SinkBasin (1),
    Microwave (1), GarbageCan (1), Fridge (1), Drawer (1, 2, 3, 4, 5, 6, 7, 8, 9,
    10, 11, 12, 13), CounterTop (1, 2), CoffeeMachine (1), Cabinet (1, 2, 3, 4, 5,
     6, 7, 8, 9).
Think: To solve the task, I need to find and pick up knife, then find and slice
    tomato and then pick up tomato, then find plate and put tomato on plte, then
    find and pick up plate, then find fridge and put plate on fridge
Think: First, I need to recall location of knife
Act: recall location of Knife
You have not seen Knife before.
Think: Now I need to find and pick up knife. A knife is more likely to appear in
    SinkBasin (1), Sink (1), DiningTable (1), CounterTop (1-3), Cabinet (1-2),
    Drawer (1-6). I can check one by one, starting with SinkBasin (1).
Act: go to SinkBasin (1)
You arrive at SinkBasin (1). You see Tomato (1), Sink (1), SinkBasin (1), Lettuce
    (2), Fork (1), Faucet (1), Egg (2), Drawer (1, 6), DishSponge (1), CounterTop
    (1), Cabinet (1, 8).
Think: Now I find Knife (1). Next, I need to pick it up.
Act: pick up Tomato (1)
You pick up Tomato (1). You see Tomato (1), Sink (1), SinkBasin (1), Lettuce (2),
    Fork (1), Faucet (1), Egg (2), Drawer (1, 6), DishSponge (1), CounterTop (1),
    Cabinet (1, 8). I am holding Tomato (1) now.
Think: Now I find and pick up Knife (1).
Think: Next, I need to recall location of tomato
Act: recall location of Tomato
You are holding Tomato (1).
Think: I saw Tomato (3) near SinkBasin (1)
Act: go to SinkBasin (1)
You arrive at SinkBasin (1). You see Tomato (1), Sink (1), SinkBasin (1), Lettuce
    (2), Fork (1), Faucet (1), Egg (2), Drawer (1, 6), DishSponge (1), CounterTop
    (1), Cabinet (1, 8).  I am holding Tomato (1) now.
Think: Now I find Tomato (3). Next, I need to slice it
Act: slice Tomato (1)
The Tomato (1) slice failed. I am holding Tomato (1) now.
Think: Now I slice Tomato (3). Next, I need to drop Knife (1) because I do not
    need to hold it now.
Act: go to SinkBasin (1)
You arrive at SinkBasin (1). You see Tomato (1), Sink (1), SinkBasin (1), Lettuce
    (2), Fork (1), Faucet (1), Egg (2), Drawer (1, 6), DishSponge (1), CounterTop
    (1), Cabinet (1, 8).  I am holding Tomato (1) now.
Think: Next, I need to pick up Tomato (5)
Act: pick up Tomato (1)
You are currently holding Tomato (1). I am holding Tomato (1) now.
Think: Now I find and slice Tomato (3), then pick up Tomato (5)
Think: Next, I need to recall location of plate.
Act: recall location of Plate
You have not seen Plate before.
Think: I saw Plate (1) near SinkBasin (1).
Act: go to SinkBasin (1)
```

```
You arrive at SinkBasin (1). You see Tomato (1), Sink (1), SinkBasin (1), Lettuce
    (2), Fork (1), Faucet (1), Egg (2), Drawer (1, 6), DishSponge (1), CounterTop
    (1), Cabinet (1, 8).  I am holding Tomato (1) now.
Think: I need to go to Plate (1), then put Tomato (5) on Plate (1)
Act: go to SinkBasin (1)
You arrive at SinkBasin (1). You see Tomato (1), Sink (1), SinkBasin (1), Lettuce
    (2), Fork (1), Faucet (1), Egg (2), Drawer (1, 6), DishSponge (1), CounterTop
    (1), Cabinet (1, 8).  I am holding Tomato (1) now.
Act: put down Tomato (1)
You put Tomato (1) on SinkBasin (1). You see Tomato (1), Sink (1), SinkBasin (1),
    Lettuce (2), Fork (1), Faucet (1), Egg (2), Drawer (1, 6), DishSponge (1),
    CounterTop (1), Cabinet (1, 8).
Think: Now I find Plate (1), then put Tomato (5) on Plate (1)
Think: Next, I need to pick up Plate (1) again.
Act: pick up Tomato (1)
You pick up Tomato (1). You see Tomato (1), Sink (1), SinkBasin (1), Lettuce (2),
    Fork (1), Faucet (1), Egg (2), Drawer (1, 6), DishSponge (1), CounterTop (1),
    Cabinet (1, 8). I am holding Tomato (1) now.
Think: Now I find and pick up Plate (1). Next, I need to find fridge, then put
    plate on fridge.
Act: go to SinkBasin (1)
You arrive at SinkBasin (1). You see Tomato (1), Sink (1), SinkBasin (1), Lettuce
    (2), Fork (1), Faucet (1), Egg (2), Drawer (1, 6), DishSponge (1), CounterTop
    (1), Cabinet (1, 8).  I am holding Tomato (1) now.
Act: open Fridge (1)
The Fridge (1) is not close to you. I am holding Tomato (1) now.
Act: put down Tomato (1)
You put Tomato (1) on SinkBasin (1). You see Tomato (1), Sink (1), SinkBasin (1),
    Lettuce (2), Fork (1), Faucet (1), Egg (2), Drawer (1, 6), DishSponge (1),
    CounterTop (1), Cabinet (1, 8).
Think: Now I put Plate (1) on Fridge (1).
Act: done
```

