# OpenReview forum: "ReAcTree: Hierarchical Task Planning with Dynamic Tree Expansion using LLM Agent Nodes"
_ICLR.cc/2025/Conference — Submitted to ICLR 2025_

### Official Review · Reviewer_HETB · 2024-10-26

**Soundness:** 3
**Presentation:** 3
**Contribution:** 3
**Rating:** 6
**Confidence:** 4

**Summary:**

This paper introduces ReAcTree, a novel hierarchical task planning framework that leverages Large Language Models (LLMs) to decompose complex tasks into manageable subgoals using a dynamic tree structure.

**Strengths:**

1. The dynamic tree expansion mechanism effectively combines the structure of behavior trees with the flexibility of LLM-based planning.
2. The experiments cover multiple LLM architectures and sizes, demonstrating the framework's robustness.
3. The framework addresses real-world constraints through partial observability in the experimental setup.
4. The improvement compared to ReAct is significant.

**Weaknesses:**

1. The paper seems to lack a comparison with other tree-based planning methods [1-3]. This makes it difficult to contextualize the contribution within the broader landscape of hierarchical planning approaches.
2. The abbreviation "WM" in Table 1 is not explicitly defined in the caption or text, reducing clarity.
3. The paper doesn't discuss the computational cost of maintaining and expanding the tree structure compared to sequential approaches.
4. There could be more discussion about when and why the system fails, which would help understand its limitations.

references:

[1] Yao S, Yu D, Zhao J, et al. Tree of thoughts: Deliberate problem solving with large language models[J]. Advances in Neural Information Processing Systems, 2024, 36.

[2] Hu M, Mu Y, Yu X, et al. Tree-planner: Efficient close-loop task planning with large language models[J]. arXiv preprint arXiv:2310.08582, 2023.

[3] Zhao Z, Lee W S, Hsu D. Large language models as commonsense knowledge for large-scale task planning[J]. Advances in Neural Information Processing Systems, 2024, 36.

**Questions:**

See weaknesses. Further:

1. What is the impact of different memory retrieval strategies on the system's performance?
2. How does ReAcTree handle cycles or repetitive tasks that might require loop structures?
3. Given that the LLM needs to perform repeated planning at different nodes in the tree structure, is there a potential issue of error accumulation? When mistakes occur at higher levels of the tree, how do they propagate through the subsequent planning steps? It would be valuable to see an analysis of error propagation and any mechanisms in place to mitigate such cascading failures.

I would consider raising my score based on the authors' response

---

> ### Author Response · Authors · 2024-11-22
>
> We appreciate for your thorough review and constructive feedback. We would like to respond to your comments on the weaknesses and questions.
>
> > **[W1] The paper seems to lack a comparison with other tree-based planning methods. This makes it difficult to contextualize the contribution within the broader landscape of hierarchical planning approaches.**
>
> Thank you for pointing this out. We clarify the key differences between ReAcTree and prior tree-based planning methods, address the limitations of these approaches, and highlight the advantages of ReAcTree:
>
> 1. Key differences: While prior studies like ToT [1] and GoT [2] focus on constructing **action/thought trees** to explore future actions or thoughts and choose the best next step, ReAcTree constructs an **LLM agent tree** for hierarchical task decomposition. When a goal is complex, ReAcTree divides it into simpler manageable subgoals, assigns specialized LLM agent nodes to each subgoal, and determines their execution strategy (e.g., sequence, fallback) using control flow nodes.
>
> 2. Limitations of Prior Studies:
> - ToT [1] and GoT [2]: These methods rely on tree search algorithms to look ahead and evaluate multiple future paths. They require accurate state predictions and evaluations to determine the optimal action. However, in uncertain environments like households, such state predictions are impractical due to the unpredictable outcomes of actions. Consequently, in both papers, the methods were applied to tasks with deterministic or predictable state transitions, such as reasoning tasks (e.g., Game of 24, Sorting) where future states are straightforward to predict and evaluate.
> - LLM-MCTS [3]: LLM-MCTS is applied in household environment, VirtualHome, but relies on ground-truth transition functions to simulate future states. This dependency limits its applicability to real-world household settings where such transition functions are unavailable.
> - Tree-Planner [4]: Tree-Planner adopts a real-world-oriented approach by constructing action trees and grounded decoding to execute action and get observation without explicit state prediction. However, it assumes the feasibility of inverse actions (e.g., "pick"-"place", "switch on"-"switch off") for backtracking. This assumption is often violated in complex tasks like ALFRED, where certain actions, such as "slice," cannot be reversed. Additionally, Tree-Planner still attempts to solve complex task without dividing them into subgoals.
>
> [1] Yao, Shunyu, et al. "Tree of thoughts: Deliberate problem solving with large language models." Advances in Neural Information Processing Systems 36 (2024). \
> [2] Besta, Maciej, et al. "Graph of thoughts: Solving elaborate problems with large language models." Proceedings of the AAAI Conference on Artificial Intelligence. Vol. 38. No. 16. 2024. \
> [3] Zhao, Zirui, Wee Sun Lee, and David Hsu. "Large language models as commonsense knowledge for large-scale task planning." Advances in Neural Information Processing Systems 36 (2024). \
> [4] Hu, Mengkang, et al. "Tree-planner: Efficient close-loop task planning with large language models." arXiv preprint arXiv:2310.08582 (2023).
>
> 3. Advantages of ReAcTree:
> - **No State Prediction**: ReAcTree avoids relying on look-ahead or state evaluation, which are impractical in household task planning with unpredictable state transitions.
> - **Dynamic Subgoal Decomposition**: By dynamically decomposing complex goals into simpler, manageable subgoals and assigning specialized agent nodes to handle each subgoal, ReAcTree allows each agent node to solve a simpler problem compared to the original task. This modular design also enables leveraging task-specific in-context examples for each agent node.
> - **Control Flow Integration**: The incorporation of control flow nodes enables efficient execution strategies without exhaustive tree traversal or backtracking.
>
> To clarify these differences, we have revised the **Section 2** to provide a detailed comparison with these methods. While their approaches address distinct challenges, we recognize their contributions to the field of tree-based reasoning. We believe that insights from these approaches could inspire future work on hybrid methods combining hierarchical decomposition with selective tree search.
>
> > **[W2] The abbreviation "WM" in Table 1 is not explicitly defined in the caption or text, reducing clarity.**
>
> Thank you for pointing out this oversight. To address this, we have explicitly defined that WM refers to working memory in the caption of **Table 1 of Section 5.4** to improve clarity and ensure readers can easily understand the context.

---

> ### Author Response · Authors · 2024-11-22
>
> (continued)
>
> > **[W3] The paper doesn't discuss the computational cost of maintaining and expanding the tree structure compared to sequential approaches.**
>
> The computational cost of both ReAct and ReAcTree **primarily arises from the LLM's decision-making process**. To address this concern, we compared the average number of decision steps required by ReAct and ReAcTree for tasks successfully completed by both methods (with working memory). Using the QWEN-2.5 72B model, the average number of decision steps was **58.08 for ReAct** and **75.00 for ReAcTree**. The higher count in ReAcTree stems from the additional reasoning and expanding actions integral to its hierarchical task planning, enabling it to handle more complex tasks effectively. Despite the higher number of decisions, ReAcTree achieves significantly higher success rates while keeping computational costs manageable. Moreover, unlike ReAct, which continuously accumulates decision-making results and observations, ReAcTree resets the input prompt whenever a new agent node is created, providing a slight computational advantage. This analysis has been updated in the revised manuscript, Section 5.2.
>
> > **[W4] There could be more discussion about when and why the system fails, which would help understand its limitations.**
>
> To address this, we have conducted a detailed analysis of the ReAcTree's failure cases, categorized into four types: *Expand-level*, *Agent-level*, *Constraints*, and *Instruction errors*. For instance, *Expand-level* failures include issues in task decomposition or control-flow selection, while *Agent-level* failures involve navigation, search, or termination reasoning errors. Additionally, *Constraints* failures arise from exceeding maximum decision counts, and *Instruction errors* occur due to incorrect or ambiguous instructions in the dataset.
>
> We further compared failure cases with and without working memory. ReAcTree without working memory exhibited 41 failures, whereas incorporating working memory reduced this to 37. In both cases, *Agent-level* failures were the most prevalent. These findings, along with detailed discussions and examples, are included in **Appendix E**. We believe this analysis enhances the understanding of ReAcTree's limitations and provides valuable insights for future improvements.
>
> > **[Q1] What is the impact of different memory retrieval strategies on the system's performance?**
>
> To address this question, we conducted additional experiments with two memory retrieval strategies.
>
> 1. Weighted Similarity Strategy: This strategy combines the cosine similarity of the agent node's goal sentence $g^n$ and initial observation sentence $o^n$ with the corresponding sentence $g^e, o^e$ stored in episodic memory. The similarity is calculated as: $similarity(n,e)=a_1{\cdot}cosinesim(g^n,g^e) + a_2{\cdot}cosinesim(o^n,o^e)$. Here $a_1$ and $a_2$ are weights that balance the contributions of goal and observation similarities, and we used 0.9 and 0.1, respectively.
>
> 2. Balanced Sampling Strategy: In this approach, episodic memory entries are divided into three groups based on the termination state: success, failure, and expand. Entries within each group are sorted by cosine similarity, and one entry is selected from the highest-ranked group for each retrieval, ensuring a balanced distribution of termination states.
>
> We tested these strategies using the QWEN-2.5 7B model with working memory enabled. The results are as follows:
> | Retrieval Strategy       | GSR  | SSR   |
> |--------------------------|------|-------|
> | Original                | 35.00 | 58.80 |
> | Weighted Similarity      | 24.00 | 48.30 |
> | Balanced Sampling        | 22.00 | 45.08 |
>
> The original retrieval strategy outperforms both alternatives. While the weighted similarity strategy introduces diversity in retrieval, it fails to ensure a balanced selection of termination states, which limits its effectiveness. The balanced sampling strategy, on the other hand, includes less relevant examples due to forced selection from each termination state, leading to reduced performance.
>
> We acknowledge that more advanced retrieval strategies could further optimize memory selection by balancing diversity and relevance. Exploring such approaches is an important direction for future work.

---

> ### Author Response · Authors · 2024-11-22
>
> (continued)
>
> > **[Q2] How does ReAcTree handle cycles or repetitive tasks that might require loop structures?**
>
> Currently, ReAcTree does not explicitly handle cycles or repetitive tasks that require loop structures. However, we believe this limitation could be addressed by introducing a new type of control flow node that allows repetitive execution of child nodes for a specified number of iterations or until a condition is met. For instance, a "loop node" could be added to the existing control flow types (e.g., sequence, fallback, parallel). This node would execute its child nodes $N$ times or continue looping until a success condition is achieved.
>
> We believe this extension could significantly enhance ReAcTree's flexibility and adaptability for tasks requiring repetition. Although not currently implemented, this is an important consideration for future improvements.
>
> > **[Q3] Given that the LLM needs to perform repeated planning at different nodes in the tree structure, is there a potential issue of error accumulation? When mistakes occur at higher levels of the tree, how do they propagate through the subsequent planning steps? It would be valuable to see an analysis of error propagation and any mechanisms in place to mitigate such cascading failures.**
>
> We acknowledge that error accumulation is a potential limitation of ReAcTree, especially when mistakes occur at higher levels of the tree. Currently, ReAcTree does not include mechanisms to correct errors or adjust the overall tree structure once planning has begun. As you noted, this absence of error correction can lead to cascading failures.
>
> In our failure case analysis (see Appendix E of the revised manuscript), we observed that high-level task decomposition errors are rare. For instance, when using QWEN-2.5 72B model with working memory, ReAcTree exhibited only 5 cases of task decomposition failures and 1 case of control flow selection failures.
>
> To address this limitation, a promising direction for future work is to introduce a verification module that evaluates the consistency of the overall tree structure during planning. Such a module could detect potential inconsistencies early in the process and prompt reevaluation at critical nodes, reducing the likelihood of cascading failures.
>
> ---
> We are grateful for your insightful feedback, and we believe that addressing these issues will enhance the overall quality of the paper. If you have any further suggestions or remaining concerns, please do not hesitate to communicate them.

---

> ### Author Response · Authors · 2024-11-27
> **Looking Forward to Your Feedback**
>
> Dear Reviewer HETB,
>
> Thank you once again for your thoughtful comments on our paper. We have carefully considered your comments in our rebuttal and made updates accordingly.
>
> With the revision deadline approaching, we kindly request your feedback on our submitted response. We hope our response has adequately addressed your concerns. If there are any additional clarifications or further questions, please do not hesitate to let us know.
>
> Thank you very much for your time and kind consideration!
>
> Best regards,
> The Authors

---

> > ### Comment · Reviewer_HETB · 2024-11-27
> >
> > Thanks for the authors' response. The response addresses most of my concerns and I raise my score from 5 to 6. It would be better if the authors included more experiments compared to tree-based planning methods, rather than ReAct alone.

---

### Official Review · Reviewer_xbbd · 2024-10-31

**Soundness:** 2
**Presentation:** 1
**Contribution:** 1
**Rating:** 3
**Confidence:** 4

**Summary:**

This paper introduces ReAcTree, a hierarchical task planning framework designed to handle complex, long-horizon tasks by decomposing them into smaller subgoals within a dynamic tree structure. ReAcTree uses control flow nodes and agent nodes, where each agent node functions as a task planner capable of reasoning, acting, and expanding goals. The model includes two memory components: episodic memory to store agent-specific experiences and working memory to share observations across nodes, allowing for more contextually relevant task planning.

**Strengths:**

1. ReAcTree’s approach to breaking down tasks into manageable subgoals enables it to handle complex, long-horizon tasks with greater efficiency than sequential approaches.

2. By incorporating episodic memory and working memory, ReAcTree improves the relevance of in-context examples and facilitates observation sharing across agent nodes, reducing task redundancy and improving task success rates.

**Weaknesses:**

1. The paper lacks a clear explanation of how memory (both episodic and working) is coupled with ReAcTree agents. The integration details are not shown in the algorithm, leaving critical explanations about memory interactions and their role in ReAcTree agents underdeveloped.

2. The paper would benefit from a substantial rewrite to improve clarity, particularly around key definitions, memory components, and the agent interaction mechanisms within ReAcTree.

3. The memory retrieval approach relies solely on cosine similarity without a discussion of its selection or adaptability as memory scales. Furthermore, it is unclear how the memory retrieval process integrates termination states and handles large datasets.

4. While the paper claims that shorter text trajectories improve retrieval efficiency, the ReAcTree framework targets complex, long-horizon tasks that likely produce lengthy trajectories, making this assertion unclear. Constraints or empirical evidence on trajectory length’s impact would add clarity.

**Questions:**

1. In L233-237, the authors introduce an **extended action space** for agent nodes, which appears to include a vast range of potential actions. Given this large action space, how does the approach address the **curse of dimensionality** in selecting and processing these actions efficiently?

2. In Algorithm 1, L11, the variable **$n_e$** is used, but it is not defined in the paper. Could you clarify what $n_e$ represents in this context?

3. Regarding the **episodic memory retrieval** mechanism:
   - How does the **cosine similarity** metric work in this context to retrieve relevant experiences? Why was this method chosen over alternative retrieval strategies?
   - How well does this method scale when the memory contains a **large number of stored experiences**?
   - What role does the **termination state $s$** play in memory retrieval? Specifically, does ReAcTree consider only successful trajectories, or are other outcomes (e.g., failures) also retrieved?

4. In L290-291, the authors mention that **shorter text trajectories** allow for more efficient context usage, facilitating a broader range of example retrieval. However:
   - Could you provide an **empirical study** or analysis supporting this claim?
   - Is there a **constraint** on the trajectory size for optimal efficiency in ReAcTree?
   - Given that ReAcTree aims to handle **complex, long-horizon tasks**, which often generate extensive text trajectories, how does the approach reconcile this apparent paradox between trajectory length and retrieval efficiency?

5. In L294-295, the paper describes **working memory** as part of the agent's skill set, $A_t^n$. This raises some clarity issues:
   - Could you clarify or redefine **skill set** to explain how working memory integrates within it?
   - Why is working memory modeled as an **action** in the task planning context, and how can it be "utilized as an action by agent nodes"?

---

> ### Author Response · Authors · 2024-11-22
>
> Thank you for your detailed review and thoughtful feedback. Your insights have highlighted several areas for improvement, particularly regarding memory integration and clarity in our framework, and we greatly appreciate the opportunity to address these concerns.
>
> > **[W1] The paper lacks a clear explanation of how memory (both episodic and working) is coupled with ReAcTree agents. The integration details are not shown in the algorithm, leaving critical explanations about memory interactions and their role in ReAcTree agents underdeveloped.**
>
> > **[Q5] In L294-295, the paper describes working memory as part of the agent's skill set, $A^n_t$. This raises some clarity issues: 1) Could you clarify or redefine skill set to explain how working memory integrates within it? 2) Why is working memory modeled as an action in the task planning context, and how can it be "utilized as an action by agent nodes"?**
>
> To address this feedback, we first explain how both memory components (episodic and working) are coupled with ReAcTree agent nodes. Subsequently, we provide a clearer explanation of their interactions and integration details.
>
> 1. Clarification on Episodic Memory (EM): EM is used to retrieve relevant past agent-level experiences and incorporate them as in-context examples before each agent node begins its decision-making process. To retrieve relevant experiences, we compute the cosine similarity between the current agent node's goal sentence and goal sentences stored in EM. Further details on the retrieval process (including terminate state) and computational cost are addressed in response to [W3] and [Q3].
>
> 2. Clarification on Working Memory (WM): WM is proposed to share key observations, such as the latest location of movable objects, across agent nodes. Its integration with agent nodes involves two key aspects: 1) When WM is enabled, the agent node's skill set $A^n_t$ is augmented with specialized actions for interacting with WM. For instance, in household tasks, "recall location of <movable object>" is added as an acting action. If this action is selected as the next action $a^n_t$, the agent node get location of the object from WM instead of interacting directly with the environment. 2) When the agent node interacts with environment and obtain observations, WM is updated with the latest locations of any movable objects detected. For instance, if an agent opens a fridge and observes juice, working memory updates the location of juice as near fridge for future use. These mechanisms allow agent nodes to effectively utilize WM for task planning without redundant interactions with the environment. We believe this explanation address your [Q5] and clarifies WM.
>
> To improve clarity, we have revised **Section 4.2** to include a detailed explanation of the interaction between EM, WM and ReAcTree agent nodes. We hope these revisions address the concerns and provide a clearer understanding of memory integration in ReAcTree.
>
> > **[W2] The paper would benefit from a substantial rewrite to improve clarity, particularly around key definitions, memory components, and the agent interaction mechanisms within ReAcTree.**
>
> To address these points, we have rewritten **Section 4.2** of the manuscript to improve clarity. The revised section includes detailed explanations of the roles of both episodic memory and working memory, their integration with agent nodes, termination states in episodic memory, and skill set augmentation when working memory is utilized. These revisions aim to clarify how memory components interact with ReAcTree agent nodes and address the concerns raised regarding key definitions and agent interaction mechanisms.

---

> ### Author Response · Authors · 2024-11-22
>
> (continued)
> > **[W3] The memory retrieval approach relies solely on cosine similarity without a discussion of its selection or adaptability as memory scales. Furthermore, it is unclear how the memory retrieval process integrates termination states and handles large datasets.**
>
> > **[Q3] Regarding the episodic memory retrieval mechanism: 1) How does the cosine similarity metric work in this context to retrieve relevant experiences? Why was this method chosen over alternative retrieval strategies? 2) How well does this method scale when the memory contains a large number of stored experiences? 3) What role does the termination state $s$ play in memory retrieval? Specifically, does ReAcTree consider only successful trajectories, or are other outcomes (e.g., failures) also retrieved?**
>
> Below, we address your concerns by clarifying the cosine similarity-based retrieval method, discussing its scalability, and explaining the role of termination states.
>
> 1. **Cosine Similarity**-Based Retrieval Method: Each agent node $n$ retrieves in-context examples from episodic memory before starting its decision-making process. The retrieval is performed by comparing the **agent node's goal sentence $g^n$** with **stored goal sentences $g^e$ in episodic memory** using **cosine similarity**. Although simple, this approach has been effectively employed in several LLM-based task planning studies [1, 2, 3]. More elaborate retrieval methods [4] could be explored in future work. However, our contribution lies in demonstrating that even this simple retrieval approach works effectively in ReAcTree due to its hierarchical task decomposition. By breaking tasks into specific subgoals, ReAcTree ensures that relevant experiences are retrieved. For example, if the target goal of current agent is "Put one cupcake and one apple on the coffee table," retrieved top-3 similar goals are "Place one apple, one cupcake and one juice on the coffee table," "Please, put a single apple, one cupcake and one juice to the coffee table," and "Put 1 cupcake, 1 wine, and 1 apple on the coffee table."
>
> 2. Scalability with Large Episodic Memory: The retrieval mechanism is computationally efficient, even with large episodic memory. Let $f_{sen}$ represent the sentence embedding function (e.g., Sentence BERT), with an embedding dimension $d$, and let $N$ represent the number of stored experiences. The retrieval process involves: 1) The goal sentence is embedded by $f_{sen}(g^n)$. 2) The **cosine similarity is computed by performing a matrix multiplication** between the episodic memory embeddings ($N{\times}d$) and the goal embedding ($d{\times}1$). This retrieval process, with its linear computational complexity, remains efficient even for large-scale episodic memory $N$.
>
> 3. Role of Termination State $s$: The **termination state $s$ helps resolve ties** when multiple experiences in episodic memory have identical cosine similarity scores with the target goal. Each stored trajectory is labeled with a termination state indicating success, failure, or expansion. When ties occur, ReAcTree uniformly samples examples with the same similarity score across termination states (success, failure, expansion), ensuring diversity in the retrieved in-context examples and enhancing decision-making robustness.
>
> 4. Episodic Memory Content: Episodic memory stores the experiences of **all ReAcTree agent nodes that participated in successfully completed tasks**. Thus the stored experiences could be tagged with success, failure, or expand termination states.
>
> [1] Huang, Wenlong, et al. "Language models as zero-shot planners: Extracting actionable knowledge for embodied agents." International conference on machine learning. PMLR, 2022. \
> [2] Choi, Jae-Woo, et al. "Lota-bench: Benchmarking language-oriented task planners for embodied agents." arXiv preprint arXiv:2402.08178 (2024). \
> [3] Zhao, Zirui, Wee Sun Lee, and David Hsu. "Large language models as commonsense knowledge for large-scale task planning." Advances in Neural Information Processing Systems 36 (2024). \
> [4] Zhao, Andrew, et al. "Expel: Llm agents are experiential learners." Proceedings of the AAAI Conference on Artificial Intelligence. Vol. 38. No. 17. 2024.

---

> ### Author Response · Authors · 2024-11-22
>
> (continued)
> > **[W4] While the paper claims that shorter text trajectories improve retrieval efficiency, the ReAcTree framework targets complex, long-horizon tasks that likely produce lengthy trajectories, making this assertion unclear. Constraints or empirical evidence on trajectory length’s impact would add clarity.**
>
> > **[Q4] In L290-291, the authors mention that shorter text trajectories allow for more efficient context usage, facilitating a broader range of example retrieval. However: 1) Could you provide an empirical study or analysis supporting this claim? 2) Is there a constraint on the trajectory size for optimal efficiency in ReAcTree? 3) Given that ReAcTree aims to handle complex, long-horizon tasks, which often generate extensive text trajectories, how does the approach reconcile this apparent paradox between trajectory length and retrieval efficiency?**
>
> We address your concerns by explaining why short trajectories are effective for in-context example retrieval, how ReAcTree leverages this advantage thorough its hierarchical design, and the constraints imposed by the model’s context window.
>
> 1. Why Short Trajectories are Effective for In-Context Example Retrieval: Shorter trajectories enable more efficient context usage in two ways: 1) They reduce the total token count in the prompt, which decreases computational requirements for LLM inference. 2) They allow more examples within a fixed token budget. Several studies in LLM-based planning [2, 5] have shown that increasing the number of relevant in-context examples can improve performance up to a certain point.
>
> 2. Why ReAcTree Can Use Shorter Trajectories: ReAcTree naturally produces shorter trajectories due to its hierarchical task decomposition. By breaking down complex goals into manageable subgoals, ReAcTree stores the trajectories of individual agent nodes, which are shorter and more specialized compared to monolithic trajectories used in ReAct. For example, given the goal "Please bring one pudding and one juice to the coffee table," ReAct stores a single trajectory covering all reasoning, acting, and observations for the entire task. In contrast, ReAcTree decomposes the task into subgoals, such as "find and pick up pudding in kitchen 1," resulting in shorter, task-specific trajectories. In the episodic memory constructed using the LLaMA-3 70B model and employed in our experiments, **the average token count per trajectory was 2474.10 for ReAct and 459.47 for ReAcTree**.
>
> 3. Constraints on Trajectory Length: While there are no constraints on individual trajectory length, the overall prompt size is limited by the model's context window size. For example, with LLaMA-3's 8k token limit, we set the maximum size for in-context examples to 4k tokens, leaving sufficient room for the target planning trajectories.
>
> [2] Choi, Jae-Woo, et al. "Lota-bench: Benchmarking language-oriented task planners for embodied agents." arXiv preprint arXiv:2402.08178 (2024). \
> [5] Agarwal, Rishabh, et al. "Many-shot in-context learning." arXiv preprint arXiv:2404.11018 (2024).
>
> > **[Q1] In L233-237, the authors introduce an extended action space for agent nodes, which appears to include a vast range of potential actions. Given this large action space, how does the approach address the curse of dimensionality in selecting and processing these actions efficiently?**
>
> We clarify how ReAcTree's extended action space is structured and explain why it does not lead to significant issues with the curse of dimensionality.
> 1. Action Space Structure: ReAcTree extends the action space to $A{\cup}L{\cup}E$, where $A$ is executable skills set (e.g., "pick up juice") for acting, $L$ is language space for reasoning, and $E=F{\times}L$ for expanding.
> 2. Comparison with ReAct: While ReAct extends the action space to $A{\cup}L$, ReAcTree further introduces $E$ for hierarchical task decomposition. However, the additional complexity from $E$ is mitigated by the limited size of $F$, where $F$ is a small set of control flow types (in this work, $|F|=3$). The LLM first select the control flow type by scoring elements in $F$, and then generates subgoals within the $L$ space.
>
> > **[Q2] In Algorithm 1, L11, the variable $n_e$ is used, but it is not defined in the paper. Could you clarify what $n_e$ represents in this context?**
>
> Thank you for pointing out this oversight. The variable $n_e$ in Algorithm 1 is indeed a typographical error and should be written as $K$, which represents the number of subgoals generated during the expansion process. We have corrected this notation in the revised manuscript to ensure consistency throughout the paper.
>
> ---
> We are grateful for your insightful feedback, and we believe that addressing these issues will enhance the overall quality of the paper. If you have any further suggestions or remaining concerns, please do not hesitate to communicate them.

---

> > ### Comment · Reviewer_xbbd · 2024-11-25
> >
> > I appreciate the authors' detailed explanations. However, I have several questions:
> >
> > 1. Clarification on Episodic Memory: What is the definition of "agent-level experiences"? This concept appears in the abstract but lacks definition throughout the paper.
> > 2. Regarding Working Memory (WM): How are the agent node's skills augmented with specialized WM actions? Are these actions predefined for each task? Please clarify what you mean by "specialized actions interacting with WM."
> > 3. Why do you cite the "Lota-bench" paper when it doesn't mention cosine similarity?
> > 4. Regarding termination states: How can two trajectories have identical cosine similarity scores with the target goal? If trajectories are identical, what is the purpose of storing duplicates? Furthermore, why would identical trajectories have different termination states?
> > 5. I understand the benefits of short trajectories, but my question remains: "**the ReAcTree framework targets complex, long-horizon tasks that likely produce lengthy trajectories**" — how do you maintain short trajectories for these complex, long-horizon tasks?
> >
> > Overall, the responses have left my concerns unaddressed and introduced more unexplained details. The presence of numerous vague and undefined concepts makes the framework difficult to understand. Thus, my score remains the same.

---

> ### Author Response · Authors · 2024-11-25
>
> Thank you for your detailed feedback and the opportunity to clarify our approach. We greatly appreciate the time you’ve taken to review our work, and we have carefully addressed each of your questions below.
>
> > [Q1] Clarification on Episodic Memory: What is the definition of "agent-level experiences"? This concept appears in the abstract but lacks definition throughout the paper.
>
> **Agent-level experience** refers to the **experience generated by an individual agent node $e$ in its attempt to accomplish a specific goal sentence $g^e$**. If the final time step in the decision-making process of agent node $e$ is $T$, the text trajectory $t^e$ of the agent can be represented as $t^e = (g^e, o^e_1, a^e_1, \dots, o^e_T, a^e_T)$, where $o^e_t$ and $a^e_t$ denote the observation and action at each time step $t$. The agent-level experience is then defined as $(t^e, v^e, s^e)$, where $v^e = f_{\text{sen}}(g^e)$ is the embedding vector of the goal sentence $g^e$, computed using the sentence embedding model $f_{\text{sen}}$, and $s^e$ represents the termination state of the agent node, categorized as *success*, *failure*, or *expand*. **Note that the entire trajectory $t^e$ is stored in text form, while only the goal sentence $g^e$ is embedded to compute $v^e$.** We have updated **Section 4.2** to clearly define and elaborate on agent-level experiences.
>
> > [Q2] Regarding Working Memory (WM): How are the agent node's skills augmented with specialized WM actions? Are these actions predefined for each task? Please clarify what you mean by "specialized actions interacting with WM."
>
> **Specialized WM actions are predefined for all movable objects in the environment**. These actions are constructed as *recall location of <movable object>*, enabling agent nodes to query Working Memory to retrieve stored information, such as the last observed location of an object. We have updated **Section 4.2** to clarify that the specialized WM actions are predefined.
>
> > [Q3] Why do you cite the "Lota-bench" paper when it doesn't mention cosine similarity?
>
> The **LoTa-Bench paper uses the term "Semantic Similarity" instead of explicitly mentioning "cosine similarity"** in the text. However, the in-context example selection process in Section 6.1 employs cosine similarity, as confirmed by the implementation. **This can be verified in the official implementation**, which confirms the use of cosine similarity as the similarity metric. You can access the implementation at the following repository: https://github.com/lbaa2022/LLMTaskPlanning (Path: src/wah/wah_task_planner.py).
>
> > [Q4] Regarding termination states: How can two trajectories have identical cosine similarity scores with the target goal? If trajectories are identical, what is the purpose of storing duplicates? Furthermore, why would identical trajectories have different termination states?
>
> **Two trajectories cannot be entirely identical**, as each trajectory reflects the unique sequence of observations and actions taken by the agent node. **However, it is possible for two agent nodes, $n_1$ and $n_2$, to share the same goal sentence, $g^{n_1} = g^{n_2}$**, which would result in identical cosine similarity scores for the goal embeddings. For instance, in our experiments, there are multiple agent-level experiences with the goal sentence "find and pick up juice." Despite having the same goal, their trajectories differ due to variations in observations or actions based on the environment.

---

> ### Author Response · Authors · 2024-11-25
>
> (continued)
> > [Q5] I understand the benefits of short trajectories, but my question remains: "the ReAcTree framework targets complex, long-horizon tasks that likely produce lengthy trajectories" — how do you maintain short trajectories for these complex, long-horizon tasks?
>
> ReAcTree maintains short trajectories by decomposing complex, long-horizon tasks into smaller subgoals, **with each subgoal handled independently by individual agent nodes**. This hierarchical decomposition ensures that each **agent node operates within a manageable scope, allowing it to maintain a short trajectory even for complex tasks.**
>
> For example, **Appendix H.1, Listing 11** illustrates the trajectories of ReAcTree’s agent nodes for the task: "Put the apple, pancake, cupcake, and juice on the kitchen table." **The root node (Agent Node 1) performs only one reasoning action and one expanding action**, decomposing the goal into four subgoals: "move the apple on the kitchen table," "move the pancake on the kitchen table," "move the cupcake on the kitchen table," and "move the juice on the kitchen table," using a parallel control flow node. **This completes the text trajectory for Agent Node 1.**
>
> Similarly, **Agent Node 4**, with the goal sentence "find and pick up the apple in kitchen 1," **performs 12 actions** to locate and pick up the apple in the kitchen. This trajectory terminates successfully after achieving its specific subgoal.
>
> In contrast, **Listing 12 shows the trajectory of ReAct**, where the entire task is treated as a single trajectory, **performing 73 actions**. This approach results in **significantly longer trajectories compared to ReAcTree**. As mentioned **in our earlier response to [W4] and [Q4], the average token count per trajectory was 2474.10 for ReAct, compared to 459.47 for ReAcTree**. This significant reduction highlights the efficiency and modularity achieved by ReAcTree's hierarchical approach.
>
>
> This feedback process has been invaluable in clarifying the contributions of ReAcTree, and we are confident that its hierarchical task decomposition offers a robust and scalable solution for complex, long-horizon tasks, setting it apart from existing approaches like ReAct.

---

> ### Author Response · Authors · 2024-11-27
> **Looking Forward to Your Feedback**
>
> Dear Reviewer xbbd,
>
> Thank you once again for your detailed comments on our paper. We greatly appreciate the feedback you provided and have carefully addressed the questions you raised in our latest response.
>
> With the revision deadline approaching, we would greatly appreciate it if you could provide feedback on our updated answers to ensure that we have adequately addressed your concerns. If there are any additional clarifications or further questions, please feel free to let us know.
>
> Thank you very much for your time and kind consideration!
>
> Best regards,
>
> The Authors

---

> ### Author Response · Authors · 2024-12-02
> **Follow-Up on Our Additional Response**
>
> Dear Reviewer xbbd,
>
> I hope this message finds you well. We are writing to kindly follow up on the additional clarifications we provided in response to your latest comments.
>
> As the discussion period is coming to an end, we wanted to ensure that our updated responses have adequately addressed your concerns. We have carefully addressed all your additional questions to the best of our ability and believe our answer resolves the issues you raised.
>
> If you feel that our responses resolve your concerns, we would be grateful if you could consider reflecting this in your final evaluation. Conversely, if there are still aspects requiring further improvement or clarification, we would deeply value your additional feedback to further improve our work.
>
> Your input is crucial to the quality of our work, and we sincerely appreciate the time and effort you have already devoted to this review. We look forward to hearing from you before the discussion period closes.
>
> Thank you very much for your time and kind consideration.
>
> Best regards,\
> The Authors

---

### Official Review · Reviewer_M92W · 2024-11-02

**Soundness:** 3
**Presentation:** 3
**Contribution:** 3
**Rating:** 6
**Confidence:** 3

**Summary:**

This paper proposes a hierarchical method for LLM-based task planning. Rather than using a single decision-making process as in ReAct, ReActTree decomposes a complex task into different subtasks at various levels using a tree structure. Additionally, to manage the complexity in context introduced by the tree structure, the paper proposes a memory module from which related experiences can be retrieved as context.

**Strengths:**

Decomposing a complex task into a tree structure makes a lot of sense to me. Humans tend to think recursively, and a tree structure can best represent that recursive nature. Retrieving related experiences is also very effective, as having too much unrelated context is not helpful.

**Weaknesses:**

Despite its usefulness, the proposed method can be overly complex.

**Questions:**

I am curious if the authors have studied the percentage of tasks that extend beyond three layers when expanding the tree. What are the statistics regarding tasks and the number of layers in their expanding trees?

---

> ### Author Response · Authors · 2024-11-22
>
> Thank you for your valuable feedback and thoughtful questions. We appreciate your insights and hope our responses address your concerns.
>
> > **Despite its usefulness, the proposed method can be overly complex.**
>
> The proposed method may initially appear complex. However, this complexity stems from the nature of the complex, long-horizon task planning that inherently require sophisticated reasoning and planning mechanisms.
>
> Our design, including the use of hierarchical tree structures and memory components, is carefully motivated to tackle these challenges effectively. For example, as shown in Table 1, ReAcTree with QWEN-2.5 72B achieves a **39 percentage-point improvement** in success rates compared to ReAct. We believe this level of complexity is justified by the significant performance improvements demonstrated in our experiments, which are critical for addressing the challenges of long-horizon tasks.
>
> > **I am curious if the authors have studied the percentage of tasks that extend beyond three layers when expanding the tree. What are the statistics regarding tasks and the number of layers in their expanding trees?**
>
> Thank you for your insightful question. For successful tasks performed by ReAcTree using the LLaMA-3 70B model, the **average depth of the tree was 3.92**, with a **minimum depth of 3** and a **maximum depth of 5**. We hope this clarifies the structural characteristics of the expanding trees in our experiments.
>
> ---
> We are grateful for your insightful feedback, and we believe that addressing these issues will enhance the overall quality of the paper. If you have any further suggestions or remaining concerns, please do not hesitate to communicate them.

---

> ### Author Response · Authors · 2024-11-27
> **Looking Forward to Your Feedback**
>
> Dear Reviewer M92W,
>
> Thank you once again for your thoughtful comments on our paper. We have carefully considered your comments in our rebuttal and made updates accordingly.
>
> With the revision deadline approaching, we kindly request your feedback on our submitted response. We hope our response has adequately addressed your concerns. If there are any additional clarifications or further questions, please do not hesitate to let us know.
>
> Thank you very much for your time and kind consideration!
>
> Best regards,
> The Authors

---

### Official Review · Reviewer_6YAS · 2024-11-04

**Soundness:** 2
**Presentation:** 3
**Contribution:** 2
**Rating:** 3
**Confidence:** 4

**Summary:**

This paper proposes an LLM-based task planning algorithm that decomposes tasks into sub-tasks following a tree structure. This paper is a tree-search-based extension of the ReAct algorithm, which jointly generates reasoning traces (through chain-of-thought) and action/sub-goal sequences to improve performance on planning tasks.

The proposed algorithm ReAcTree enhances ReAct by allowing the agent to generate multiple possible sub-tasks at a state. The candidate sub-tasks will be explored in subsequent steps. When exploring/solving each sub-task, the agent also leverages relevant experiences explored in the past by retrieving them according to a similarity metric (similar to retrieval augmented generation). ReAcTree is tested against ReAct on one dataset and demonstrated improved performance.

**Strengths:**

- Solving complex real-world planning tasks is very important and relevant to AI research.
- This paper incorporates the idea of tree search into the ReAct algorithm and achieves improved performance.

**Weaknesses:**

- The idea of using tree search or graph search to augment LLMs’ reasoning/planning capabilities has been explored by prior work (e.g., [1,2]). It is hard to evaluate the contribution of this paper without a detailed comparison/discussion of these very related approaches. If applicable, it is highly recommended to also include these methods as baselines in the experiments.

- Although ReAcTree can improve upon ReAct as shown by the experiments, it is unclear the difference between their efficiency. For example, by increasing the “maximum decision length” (terminology borrowed from the paper), the performance of ReAcTree will very likely also increase. However, this will render the algorithm more inefficient. If ReAct is given such additional time/compute, it can at least rerun the same algorithm multiple times to increase the success rate. Therefore, I think it is better to compare different methods with both objectives (performance and efficiency) in mind.

- It would be nice to see how much does the tree search component alone contribute to the overall performance. That is, if no memory (episodic memory and working memory in the paper) is used, how does ReAcTree performs compared to ReAct.

[1] Yao, Shunyu, Dian Yu, Jeffrey Zhao, Izhak Shafran, Tom Griffiths, Yuan Cao, and Karthik Narasimhan. "Tree of thoughts: Deliberate problem solving with large language models." Advances in Neural Information Processing Systems 36 (2024).

[2] Besta, Maciej, Nils Blach, Ales Kubicek, Robert Gerstenberger, Michal Podstawski, Lukas Gianinazzi, Joanna Gajda et al. "Graph of thoughts: Solving elaborate problems with large language models." In Proceedings of the AAAI Conference on Artificial Intelligence, vol. 38, no. 16, pp. 17682-17690. 2024.

**Questions:**

What's the runtime of the proposed algorithm and the baselines? How would they scale with longer runtime?

---

> ### Author Response · Authors · 2024-11-23
>
> We appreciate your thorough review and constructive feedback. Below, we address your comments on the weaknesses and questions.
>
> > **The idea of using tree search or graph search to augment LLMs’ reasoning/planning capabilities has been explored by prior work (e.g., [1,2]). It is hard to evaluate the contribution of this paper without a detailed comparison/discussion of these very related approaches. If applicable, it is highly recommended to also include these methods as baselines in the experiments.**
>
> Thank you for raising this insightful point. We have clarified the distinctions between ReAcTree and existing tree-based planning methods, discussed their limitations, and highlighted the unique strengths of ReAcTree:
>
> 1. Key Differences: Unlike prior approaches such as ToT [1] and GoT [2], which focus on constructing **action/thought trees** to evaluate potential future actions or thoughts and select the optimal next step, ReAcTree introduces an **LLM agent tree** specifically designed for hierarchical task decomposition. When faced with a complex goal, ReAcTree breaks it down into manageable subgoals, assigns these subgoals to specialized LLM agent nodes, and employs control flow nodes (e.g., sequence, fallback) to determine their execution strategy.
>
> 2. Limitations of Previous Work:
> - ToT [1] and GoT [2]: Both methods rely heavily on simulating future outcomes and evaluate multiple potential paths. This requires precise state predictions, which are infeasible in dynamic, real-world environments such as households where outcomes are inherently unpredictable. Consequently, these approaches have primarily been applied to deterministic or predictable settings (e.g., reasoning tasks like Game of 24 or Sorting), where future states are relatively straightforward to anticipate.
> - LLM-MCTS [3]: While LLM-MCTS has been applied in a household environment (VirtualHome), it depends on ground-truth transition functions to simulate future states. This dependency limits its practical application in real-world scenarios where such transition functions are unavailable.
> - Tree-Planner [4]: Tree-Planner provides a more grounded approach by constructing action trees and using observation-based grounded decoding. However, it assumes the reversibility of certain actions (e.g., "pick"-"place," "switch on"-"switch off") for backtracking. This assumption does not hold in complex scenarios like ALFRED, where irreversible actions (e.g., "slice") are required. Additionally, Tree-Planner lacks the capability to decompose complex tasks into subgoals, tackling the entire task as a single problem.
>
> [1] Yao, Shunyu, et al. "Tree of thoughts: Deliberate problem solving with large language models." Advances in Neural Information Processing Systems 36 (2024). \
> [2] Besta, Maciej, et al. "Graph of thoughts: Solving elaborate problems with large language models." Proceedings of the AAAI Conference on Artificial Intelligence. Vol. 38. No. 16. 2024. \
> [3] Zhao, Zirui, Wee Sun Lee, and David Hsu. "Large language models as commonsense knowledge for large-scale task planning." Advances in Neural Information Processing Systems 36 (2024). \
> [4] Hu, Mengkang, et al. "Tree-planner: Efficient close-loop task planning with large language models." arXiv preprint arXiv:2310.08582 (2023).
>
> 3. Unique Strengths of ReAcTree:
> - **No State Prediction**: ReAcTree eliminates the need for look-ahead or state evaluations, which are impractical for household task planning due to the unpredictable nature of state transitions.
> - **Dynamic Subgoal Decomposition**: By dynamically breaking down complex tasks into simpler subgoals and assigning specialized agent nodes to each, ReAcTree reduces the complexity of each agent’s task. This modular approach also allows for the effective use of task-specific in-context examples tailored to individual subgoals.
> - **Control Flow Integration**: The inclusion of control flow nodes enables efficient task execution without requiring exhaustive tree traversal or backtracking.
>
> To address this feedback, we have revised **Section 2** to provide a more detailed comparison with these methods. While these approaches tackle different challenges, we acknowledge their contributions and see potential for future hybrid frameworks that combine hierarchical decomposition with selective tree search methodologies.

---

> ### Author Response · Authors · 2024-11-23
>
> (continued)
> > **Although ReAcTree can improve upon ReAct as shown by the experiments, it is unclear the difference between their efficiency. For example, by increasing the “maximum decision length” (terminology borrowed from the paper), the performance of ReAcTree will very likely also increase. However, this will render the algorithm more inefficient. If ReAct is given such additional time/compute, it can at least rerun the same algorithm multiple times to increase the success rate. Therefore, I think it is better to compare different methods with both objectives (performance and efficiency) in mind.**
>
> In response to your concern regarding efficiency, we have carefully structured our experiments and analysis to ensure fairness and provide meaningful comparisons:
>
> 1. **Same Maximum Decision Count**: In all our experiments, both ReAct and ReAcTree were evaluated under the same maximum decision count of 199 for a fair comparison. This constraint ensures that tasks requiring long decision steps were equally challenging for both methods. As described in **Section 5.1** of the revised manuscript, this uniform constraint was consistently applied across all experiments.
>
> 2. **199 is a Sufficiently Large Limit**: The maximum decision count of **199** was chosen to be large enough to accommodate the vast majority of tasks. Our failure analysis in **Appendix E** of the revised manuscript shows that **ReAcTree with working memory had zero failures** due to decision limit, while **ReAcTree without working memory had only one failure** for this reason. This demonstrates that the chosen decision limit was more than sufficient for nearly all tasks in WAH-NL.
>
> 3. Efficiency analysis: To further address your concern, we analyzed the **average number of decision steps** required by ReAct and ReAcTree for tasks successfully completed by both methods (with working memory). Using the QWEN-2.5 72B model, the **average number of decision steps required by ReAct and ReAcTree were 58.08 and 75.00, respectively**. While ReAcTree requires a slightly higher number of decisions, this is primarily due to the additional reasoning and expanding actions. These additional steps enable ReAcTree to effectively address more complex tasks. Importantly, ReAcTree achieves higher success rates without excessively increasing computational overhead, demonstrating its efficiency in solving long-horizon tasks.
>
> > **It would be nice to see how much does the tree search component alone contribute to the overall performance. That is, if no memory (episodic memory and working memory in the paper) is used, how does ReAcTree performs compared to ReAct.**
>
> To address this, we conducted a memory ablation study using the QWEN-2.5 72B model to evaluate the standalone effectiveness of ReAcTree’s hierarchical planning mechanism across four memory configurations: 1) both episodic memory (EM) and working memory (WM), 2) EM only, 3) WM only, and 4) No memory. In the absence of EM, task planning relied on a single fixed in-context example for both ReAct and ReAcTree. For consistency, the same task from the WAH-NL training set, "put fridge," was selected, and human-annotated trajectories were collected for both methods.
>
> Note that the results for configurations 1 and 2 have already been reported in **Table 1 of Section 5.4** in the manuscript. In the revised manuscript, we clarified that WM refers to working memory. The experimental results for all configurations are summarized below:
>
> | Memory Configuration | ReAcTree (SR / SSR) | ReAct (SR / SSR)  |
> |-----------------------|---------------------|-------------------|
> | **Both memory**       | 63 / 79.37         | 24 / 48.43       |
> | **EM only**           | 51 / 70.95         | 20 / 45.13       |
> | **WM only**           | 15 / 31.72         | 12 / 38.45        |
> | **No memory**         | 19 / 35.32         | 14 / 33.18        |
>
> The results of the memory ablation study demonstrate that ReAcTree consistently outperforms ReAct across almost all memory configurations, showing its robust hierarchical planning capabilities. EM plays a critical role in performance improvement, with configurations that include EM ("Both memory" and "EM only") significantly outperforming those without it. However, the "WM only" and "No memory" configurations both relied on a single fixed in-context example, and the performance in these configurations could vary depending on the choice of this example.

---

> ### Author Response · Authors · 2024-11-23
>
> (continued)
> > **What's the runtime of the proposed algorithm and the baselines? How would they scale with longer runtime?**
>
> The computational cost of both ReAct and ReAcTree **primarily arises from the LLM's decision-making process**. Therefore, we compare the average number of decision steps for the two methods. As mentioned in the response regarding maximum decision count and efficiency, using the QWEN-2.5 72B model, the average number of decision steps for successfully completed tasks was **58.08 for ReAct** and **75.00 for ReAcTree**, indicating that ReAcTree incurs higher computational cost. However, while ReAct continuously accumulates decision-making results and observations, ReAcTree resets the input prompt whenever a new agent node is created, providing a slight computational advantage.
>
> To provide a more concrete comparison, we measured the runtime for one successfully completed task. For this task, **ReAcTree performed 88 decisions, taking 579 seconds**, while **ReAct performed 60 decisions, taking 310 seconds**. The longer runtime of ReAcTree is directly attributed to its additional decisions, reflecting its hierarchical task decomposition approach.
>
> While ReAcTree requires more decision steps and thus incurs higher runtime costs, it achieves significantly higher success rates and demonstrates robustness in handling complex tasks. As tasks grow more complex, the hierarchical structure of ReAcTree is expected to provide better scalability by managing subgoals independently, although this will naturally involve higher computational overhead.
>
> ---
>
> We are grateful for your insightful feedback, and we believe that addressing these issues will enhance the overall quality of the paper. If you have any further suggestions or remaining concerns, please do not hesitate to communicate them.

---

> ### Author Response · Authors · 2024-11-27
> **Looking Forward to Your Feedback**
>
> Dear Reviewer 6YAS,
>
> Thank you once again for your thoughtful comments on our paper. We have carefully considered your comments in our rebuttal and made updates accordingly.
>
> With the revision deadline approaching, we kindly request your feedback on our submitted response. We hope our response has adequately addressed your concerns. If there are any additional clarifications or further questions, please do not hesitate to let us know.
>
> Thank you very much for your time and kind consideration!
>
> Best regards,
> The Authors

---

> ### Author Response · Authors · 2024-12-02
> **Follow-Up on Our Revised Manuscript and Response**
>
> Dear Reviewer 6YAS,
>
> I hope this message finds you well. We are writing to kindly follow up on our revised manuscript and response to your concerns, which you have been assigned to review. We sincerely appreciate the time and effort you have taken to review our work and provide your initial evaluation.
>
> As the discussion period is coming to an end, we wanted to ensure that our response has adequately addressed your concerns. We have carefully addressed all your comments to the best of our ability and believe our response resolves the issues you raised.
>
> If you feel that our response adequately addresses your concerns, we would be grateful if you could consider reflecting this in your final evaluation. Conversely, if there are still aspects requiring further clarification, we would deeply value your additional feedback to further improve our work.
>
> Your input is incredibly important to us, and we hope to hear from you before the discussion period closes. Thank you very much for your time and kind consideration.
>
> Best regards, \
> The Authors

---

### Author Response · Authors · 2024-11-25
**Overall Response to All Reviewers (Revision Uploaded)**

We sincerely thank all reviewers for their valuable time and thoughtful feedback. In response to your comments, we have carefully revised our manuscript and highlighted the changes in blue in the updated draft. Below is a summary of the main revisions:

- Tree Search-Based Planning with LLMs (**Section 2**): In response to Reviewer 6YAS and Reviewer HETB, we expanded our discussion of related methods such as ToT, GoT, LLM-MCTS, and Tree-Planner. This enhancement highlights ReAcTree's distinctive approach to hierarchical task decomposition and dynamic subgoal allocation.

- Memory Systems Revision (**Section 4.2**): We clarified the interaction between episodic and working memory within agent nodes, addressing integration and retrieval mechanisms, as suggested by Reviewer xbbd.

- Maximum Decision Count Specification (**Section 5.1**): To ensure fair and consistent experimental conditions, we explicitly stated a uniform maximum decision count of 199 across all methods, incorporating Reviewer 6YAS's feedback.

- Computational Cost Analysis (**Section 5.2**): At the suggestion of Reviewer 6YAS and Reviewer HETB, we added an analysis comparing decision steps between ReAcTree and ReAct to demonstrate computational cost differences.

- Failure Case Analysis (**Appendix E**): Following Reviewer HETB’s request, we analyzed failure cases in detail, categorized by type. This analysis highlights the most prevalent failures, demonstrates how working memory reduces failure across multiple categories, and identifies persistent challenges.

We hope these revisions adequately address your concerns and enhance the overall quality of our manuscript. We are open to any additional suggestions or questions you may have.

---

### Meta-Review · Area_Chair_yHd9 · 2024-12-21

**Metareview:**

The paper studies LLM-based agents for long-horizon tasks, and proposes a hierarchical decomposition technique
(ReAcTree) that substantially outperforms the ReAct baseline on simulated tasks in a benchmark dataset (WAH-NL).
All the reviewers agreed that the paper studies a well-motivated and timely problem,
but found the paper's contributions to be borderline for publication at ICLR due to the incremental significance
and room for improvement in the execution of the empirical findings.

1. Missing baselines: Several techniques have been proposed to expend tokens at inference time to improve LLM agents'
task performance, and during the rebuttal the authors discussed a few of them and how ReAcTree's emphasis differs from them.
Benchmarking those other tree-based methods empirically will provide the evidence required to claim that
ReAcTree's hierarchical task decomposition is indeed superior to other tree-based approaches.

2. Benefits and limits of hierarchical decomposition of tasks: The core argument of the paper is that, through an appropriate
hierarchical decomposition, many long horizon tasks can be tackled by *independently* solving different sub-tasks.
Identifying the limits (e.g. one reviewer pointed out that the decomposition may be incorrect; another failure mode is that
no such decomposition may exist for some task; another reviewer question was that the number of sub-tasks may need to be dynamically determined, etc.)
will help better understand ReAcTree's applicability to LLM planning.

3. Understanding the cost of ReAcTree vs. other baselines. The authors highlight that they performed a fair comparison in terms of decision counts.
However reviewers pointed out that there are many meaningful notions of cost: e.g. tokens used, latency in getting to the final answer
(both of which are correlated with decision counts; but different tree-based methods could rank differently according to these costs).

4. Finally, the paper can be strengthened by including a discussion around newer inference-compute-scaling models like OpenAI o1,
and the enduring contributions of approaches for explicit decomposition like ReAcTree over and above those models which are fine-tuned to perform implicit task decompositions.

**Additional Comments On Reviewer Discussion:**

During the rebuttal, authors provided an ablation experiment to isolate the contributions of tree search vs. the memory mechanisms
to ReAcTree's performance gains over ReAct. They also tried other similarity metrics for memory retrieval (in response to a reviewer question)
and found that their original retrieval metric worked best.
However these experiments did not answer the broader question raised by reviewers around scale (does the recall and precision of the memory retrieval
remain acceptable as the number of experiences in the storage increases?)

---

### Decision · Program_Chairs · 2025-01-22

Reject